# Revisiting Agnostic Boosting

**Arthur da Cunha**
Aarhus University
dac@cs.au.dk

**Mikael Møller Høgsgaard**
Aarhus University
hogsgaard@cs.au.dk

**Andrea Paudice**
Aarhus University
apaudice@cs.au.dk

**Yuxin Sun**
Aarhus University
yxsau@cs.au.dk

## Abstract

Boosting is a key method in statistical learning, allowing for converting weak learners into strong ones. While well studied in the realizable case, the statistical properties of weak-to-strong learning remain less understood in the agnostic setting, where there are no assumptions on the distribution of the labels. In this work, we propose a new agnostic boosting algorithm with substantially improved sample complexity compared to prior works under very general assumptions. Our approach is based on a reduction to the realizable case, followed by a margin-based filtering of high-quality hypotheses. Furthermore, we show a nearly-matching lower bound, settling the sample complexity of agnostic boosting up to logarithmic factors.

## 1 Introduction

Binary classification under the Probably Approximately Correct (PAC) learning model is perhaps the most fundamental paradigm of statistical learning theory.

In the *realizable* version of the problem, we consider an input space $\mathcal{X}$, a known hypothesis class $\mathcal{F} \subseteq \{\pm 1\}^{\mathcal{X}}$, an *unknown* target classifier $f \in \mathcal{F}$, and a training sequence $\boldsymbol{S} = \big((\mathbf{x}_1, f(\mathbf{x}_1)), \ldots, (\mathbf{x}_m, f(\mathbf{x}_m))\big) \in (\mathcal{X} \times \{\pm 1\})^m$ of independent samples drawn from an *unknown* but fixed distribution $\mathcal{D}$ and each labeled according to $f$. The objective is to ensure that, for any desired accuracy and confidence parameters, $\varepsilon, \delta > 0$, the algorithm can, with probability at least $1 - \delta$, learn from a training sequence of size $m(\varepsilon, \delta)$ and find a classifier $h$ with an expected error, $\mathbb{P}_{\mathbf{x} \sim \mathcal{D}}[h(\mathbf{x}) \neq f(\mathbf{x})]$, less than $\varepsilon$. A learning algorithm achieving such is called a *strong learner* and the amount of training data $m(\varepsilon, \delta)$ necessary to reach this goal is called its *sample complexity*.

It is known that in the realizable setting the *Empirical Risk Minimization* procedure (ERM) of choosing any hypothesis $h \in \mathcal{F}$ minimizing the empirical loss $\sum_{(x,y) \in \boldsymbol{S}} \mathbb{1}\{h(x) \neq y\}$, which is $0$ in the realizable case, yields a strong learner—as long as $\mathcal{F}$ has bounded VC dimension.

An interesting question posed by Kearns and Valiant is whether it is possible to obtain a strong learner starting from humbler requirements [Kearns, 1988, Kearns and Valiant, 1989]. Namely, the authors consider $\gamma$–*weak learners* that are only guaranteed to produce classifiers $w$ from a base class $\mathcal{H}$ with error lower than $1/2 - \gamma$ for some $\gamma > 0$. More precisely, for any distribution $\mathcal{Q}$ over $\mathcal{X}$, the weak learner produces, with probability at least $1 - \delta_0$, hypothesis $w$ such that

$$\mathbb{P}_{\mathbf{x} \sim \mathcal{Q}}[w(\mathbf{x}) \neq f(\mathbf{x})] < 1/2 - \gamma, \tag{1}$$

thus having $\gamma$ *advantage* over random guessing. Answering whether such simple learners could be boosted to achieve arbitrarily good generalization led to intense research and, eventually, to many such *weak-to-strong* learning algorithms, including the celebrated ADABOOST [Freund and Schapire, 1997].

Realizability can, however, be too strong of a hypothesis. To capture a broader scenario, we consider the *agnostic* learning model [Haussler, 1992, Kearns et al., 1994], where the samples are generated by an arbitrary distribution $\mathcal{D}$ over $\mathcal{X} \times \mathcal{Y}$. Thus, obtaining a learner with arbitrarily small error

$$\mathrm{err}_{\mathcal{D}}(h) \coloneqq \mathbb{P}_{(\mathbf{x}, \mathbf{y}) \sim \mathcal{D}}[h(\mathbf{x}) \neq \mathbf{y}]$$

39th Conference on Neural Information Processing Systems (NeurIPS 2025).

in this setting is not always possible. Accordingly, the goal of the learner is to find a classifier $h$ with error close to that of the best classifier in a reference class $\mathcal{F}$. Formally, in the agnostic setting, we say that a learning algorithm is a *strong learner* if, for any $\varepsilon, \delta > 0$, given $m(\varepsilon, \delta)$ examples from $\mathcal{D}$, with probability at least $1 - \delta$ the learner outputs a classifier $h$ such that $\mathrm{err}_{\mathcal{D}}(h) \leq \inf_{f \in \mathcal{F}} \mathrm{err}_{\mathcal{D}}(f) + \varepsilon$. Hereon, we assume without loss of generality that the infimum error is achieved by some $f^{\star} \in \mathcal{F}$. It is worth noting that ERM is still a strong learner in this setting.

A natural question, first posed by Ben-David et al. [2001], is whether it is possible to boost the performance of weak learners in the agnostic setting. To formalize this question, it is useful to measure the performance of hypotheses in terms of correlation under the data distribution:

$$\mathrm{corr}_{\mathcal{D}}(h) := \mathbb{E}_{(\mathbf{x}, \mathbf{y}) \sim \mathcal{D}}[\mathbf{y} \cdot h(\mathbf{x})].$$

Notice that for binary hypothesis $h$, $\mathrm{corr}_{\mathcal{D}}(h) = 1 - 2\,\mathrm{err}_{\mathcal{D}}(h)$ so maximizing the correlation is equivalent to minimizing the error. With that, we have the following definition.

**Definition 1.1** (Agnostic Weak Learner). Let $\gamma, \varepsilon_0, \delta_0 \in [0, 1]$, $m_0 \in \mathbb{N}$, $\mathcal{F} \subseteq \{\pm 1\}^{\mathcal{X}}$, and $\mathcal{H} \subseteq [-1, 1]^{\mathcal{X}}$. A learning algorithm $\mathcal{W} \colon (\mathcal{X} \times \{\pm 1\})^* \to \mathcal{H}$ is an agnostic weak learner with advantage parameters $\gamma$ and $\varepsilon_0$, failure probability $\delta_0$, sample complexity $m_0$, reference hypothesis class $\mathcal{F}$ and base hypothesis class $\mathcal{H}$ iff: for any distribution $\mathcal{D}$ over $\mathcal{X} \times \{\pm 1\}$, given sample $\boldsymbol{S} \sim \mathcal{D}^{m_0}$, with probability at least $1 - \delta_0$ over $\boldsymbol{S}$ the hypothesis $\mathbf{w} = \mathcal{W}(\boldsymbol{S})$ satisfies that

$$\mathrm{corr}_{\mathcal{D}}(\mathbf{w}) \geq \gamma \cdot \sup_{f \in \mathcal{F}} \mathrm{corr}_{\mathcal{D}}(f) - \varepsilon_0.$$

For short, we call such an algorithm a $(\gamma, \varepsilon_0, \delta_0, m_0, \mathcal{F}, \mathcal{H})$ *agnostic weak learner*, and we may omit some of those parameters when the context allows for no ambiguity.

At the cost of some verbosity, the definition above is quite general. Indeed, it encompasses the original definition from Ben-David et al. [2001], and, from it, one can readily recover the usual definition of weak learner.[1] The generality of Definition 1.1 aims to capture as much as possible of the diverse set of alternatives proposed in the literature, thus allowing for a fairer comparison with previous works, as discussed in Section 1.1. Despite the broad definition, we obtain the following lower bound on the sample complexity of learning under the agnostic model stemming from Definition 1.1.

**Theorem 1.2.** *There exist universal constants $C_1, C_2, C_3, C_4 > 0$ for which the following holds. Given any $L \in (0, 1)$, any $\gamma, \varepsilon_0, \delta_0 \in (0, 1]$, and any integer $d \geq C_1 \ln(1/\gamma^2)$, for $m_0 = \left\lceil C_2 d \ln(\frac{1}{\delta_0 \gamma^2}) / (\varepsilon_0^2 \ln(\frac{1}{\gamma^2})) \right\rceil$ there exist domain $\mathcal{X}$, reference class $\mathcal{F} \subseteq \{\pm 1\}^{\mathcal{X}}$, and base class $\mathcal{H} \subseteq \{\pm 1\}^{\mathcal{X}}$ with $\mathrm{VC}(\mathcal{H}) \leq d$, such that there exists a $(\gamma, \varepsilon_0, \delta_0, m_0, \mathcal{F}, \mathcal{H})$ agnostic weak learner and, yet, the following also holds. For any learning algorithm $\mathcal{A} \colon (\mathcal{X} \times \{\pm 1\})^* \to \{\pm 1\}^{\mathcal{X}}$ there exists a distribution $\mathcal{D}$ over $\mathcal{X} \times \{\pm 1\}$ such that $\mathrm{corr}_{\mathcal{D}}(f^{\star}) = L$ and for sample size $m \geq C_3 \frac{d}{\gamma^2(1-L)} \frac{1}{L^2}$ we have that*

$$\mathbb{E}_{\boldsymbol{S} \sim \mathcal{D}^m}[\mathrm{corr}_{\mathcal{D}}(\mathcal{A}(\boldsymbol{S}))] \leq \mathrm{corr}_{\mathcal{D}}(f^{\star}) - \sqrt{C_4(1 - \mathrm{corr}_{\mathcal{D}}(f^{\star})) \cdot \frac{d}{(\gamma - \varepsilon_0)^2 m \ln(1/\gamma)}}.$$

We know from classic results that agnostically learning relative to a reference class with VC dimension $d$ implies an excess error of $\Omega(\sqrt{d/m})$. Accordingly, the basic idea behind the bound above is to construct a base class $\mathcal{H}$ with $\mathrm{VC}(\mathcal{H}) \leq d$ that is sufficient to agnostically weak learn a reference class $\mathcal{F}$ with VC dimension of at least $d/\gamma^2$ so that learning relative to $\mathcal{F}$ would incur an excess error of $\Omega(\sqrt{d/(\gamma^2 m)})$. However, with our construction we were only able to show that $\mathrm{VC}(\mathcal{F}) \geq d/(\gamma^2 \ln(1/\gamma))$, leading to the extra logarithmic factor in the bound. Our argument draws inspiration from Alon et al. [2023] and is deferred to Appendix E, which also contains versions of the theorem with different trade-offs.

We show that the bound from Theorem 1.2 is nearly tight by providing an algorithm that matches it up to logarithmic factors. The following summarizes the statistical properties of the method.

---

[1]To recover the definition of Ben-David et al. [2001], it suffices to restrict definition Definition 1.1 to the case $\gamma = 1$ and $\mathcal{H}$ being a binary class. If, instead, $\mathcal{D}$ is realized by some $f \in \mathcal{F}$, so that $\sup_{f \in \mathcal{F}} \mathrm{corr}_{\mathcal{D}}(f) = 1$, we can use that $\mathrm{corr}_{\mathcal{D}}(h) = 1 - 2\,\mathrm{err}_{\mathcal{D}}(h)$ to obtain a $(\frac{\gamma - \varepsilon_0}{2})$–weak learner, as in Eq. (1).

**Theorem 1.3.** *There exist universal constants $C, c > 0$ and learning algorithm $\mathcal{A}$ such that the following holds. Let $\mathcal{W}$ be a $(\gamma, \varepsilon_0, \delta_0, m_0, \mathcal{F}, \mathcal{H})$ agnostic weak learner. If $\gamma > \varepsilon_0$ and $\delta_0 < 1$, then, for all $\delta \in (0,1)$, $m \in \mathbb{N}$, and distribution $\mathcal{D}$ over $\mathcal{X} \times \{\pm 1\}$, given training sequence $\boldsymbol{S} \sim \mathcal{D}^m$, we have that $\mathcal{A}$ given $(\boldsymbol{S}, \mathcal{W}, \delta, \delta_0, m_0)$, with probability at least $1 - \delta$ over $\boldsymbol{S}$ and the internal randomness of the algorithm, returns $\mathbf{v}$ satisfying that*

$$\mathrm{corr}_{\mathcal{D}}(\mathrm{sign}(\mathbf{v})) \geq \mathrm{corr}_{\mathcal{D}}(f^\star) - \sqrt{C(1 - \mathrm{corr}_{\mathcal{D}}(f^\star)) \cdot \beta} - C \cdot \beta,$$

*where $\beta = \frac{\hat{d}}{(\gamma - \varepsilon_0)^2 m} \cdot \mathrm{Ln}^{3/2}\left(\frac{(\gamma - \varepsilon_0)^2 m}{\hat{d}}\right) + \frac{1}{m} \ln \frac{\ln m}{\delta}$ with $\mathrm{Ln}(x) := \ln(\max\{x, e\})$ and $\hat{d} = \mathrm{fat}_{c(\gamma - \varepsilon_0)}(\mathcal{H})$ being the fat-shattering[2] dimension of $\mathcal{H}$ at level $c(\gamma - \varepsilon_0)$.*

The bound above improves on previous results by a polynomial factor, as detailed in Section 1.1. The theorem incorporates the fact that Definition 1.1 allows for non-binary weak hypotheses, expressing the bounds in terms of the fat-shattering dimension of $\mathcal{H}$, which reduces to $\mathrm{VC}(\mathcal{H})$ in the binary case. Moreover, the bound in Theorem 1.3 desirably interpolates between the agnostic, $\mathrm{corr}_{\mathcal{D}}(f^\star) < 1$, and the realizable, $\mathrm{corr}_{\mathcal{D}}(f^\star) = 1$, settings. Lastly, since random guessing leads to correlation 0, the theorem above establishes that any non-trivial weak learner (with $\varepsilon_0 < \gamma$) can be boosted to a strong learner in the agnostic setting. As discussed in Section 1.1, previous results were not sufficient to ensure this general fact.

We highlight that the method underlying Theorem 1.3 attains the performance ensured by the theorem under remarkably mild assumptions. In addition to leveraging a fairly general weak learner (see Definition 1.1), the method does not require direct access to the reference class $\mathcal{F}$ or the base class $\mathcal{H}$, relying only on the weak hypotheses returned by $\mathcal{W}$. The technique also does not require knowledge of the advantage parameters $\gamma$ and $\varepsilon_0$, thus preserving the characteristic adaptability of boosting.

As detailed in Section 3, while nearly optimal in terms of sample complexity, the algorithm proposed is not computationally efficient. Hence, on this regard, it is comparable to ERM which can be shown to be a strong agnostic learner with near-optimal sample complexity [Shalev-Shwartz and Ben-David, 2014] (albeit, demanding direct access to the reference class $\mathcal{F}$ and a way to bound its VC dimension). However, while ERM is bound to be inefficient even for simple classes (unless P = NP) [Bartlett and Ben-David, 1999, Ben-David et al., 2000], boosting can, in principle, attain optimal sample complexity within manageable computational cost. By showing that agnostic weak-to-strong learning is possible with nearly optimal sample complexity under minimal assumptions, we take a step towards agnostic boosting algorithms that are both computationally and statistically efficient.

## 1.1 Related Works

The work that is closest to ours is the recent contribution by Ghai and Singh [2024]. Notably, it employs a definition of agnostic weak learner very close to Definition 1.1. The authors devise a method requiring $\tilde{\mathcal{O}}\big(\mathrm{VC}(\mathcal{H})/(\varepsilon^3 \gamma^3)\big)$ samples to produce a classifier $\mathbf{v}$ satisfying

$$\mathrm{corr}_{\mathcal{D}}(\mathbf{v}) \geq \mathrm{corr}_{\mathcal{D}}(f^\star) - \frac{2\varepsilon_0}{\gamma} - \varepsilon \tag{2}$$

with high probability.[3] Crucially, for their algorithm to yield a strong learner, the advantage parameters ($\gamma$ and $\varepsilon_0$) must satisfy that $\varepsilon_0 = \mathcal{O}(\varepsilon\gamma)$. In contrast, Theorem 1.3 shows that as long as $\gamma > \varepsilon_0$, that is, for any non-trivial weak learner, it is possible to achieve weak-to-strong learning in the agnostic setting. Under the further mild assumption that $\gamma \geq \varepsilon_0/2$ we obtain a sample complexity of order $\tilde{\mathcal{O}}\big((1 - \mathrm{corr}_{\mathcal{D}}(f^\star)) \cdot \hat{d}/(\varepsilon^2 \gamma^2) + \hat{d}/(\varepsilon\gamma^2)\big)$, improving on Ghai and Singh [2024] by a polynomial factor. Also, Theorem 1.3 provides a bound in terms of the fat-shattering dimension of the base class $\mathcal{H}$, which reduces to the VC dimension when only considering binary classifiers while also allowing more general hypothesis classes.

A body of agnostic boosting literature diverges from the original definition of weak learner from Ben-David et al. [2001]. Most notably, the works Kalai and Kanade [2009], Brukhim et al. [2020], Feldman [2010] propose agnostic boosting methods based on the re-labeling of examples rather than

---

[2]We recall the definition of the fat-shattering dimension and other notations in Section 2.

[3]Concurrent work by the same authors shows that with $\tilde{\mathcal{O}}(\mathrm{VC}(\mathcal{H})/(\varepsilon^2 \gamma^2))$ labelled and $\tilde{\mathcal{O}}(\mathrm{VC}(\mathcal{H})/(\varepsilon^3 \gamma^3))$ unlabelled samples one can achieve the bound in Eq. (2). See Ghai and Singh [2025].

re-weighting, as in traditional boosting and in the method we present here. While those works all introduce different definitions of weak learner, the re-labeling strategy allows those definitions to only require that all distributions have the same marginal over $\mathcal{X}$ as the data distribution $\mathcal{D}$. Except for this aspect, we tried to make our definition as general as possible, also to better encompass the alternatives. In the following, we strive to compare our results while accounting for the different definitions used by others. We compile and further discuss the multiple definitions of agnostic weak learners in Appendix A.

Another recent work, by Brukhim et al. [2020], proposes, under a different empirical weak learning assumption (cf. Appendix A), an algorithm that after $T$ rounds of boosting produces a classifier $v$ with expected empirical correlation satisfying that

$$\mathbb{E}\left[\frac{1}{m}\sum_{i=1}^{m} y_i v(x_i)\right] \geq \sup_{f\in\mathcal{F}} \mathbb{E}\left[\frac{1}{m}\sum_{i=1}^{m} y_i f(x_i)\right] - \left(\frac{\varepsilon_0}{\gamma} + \mathcal{O}\left(\frac{1}{\gamma\sqrt{T}}\right)\right),$$

where $\varepsilon_0$ plays a similar role as in Definition 1.1. While this is a bound on the empirical performance of the classifier, the authors argue that one can obtain a generalization bound up to an $\varepsilon$ term with a sample complexity of $m = \tilde{\mathcal{O}}(Tm_0/\varepsilon^2)$, where $m_0$ is the sample complexity of their weak learner, usually assumed to be $\Theta(1/\varepsilon_0^2)$. Overall, this yield $m = \tilde{\mathcal{O}}(T/(\varepsilon^2\varepsilon_0^2))$. Now, for their algorithm to be a strong learner one would have to set $T = \Theta(1/(\varepsilon^2\gamma^2))$ and assume that $\varepsilon_0 = \Theta(\varepsilon\gamma)$, implying a sample complexity of $m = \tilde{\mathcal{O}}(1/(\varepsilon^6\gamma^4))$.

The work of Feldman [2010] is the hardest to compare to ours as the authors employ a definition not encompassed by Definition 1.1. Under a definition parameterized by $\alpha$ and $\gamma$, they propose a learning algorithm yielding a classifier $v$ such that

$$\mathrm{err}_{\mathcal{D}}(v) = \inf_{f\in\mathcal{F}} \mathrm{err}_{\mathcal{D}}(f) + 2\alpha + \varepsilon.$$

Our understanding is that the associated sample complexity is of order $\mathcal{O}(1/\gamma^4 + 1/\varepsilon^4)$, where $\gamma \leq \alpha$. As in the previous cases, to obtain a strong learning guarantee one has constrain the weak learner non-trivially. For Feldman [2010], the authors require that $\alpha = \Theta(\varepsilon)$. Moreover, the sample complexity of their proposed method is $\mathcal{O}(1/\varepsilon^4)$ regardless of the weak learning guarantee. In contrast, Theorem 1.3 does not require that $\gamma \leq \varepsilon$, so when the advantage $\gamma - \varepsilon_0$ is constant, the theorem ensures a sample complexity of order $\tilde{\mathcal{O}}(d/\varepsilon^2)$.

Nonetheless, we stress that Ghai and Singh [2024], Brukhim et al. [2020], Feldman [2010], Kalai and Kanade [2009] propose computationally efficient algorithms, bringing insights both to the computational and statistical aspects of agnostic boosting, while we only consider the latter.

Besides the works mentioned above, the literature on agnostic boosting includes several other with some of the most related to ours being Gavinsky [2003], Kalai and Servedio [2005], Kalai et al. [2008], Long and Servedio [2008], Chen et al. [2016].

## 2 Additional Notations

We let $\mathrm{Ln}\colon \mathbb{R} \to [1,\infty)$ be the truncated logarithm, given by $x \mapsto \ln(\max\{x, e\})$. Given a set $A$, we let $A^* := \bigcup_{n=0}^{\infty} A^n$ be the set of all finite sequences of elements of $A$. The notation $\Delta(A)$ stands for the set of all probability distributions over $A$. For a distribution $\mathcal{D} \in \Delta(A)$ and a integer $m \geq 1$, we let $\mathcal{D}^m$ be the distribution over $A^m$ obtained by taking $m$ independent samples from $\mathcal{D}$. Given real-valued functions $f, g$ and $\alpha, \beta \in \mathbb{R}$, we denote by $\alpha f + \beta g$ the mapping $x \mapsto \alpha f(x) + \beta g(x)$. We represent the set of convex combinations of at most $T \in \mathbb{N}$ functions from a family $\mathcal{F}$ as $\mathrm{conv}^T(\mathcal{F})$. That is,

$$\mathrm{conv}^T(\mathcal{F}) := \Big\{ \sum_{i\in[T]} \alpha_i f_i : \alpha_i \in [0,1], f_i \in \mathcal{F}, \sum_{i\in[T]} \alpha_i = 1 \Big\}.$$

For the entire convex hull, we write $\mathrm{conv}(\mathcal{F}) := \cup_{t=1}^{\infty} \mathrm{conv}^t(\mathcal{F})$. We let $\mathrm{sign}(x) = \mathbb{1}\{x \geq 0\} - \mathbb{1}\{x < 0\}$ and $\mathrm{sign}(f)$ denote the mapping $x \mapsto \mathrm{sign}(f(x))$. Notably, $\mathrm{sign}(0) = 1$.

For a classifier $h\colon \mathcal{X} \to \{\pm 1\}$ and a distribution $\mathcal{D}$ over $\mathcal{X} \times \{\pm 1\}$, we define $\mathcal{D}_f$ as the distribution over $\mathcal{X} \times \{\pm 1\}$ such that $\mathbb{P}_{(\mathbf{x},\mathbf{y})\sim\mathcal{D}_{f^\star}}[A] = \mathbb{P}_{(\mathbf{x},\mathbf{y})\sim\mathcal{D}}[(\mathbf{x}, f^\star(\mathbf{x})) \in A]$ for all measurable sets $A \subseteq \mathcal{X} \times \{\pm 1\}$. That is, $\mathcal{D}_f$ has the same distribution as $\mathcal{D}$ over $\mathcal{X}$ but with the labels given by $f$.

Given an input space $\mathcal{X}$, which for simplicity we always assume to be countable, let $\mathcal{D} \in \Delta(\mathcal{X} \times \mathbb{R})$. For any $\lambda \geq 0$ we let the $\lambda$-margin loss of a hypothesis $g\colon \mathcal{X} \to \mathbb{R}$ with respect to $\mathcal{D}$ be given by

$$\mathcal{L}_{\mathcal{D}}^{\lambda}(g) = \mathbb{P}_{(\mathbf{x},\mathbf{y}) \sim \mathcal{D}}[\mathbf{y} \cdot g(\mathbf{x}) \leq \lambda]$$

with the shorthand $\mathcal{L}_{\mathcal{D}}(g) \coloneqq \mathcal{L}_{\mathcal{D}}^{0}(g)$. Despite the generality of this definition, we reserve the notation $\mathrm{err}_{\mathcal{D}}(\,\cdot\,)$ for the error of binary classifiers.

Given function class $\mathcal{H} \subseteq \mathbb{R}^{\mathcal{X}}$ and $\alpha > 0$ we define the fat-shattering dimension of $\mathcal{H}$ at level $\alpha$ as the largest natural number $d = \mathrm{fat}_{\alpha}(\mathcal{H})$ such that there exist points $x_1, \ldots, x_d \in \mathcal{X}$ and level sets $r_1, \ldots, r_d \in \mathbb{R}$ satisfying the following: For all $b \in \{\pm 1\}^d$ there exists $h_b \in \mathcal{H}$ such that for all $i \in [d]$ it holds that $h_b(x_i) \geq r_i + \alpha$ if $b_i = 1$; and $h_b(x_i) \leq r_i - \alpha$ if $b_i = -1$. In words, whenever $d = \mathrm{fat}_{\alpha}(\mathcal{H})$ there exist a set of points and a set of levels, each of size $d$, such that the hypotheses in $\mathcal{H}$ can oscillate around those with margin $\alpha$.

Finally, we adopt the convention that $\arg\min$ and analogous functions resolve ties arbitrarily so to return a single element even when multiple ones realize the extremum under consideration. Also, whenever we write a set or sequence in place of a distribution, we mean the uniform distribution over that set or sequence. As an example, $\mathrm{err}_S(\,\cdot\,)$ refers to the empirical error of hypotheses on the sequence $S$. As the reader may have noticed, we use boldface letters to denote random variables.

## 3   Our argument

In this section, we overview the arguments underlying the proof of Theorem 1.3 and the associated method, Algorithm 2. The detailed proofs are deferred to the appendices.

We start, however, with a brief overview of the proof of Theorem 1.2. We know from classic results that agnostically learning relative to a reference class with VC dimension $d$ implies an excess error of $\Omega(\sqrt{d/m})$. Accordingly, the basic idea behind the Theorem 1.2 is to construct a base class $\mathcal{H}$ with $\mathrm{VC}(\mathcal{H}) \leq d$ that is sufficient to agnostically weak learn a reference class $\mathcal{F}$ with VC dimension of at least $d/\gamma^2$ so that learning relative to $\mathcal{F}$ would incur an excess error of $\Omega(\sqrt{d/(\gamma^2 m)})$. However, with our construction we were only able to show that $\mathrm{VC}(\mathcal{F}) \geq d/(\gamma^2 \ln(1/\gamma))$, leading to the extra logarithmic factor in the bound. Our argument draws inspiration from Alon et al. [2023] and is deferred to Appendix E, which also contains versions of the theorem with different trade-offs.

We now turn our attention to Theorem 1.3.

As we shall see, the difference between the advantage parameters $\gamma$ and $\varepsilon_0$ from Definition 1.1 plays a role similar to that of the *advantage* in the realizable setting. Throughout this section, we let $\theta \coloneqq \gamma - \varepsilon_0$ to both highlight this analogy and simplify the notation.

The argument is based on the abstract framework proposed in the seminal work Hopkins et al. [2024] to derive agnostic learning algorithms from their realizable counterparts. Their framework can be summarized into two steps:

1. Run the realizable learner on all possible re-labelings of the training set;
2. Among the hypotheses generated in the first step, return the one with the lowest empirical error on a validation dataset.

Some notes about this framework are in order. First, notice that the first step already requires exponential time, thus Hopkins et al. [2024] focuses only on statistical and information theoretic aspects of the problem, forgoing computational considerations, and so does our work. Second, while the framework above is quite abstract and, thus, suites many settings, obtaining concrete results from it usually requires some extensions and adaptations, as illustrated by many of the results in Hopkins et al. [2024]. This note is especially pertinent when aiming for (near) optimal bounds, which is the case for our work, and we will detail the adaptations we made to the framework later in the text.

Our proposed method can be decomposed into three steps. Our argument mirrors this separation, being organized as follows:

- We first show how to reduce the problem to the realizable boosting setting, which allows us to apply more standard techniques. We leverage those to show that enumeration can produce

an exponentially large set of hypotheses containing, with high probability, a classifier with good generalization guarantees.

- Then, we filter those hypotheses to obtain a new set with much smaller size (logarithmic), while preserving at least one good hypothesis with high probability.

- In the final step, we identify the good classifier via a standard validation procedure over the set of hypotheses obtained in the previous step.

### 3.1 Reduction to the realizable setting

Consider a training sequence $\boldsymbol{S} = \big((\mathbf{x}_1, \mathbf{y}_1), \ldots, (\mathbf{x}_m, \mathbf{y}_m)\big) \sim \mathcal{D}^m$ and, given $f \in \mathcal{F}$, let $\boldsymbol{S}_f = \big((\mathbf{x}_1, f(\mathbf{x}_1)), \ldots, (\mathbf{x}_m, f(\mathbf{x}_m))\big)$ be the re-labeling of $\boldsymbol{S}$ according to $f$. Notice that for any $f \in \mathcal{F}$ we have that $\sup_{f \in \mathcal{F}} \operatorname{corr}_{\mathcal{Q}}(f) = 1$ for any $\mathcal{Q} \in \Delta(\boldsymbol{S}_f)$. Hence, an agnostic weak learner $\mathcal{W}$ as in Definition 1.1 will, with probability at least $1 - \delta_0$ over a training sequence $\boldsymbol{S}' \sim \mathcal{Q}^{m_0}$, output a hypothesis with correlation at least $\gamma - \varepsilon_0$ under the given distribution. As this is equivalent to having a (realizable) weak learner with advantage $\theta/2$, standard realizable boosting methods can produce, with high probability, a voting classifier that approximates $f$ well—relative to the data distribution $\mathcal{D}$.

Accordingly, we begin by providing a variation of ADABOOST adapted to our settings (see Algorithm 1). It starts with a confidence amplification step to ensure that the weak learner outputs hypotheses with sufficient correlation with probability of at least $1 - \delta/T$ rather than the $1 - \delta_0$ ensured by Definition 1.1. Then, the algorithm performs a boosting step based on correlations to accommodate for weak hypotheses with the continuous range $[-1, 1]$ instead of the usual $\{\pm 1\}$.

**Input** : Training sequence $S = \big((x_1, y_1), \ldots, (x_m, y_m)\big)$, weak learner $\mathcal{W}$ and its sample complexity $m_0$, number of iterations $T$, confidence parameters $\delta, \delta_0 \in (0, 1)$

1   $D_1 = \left(\frac{1}{m}, \ldots, \frac{1}{m}\right)$
2   **for** $t \leftarrow 1$ **to** $T$ **do**
     // Confidence amplification
3      $k \leftarrow \left\lceil \frac{8}{1 - \delta_0} \cdot \ln \frac{10eT}{\delta} \right\rceil$
4      **for** $\ell \leftarrow 1$ **to** $k$ **do**
5         Sample $\boldsymbol{S}_t^\ell$ according to $\mathbf{D}_t^{m_0}$           // $m_0$ i.i.d. samples from $\mathbf{D}_t$
6         $\mathbf{h}_t^\ell \leftarrow \mathcal{W}(\boldsymbol{S}_t^\ell)$
7      $\mathbf{h}_t \leftarrow \arg\max_{\mathbf{h} \in \{\mathbf{h}_t^1, \ldots, \mathbf{h}_t^k\}} \{\operatorname{corr}_{\mathbf{D}_t}(\mathbf{h})\}$
     // Correlation based boosting step
8      $\mathbf{c}_t \leftarrow \operatorname{corr}_{\mathbf{D}_t}(\mathbf{h}_t)$
9      $\boldsymbol{\alpha}_t \leftarrow \frac{1}{2} \ln \frac{1 + \mathbf{c}_t}{1 - \mathbf{c}_t}$
10     **for** $i \leftarrow 1$ **to** $m$ **do**
11        $\mathbf{D}_{t+1}(i) \leftarrow \mathbf{D}_t(i) \exp(-\boldsymbol{\alpha}_t y_i \mathbf{h}_t(x_i))$
12     $\mathbf{Z}_t \leftarrow \sum_{i=1}^m \mathbf{D}_{t+1}(i)$
13     $\mathbf{D}_{t+1} \leftarrow \mathbf{D}_{t+1}/\mathbf{Z}_t$
14   **return** Voting classifier $\mathbf{v} = \frac{1}{\sum_{t=1}^T \boldsymbol{\alpha}_t} \cdot \sum_{t=1}^T \boldsymbol{\alpha}_t \mathbf{h}_t$

**Algorithm 1:** Modified ADABOOST

As usual for classic boosting methods, we provide a margin-based argument for the generalization properties of the classifier output by Algorithm 1. Suitably, our first lemma ensures that the algorithm outputs a hypothesis with large margins on the input training sequence.

**Lemma 3.1** (Realizable Learning Gaurantee of Algorithm 1). *Let $\gamma', \delta_0 \in (0, 1)$, and given $m, m_0 \in \mathbb{N}$, let $S \in (\mathcal{X} \times \{\pm 1\})^m$. If a learning algorithm $\mathcal{W} \colon (\mathcal{X} \times \{\pm 1\})^* \to \mathcal{H} \subseteq [-1, 1]^{\mathcal{X}}$ is such that for any $\mathcal{Q} \in \Delta(S)$ with probability at least $1 - \delta_0$ over a sample $S' \sim \mathcal{Q}^{m_0}$ the hypothesis $\mathbf{h} = \mathcal{W}(S')$ satisfies $\operatorname{corr}_{\mathcal{Q}}(\mathbf{h}) \geq \gamma'$, then, for $T \geq \lceil 32 \ln(em)/\gamma'^2 \rceil$, running Algorithm 1 on input $(S, \mathcal{W}, m_0, T, \delta, \delta_0)$ yields a voting classifier $\mathbf{v} \in \operatorname{conv}(\mathcal{H})$ such that with probability at least $1 - \delta$ over the random draws from Line 5 it holds that $y\mathbf{v}(x) > \gamma'/8$ for all $(x, y) \in S$.*

The proof of Lemma 3.1 is based on the standard analysis of ADABOOST (e.g., Schapire and Freund [2012]) and is deferred to the appendix.

As we discussed, for re-labelings of the training sequence according to a function $f$ in the reference class $\mathcal{F}$, a $(\gamma, \varepsilon_0)$ agnostic weak learner with $\mathcal{F}$ as reference class behaves like the weak learner in Lemma 3.1 with $\gamma' = \gamma - \varepsilon_0 =: \theta$. We are interested in the re-labeling $S_{f^\star}$, where $f^\star$ is such that $\operatorname{corr}_{\mathcal{D}}(f^\star) = \sup_{f \in \mathcal{F}} \operatorname{corr}_{\mathcal{D}}(f)$, with $\mathcal{D}$ being the true data distribution. So, we will regard $f^\star$ as our target function since approximating it concludes the proof of Theorem 1.3. As we lack direct access to $f^\star$, the first step of our proposed method (Algorithm 2) is to run Algorithm 1 on all possible re-labelings of $S_1$ and accumulate the obtained hypotheses in a bag, $\mathcal{B}_1$ (cf. **for** loop starting at Line 3). Let $\mathbf{v}_g$ be the hypothesis obtained when running Algorithm 1 on the re-labeling $S_{f^\star}$. The subsequent steps of Algorithm 2 are designed to find $\mathbf{v}_g$ within $\mathcal{B}_1$.

---

**Input** : Training sequence $S \in (\mathcal{X} \times \{\pm 1\})^m$ (with $m$ multiple of 3)[4], weak learner $\mathcal{W}$, confidence parameters $\delta_0, \delta \in (0, 1)$

1   $\mathcal{B}_1 \leftarrow \emptyset, \mathcal{B}_2 \leftarrow \emptyset$

2   Let $S_1, S_2$, and $S_3$ be the first, second, and third thirds of $\mathbf{S}$, respectively

    // Reduce to realizable case

3   **foreach** $\mathcal{Y} \in \{\pm 1\}^{m/3}$ **do**

4     $S' \leftarrow \big((S_1|_{\mathcal{X}})_i, \mathcal{Y}_i\big)_{i=1}^{m/3}$                // Re-label $S_1$ with $\mathcal{Y}$

5     $T \leftarrow \lceil 32m \ln(em) \rceil$                         // Number of rounds

6     $\mathbf{v} \leftarrow \text{ALGORITHM}1(S', \mathcal{W}, m_0, T, \delta/10, \delta_0)$

7     $\mathcal{B}_1 \leftarrow \mathcal{B}_1 \cup \{\mathbf{v}\}$

    // Filter out good hypotheses without knowledge of $\gamma$

8   **foreach** $\gamma' \in \{1, 1/2, 1/4, \ldots, 1/2^{\lceil \log_2 \sqrt{m} \rceil}\}$ **do**

9     $\mathbf{v}_{\gamma'}^* \leftarrow \arg\min_{\mathbf{v} \in \mathcal{B}_1} \{\mathcal{L}_{S_2}^{\gamma'}(\mathbf{v})\}$         // Minimizer of the $\gamma'$-margin loss

10    $\mathcal{B}_2 \leftarrow \mathcal{B}_2 \cup \{\mathbf{v}_{\gamma'}^*\}$

    // Return hypothesis with the lowest validation error

11   **return** $\mathbf{v} = \arg\min_{\mathbf{v} \in \mathcal{B}_2} \{\mathcal{L}_{S_3}(\mathbf{v})\}$

**Algorithm 2:** Agnostic boosting algorithm

---

Leveraging Lemma 3.1, we have that with probability at least $1 - \delta/10$ over the randomness used in Algorithm 1, there exists $\mathbf{v}_g \in \mathcal{B}_1$ such that

$$\mathcal{L}_{S_{1,f^\star}}^{\theta/8}(\mathbf{v}_g) = 0. \tag{3}$$

That is, $\mathbf{v}_g$ has zero *empirical* $\theta/8$-margin loss on $S_{1,f^\star}$, which is the re-labeling of $S_1$ according to $f^\star$. In the following lemma, we convert this into a bound on the *population* loss of $\mathbf{v}_g$.

**Lemma 3.2.** *There exist universal constants $C' \geq 1$ and $\hat{c} > 0$ for which the following holds. For all margin levels $0 \leq \gamma < \gamma' \leq 1$, hypothesis class $\mathcal{H} \subseteq [-1, 1]^{\mathcal{X}}$, and distribution $\mathcal{D} \in \Delta(\mathcal{X} \times \{\pm 1\})$, it holds with probability at least $1 - \delta$ over $\mathbf{S} \sim \mathcal{D}^m$ that for all $v \in \operatorname{conv}(\mathcal{H})$*

$$\mathcal{L}_{\mathcal{D}}^{\gamma}(v) \leq \mathcal{L}_{\mathbf{S}}^{\gamma'}(v) + C'\left(\sqrt{\mathcal{L}_{\mathbf{S}}^{\gamma'}(v) \cdot \frac{\beta}{m}} + \frac{\beta}{m}\right),$$

*where $\beta = \frac{d}{(\gamma' - \gamma)^2} \operatorname{Ln}^{3/2}\left(\frac{(\gamma' - \gamma)^2 m}{d} \frac{\gamma'}{\gamma' - \gamma}\right) + \ln \frac{1}{\delta}$ with $d = \operatorname{fat}_{\hat{c}(\gamma' - \gamma)}(\mathcal{H})$.*

The proof of Lemma 3.2 is based on techniques similar to those used in Høgsgaard and Larsen [2025], and it is deferred to Appendix C. The argument requires controlling the complexity of $\operatorname{conv}(\mathcal{H})$. Specifically, we show for $\alpha > 0$ that $\operatorname{fat}_{\alpha}(\operatorname{conv}(\mathcal{H})) = \mathcal{O}(\operatorname{fat}_{c\alpha}(\mathcal{H})/\alpha^2)$ for some universal constant $c > 0$. Our strategy to achieve that builds on Larsen and Ritzert [2022, Lemma 9]. We believe this result could be of independent interest.

Applying Lemma 3.2 with $\gamma = \theta/16$, $\gamma' = \theta/8$ and any distribution $\mathcal{Q}$, we have that with probability at least $1 - \delta/10$ over $\mathbf{S}' \sim \mathcal{Q}^m$ it holds simultaneously for all $v \in \operatorname{conv}(\mathcal{H})$ that

$$\mathcal{L}_{\mathcal{Q}}^{\theta/16}(v) = \mathcal{L}_{\mathbf{S}'}^{\theta/8}(v) + \tilde{\mathcal{O}}\left(\sqrt{\mathcal{L}_{\mathbf{S}'}^{\theta/8}(v) \cdot \frac{d}{m\theta^2}} + \frac{d}{m\theta^2}\right),$$

---

[4]We assume that $m$ is a multiple of 3 merely for simplicity.

with $d = \text{fat}_{\hat{c}\theta/16}(\mathcal{H})$. Now consider the distribution $\mathcal{D}_{f^\star}$ associated with $\boldsymbol{S}_{f^\star}$, i.e., obtained by re-labeling samples from $\mathcal{D}$ according to $f^\star$. Setting $\mathcal{Q} = \mathcal{D}_{f^\star}$, $\boldsymbol{S}' = \boldsymbol{S}_{1,f^\star}$, and using a union bound to have the event associated with Eq. (3) also hold, we obtain that with probability at least $1 - 2\delta/10$ over $\boldsymbol{S}_1$ and the randomness used in Algorithm 1, there exists $\mathbf{v}_g \in \boldsymbol{\mathcal{B}}_1$ such that

$$\mathcal{L}_{\mathcal{D}_{f^\star}}^{\theta/16}(\mathbf{v}_g) = \tilde{\mathcal{O}}\Big(\frac{d}{m\theta^2}\Big). \tag{4}$$

Still, Eq. (4) bounds the loss of $\mathbf{v}_g$ relative to distribution $\mathcal{D}_{f^\star}$ while we are interested in the loss on the data distribution $\mathcal{D}$. We convert between these by noticing that given $(x, y) \in \mathcal{X} \times \{\pm 1\}$, if $y \cdot \mathbf{v}_g(x) \leq \theta/16$, then either $f^\star(x) \neq y$, so that $f^\star(x) \cdot y \leq \theta/16$; or $f^\star(x) = y$, so that $f^\star(x) \cdot \mathbf{v}_g(x) \leq \theta/16$. Thus,

$$\mathcal{L}_{\mathcal{D}}^{\theta/16}(\mathbf{v}_g) \leq \text{err}_{\mathcal{D}}(f^\star) + \mathcal{L}_{\mathcal{D}_f^\star}^{\theta/16}(\mathbf{v}_g) = \text{err}_{\mathcal{D}}(f^\star) + \tilde{\mathcal{O}}\Big(\frac{d}{m\theta^2}\Big), \tag{5}$$

where the last equality follows from Eq. (4) which holds with probability at least $1 - 2\delta/10$.

## 3.2 Filtering the hypotheses

For this step, we use the second *independent* portion of the training data, $\boldsymbol{S}_2 \sim \mathcal{D}^{m/3}$. We saw that with high probability $\boldsymbol{\mathcal{B}}_1$ contains a good hypothesis $\mathbf{v}_g$, so our goal is to distinguish $\mathbf{v}_g$—or some hypothesis at least as good—from the other hypotheses in $\boldsymbol{\mathcal{B}}_1$. A naïve way to do so would be to use $\boldsymbol{S}_2$ as a validation set, leveraging its independency. To succeed, we would need to ensure that all hypotheses in $\boldsymbol{\mathcal{B}}_1$ have generalization error close to their error on $\boldsymbol{S}_2$. Alas, since $|\boldsymbol{\mathcal{B}}_1| = 2^{m/3}$, a union bound over the entire bag would lead to an extra term of order $\Theta(\ln(2^m/\delta)/m) = \Theta(1)$ in the final bound, making it vacuous. Thus, to proceed we must first reduce the size of $\boldsymbol{\mathcal{B}}_1$ while keeping at least one good hypothesis.

Given Eq. (5), we know that $\boldsymbol{\mathcal{B}}_1$ contains, with high probability, a hypothesis with relatively low generalization loss relative to margin $\theta/16$. However, crucially, we do not assume knowledge of $\gamma$ or $\varepsilon_0$ and, thus, of $\theta = \gamma - \varepsilon_0$. We overcome this by considering a range of possible values for the margin and storing the minimizer of the empirical loss relative to each value. More precisely, we consider each $\gamma' \in \{1, 1/2, 1/4, \ldots, 1/\sqrt{m}\}$ where we can dismiss smaller margin values since the upper bound in Theorem 1.3 becomes vacuous—greater than 1—for $\gamma' - \varepsilon_0 < 1/\sqrt{m}$. Then, in the **for** loop starting at Line 8 of Algorithm 2, we let

$$\boldsymbol{\mathcal{B}}_2 = \Big\{\arg\min_{\mathbf{v} \in \boldsymbol{\mathcal{B}}_1} \mathcal{L}_{\boldsymbol{S}_2}^{\gamma'}(\mathbf{v}) : \gamma' \in \{1, 1/2, 1/4, \ldots, 1/2^{\lceil \log_2 \sqrt{m}\rceil}\}\Big\},$$

so that $|\boldsymbol{\mathcal{B}}_2| = \mathcal{O}(\ln m)$ and for some $\gamma'_g \in (\theta/32, \theta/16]$ we have $\mathbf{v}'_g := \arg\min_{\mathbf{v} \in \boldsymbol{\mathcal{B}}_1} \mathcal{L}_{\boldsymbol{S}_2}^{\gamma'_g}(\mathbf{v}) \in \boldsymbol{\mathcal{B}}_2$.

From here, we follow a chain of inequalities between different losses. We start by applying Lemma 3.2 once more, this time with sample $\boldsymbol{S}_2$ and margin levels $\gamma = 0$ and $\gamma' = \gamma'_g$. We obtain that, with probability at least $1 - \delta/10$ over $\boldsymbol{S}_2$ it holds for $\mathbf{v}'_g$, in particular, that

$$\mathcal{L}_{\mathcal{D}}(\mathbf{v}'_g) = \mathcal{L}_{\mathcal{D}}^0(\mathbf{v}'_g) = \mathcal{L}_{\boldsymbol{S}_2}^{\gamma'}(\mathbf{v}'_g) + \tilde{\mathcal{O}}\Big(\sqrt{\mathcal{L}_{\boldsymbol{S}_2}^{\gamma'}(\mathbf{v}'_g) \cdot \frac{d'}{(\gamma')^2 m}} + \frac{d'}{(\gamma')^2 m}\Big), \tag{6}$$

where $d' = \text{fat}_{\hat{c}\gamma'_g}(\mathcal{H})$. Recalling that $\mathbf{v}'_g$ is the minimizer of $\mathcal{L}_{\boldsymbol{S}_2}^{\gamma'_g}$ over $\boldsymbol{\mathcal{B}}_1$ and that $\mathbf{v}_g$ belongs to $\boldsymbol{\mathcal{B}}_1$, we must, then, have that $\mathcal{L}_{\boldsymbol{S}_2}^{\gamma'_g}(\mathbf{v}'_g) \leq \mathcal{L}_{\boldsymbol{S}_2}^{\gamma'_g}(\mathbf{v}_g)$. Now we leverage that $\gamma' \in (\theta/32, \theta/16]$: on the one hand, $\gamma'_g > \theta/32$ so, as the fat-shattering dimension is decreasing in its level parameter, $d' = \text{fat}_{\hat{c}\gamma'_g}(\mathcal{H}) \leq \text{fat}_{\hat{c}\theta/32}(\mathcal{H}) =: \hat{d}$; on the other hand, $\gamma'_g \leq \theta/16$, thus $\mathcal{L}_{\boldsymbol{S}_2}^{\gamma'_g}(\mathbf{v}_g) \leq \mathcal{L}_{\boldsymbol{S}_2}^{\theta/16}(\mathbf{v}_g)$. Finally, we use a standard concentration bound (Lemma D.1) to ensure that, with probability at least $1 - \delta/10$ over $\boldsymbol{S}_2$, it holds that $\mathcal{L}_{\boldsymbol{S}_2}^{\theta/16}(\mathbf{v}_g) = \mathcal{L}_{\mathcal{D}}^{\theta/16}(\mathbf{v}_g) + \tilde{\mathcal{O}}(\sqrt{\mathcal{L}_{\mathcal{D}}^{\theta/16}(\mathbf{v}_g) \cdot \ln(1/\delta)/m} + \ln(1/\delta)/m)$, and, thus, by the chain of inequalities above, that

$$\mathcal{L}_{\boldsymbol{S}_2}^{\theta/16}(\mathbf{v}'_g) = \mathcal{L}_{\mathcal{D}}^{\theta/16}(\mathbf{v}_g) + \tilde{\mathcal{O}}\Big(\sqrt{\frac{\mathcal{L}_{\mathcal{D}}^{\theta/16}(\mathbf{v}_g)}{m} \cdot \ln\frac{1}{\delta}} + \frac{1}{m}\ln\frac{1}{\delta}\Big). \tag{7}$$

Combining Equations 5, 6 and 7, and taking a union bound over the respective events required to hold simultaneously, the appropriate calculations yield the following lemma, which summarizes the progress made so far.

**Lemma 3.3.** *There exist universal constants $C \geq 1$ and $\hat{c} > 0$ for which the following holds. After the **for** loop starting at Line 8 of Algorithm 2 it holds that $|\mathcal{B}_2| \leq \log_2 m$ and with probability at least $1 - \delta/2$ over $\boldsymbol{S}_1$, $\boldsymbol{S}_2$ and randomness used in Algorithm 1 that $\mathcal{B}_2$ contains a voting classifier $v$ such that*

$$\mathcal{L}_{\mathcal{D}}(v) \leq \operatorname{err}_{\mathcal{D}}(f^\star) + \sqrt{C \operatorname{err}_{\mathcal{D}}(f^\star) \cdot \beta} + C \cdot \beta,$$

*where $\beta = \frac{\hat{d}}{\theta^2 m} \cdot \operatorname{Ln}^{3/2}\left(\frac{\theta^2 m}{\hat{d}}\right) + \ln\frac{10}{\delta}$ with $\hat{d} = \operatorname{fat}_{\hat{c}\theta/32}(\mathcal{H})$.*

### 3.3 Extracting the final classifier

For the final step, we use the third and last independent portion of the training data, $\boldsymbol{S}_3 \sim \mathcal{D}^{m/3}$, as a validation set. We saw previously that naïvely using a validation set to extract a good hypothesis from $\mathcal{B}_1$ would not work due to the exponential size of $\mathcal{B}_1$. In the previous step, we overcame this limitation by reducing the number of hypotheses to $\mathcal{O}(\ln m)$ while, with high probability, preserving a hypothesis with good generalization properties. Therefore, we can now use $\boldsymbol{S}_3$ to select the best hypothesis from $\mathcal{B}_2$.

Given $v \in \mathcal{B}_2$, we can use standard concentration results (Lemma D.1) to bound the probability that the empirical loss of $v$ on $\boldsymbol{S}_3$ is close to its true loss $\mathcal{L}_{\mathcal{D}}(v)$. By the union bound, with probability at least $1 - \delta/2$ over $\boldsymbol{S}_3$,

$$\mathcal{L}_{\mathcal{D}}(v) = \mathcal{L}_{\boldsymbol{S}_3}(v) + \tilde{\mathcal{O}}\left(\sqrt{\frac{\mathcal{L}_{\boldsymbol{S}_3}(v)}{m} \cdot \ln\frac{|\mathcal{B}_2|}{\delta}} + \frac{1}{m} \cdot \ln\frac{|\mathcal{B}_2|}{\delta}\right)$$

and

$$\mathcal{L}_{\boldsymbol{S}_3}(v) = \mathcal{L}_{\mathcal{D}}(v) + \tilde{\mathcal{O}}\left(\sqrt{\frac{\mathcal{L}_{\mathcal{D}}(v)}{m} \cdot \ln\frac{|\mathcal{B}_2|}{\delta}} + \frac{1}{m} \cdot \ln\frac{|\mathcal{B}_2|}{\delta}\right)$$

for all $v \in \mathcal{B}_2$, simultaneously. Finally, under the event associated with Lemma 3.3, $\mathcal{B}_2$ contains a hypothesis $\mathbf{v}'_g$ with low generalization error. Leveraging this and the equalities above, we can bound the generalization error of $\arg\min_{v \in \mathcal{B}_2} \mathcal{L}_{\boldsymbol{S}_3}(v)$, which is the final classifier output by Algorithm 2.

The complete argument proves our main result, stated here in terms of errors rather than correlations.

**Theorem 3.4.** *There exist universal constants $C, c > 0$ such that the following holds. Let $\mathcal{W}$ be a $(\gamma, \varepsilon_0, \delta_0, m_0, \mathcal{F}, \mathcal{H})$ agnostic weak learner. If $\gamma > \varepsilon_0$ and $\delta_0 < 1$, then, for all $\delta \in (0, 1)$, $m \in \mathbb{N}$, and $\mathcal{D} \in \Delta(\mathcal{X} \times \{\pm 1\})$, given training sequence $\boldsymbol{S} \sim \mathcal{D}^m$, we have that Algorithm 2 on inputs $(\boldsymbol{S}, \mathcal{W}, \delta, \delta_0, m_0)$ returns, with probability at least $1 - \delta$ over $\boldsymbol{S}$ and the internal randomness of the algorithm, the output $\mathbf{v}$ of Algorithm 2 satisfies that*

$$\mathcal{L}_{\mathcal{D}}(\mathbf{v}) \leq \operatorname{err}_{\mathcal{D}}(f^\star) + \sqrt{C \operatorname{err}_{\mathcal{D}}(f^\star) \cdot \beta} + C \cdot \beta,$$

*where $\beta = \frac{\hat{d}}{(\gamma - \varepsilon_0)^2 m} \cdot \operatorname{Ln}^{3/2}\left(\frac{(\gamma - \varepsilon_0)^2 m}{\hat{d}}\right) + \frac{1}{m} \ln\frac{\ln m}{\delta}$ with $\hat{d} = \operatorname{fat}_{c(\gamma - \varepsilon_0)/32}(\mathcal{H})$.*

To recover Theorem 1.3, the bound in terms of correlation, we use that $\mathcal{L}_{\mathcal{D}}(\mathbf{v}) \geq \operatorname{err}_{\mathcal{D}}(\operatorname{sign}(v))$, since $\operatorname{sign}(0) = 1$, and for the binary hypotheses, $f^\star$ and $\operatorname{sign}(v)$ we have that $\operatorname{err}_{\mathcal{D}}(\operatorname{sign}(v)) = (1 - \operatorname{corr}(\operatorname{sign}(v)))/2$ and $\operatorname{err}_{\mathcal{D}}(f^\star) = (1 - \operatorname{corr}(f^\star))/2$.

## 4 Conclusion and Future Work

In this work, we provided a novel algorithm for agnostic weak-to-strong learning and proved that it achieves nearly optimal sample complexity under fairly general assumptions. Notoriously, while previous works set varying conditions on the relationships between parameters, Theorem 1.3 recovers the iconic generality of classic boosting by allowing for any non-trivial setting. Furthermore, our algorithm bound incorporates the error of the best hypotheses in the reference class, interpolating between the agnostic boosting setting and the realizable boosting setting, which, to the best of

our knowledge, is not the case for any previous bounds. This also implies an even better sample complexity when $\mathrm{corr}(f^\star)$ is large.

As for future work directions, providing efficient algorithms with sample complexity close to the error rates of Theorem 1.2 is the most natural next step. Besides that, we conjecture that the logarithmic factors in our bounds could be removed, as in the realizable case (e.g., Larsen and Ritzert [2022]). Finally, given the line of works stemming from Kalai and Kanade [2009], it is also pressing to further improve the sample complexity of agnostic boosting algorithms based on sample re-labelings.

## Acknowledgments and Disclosure of Funding

While this work was carried out, Arthur da Cunha, Mikael Møller Høgsgaard, and Yuxin Sun were supported by the European Union (ERC, TUCLA, 101125203). Views and opinions expressed are however those of the author(s) only and do not necessarily reflect those of the European Union or the European Research Council. Neither the European Union nor the granting authority can be held responsible for them. Arthur da Cunha and Mikael Møller Høgsgaard were also supported by Independent Research Fund Denmark (DFF) Sapere Aude Research Leader grant No. 9064-00068B.

Andrea Paudice is supported by Novo Nordisk Fonden Start Package Grant No. NNF24OC0094365 (Actionable Performance Guarantees in Machine Learning).

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

## A   Notes on different concepts of agnostic weak learning

### A.1   Distribution-Free Boosting

The notion of a weak agnostic learner was introduced by Ben-David et al. [2001]. The following is a small extension of their definition, where we added the possibility of failure to the weak learner and made explicit the base hypothesis class $\mathcal{H}$.

**Definition A.1** (Agnostic Weak Learner of Ben-David et al. [2001]). Given $\varepsilon_0, \delta_0 \in [0,1]$, a learning algorithm $\mathcal{W}: (\mathcal{X} \times \{\pm 1\})^* \to \{\pm 1\}^{\mathcal{X}}$ is a $(\varepsilon_0, \delta_0)$–*agnostic weak learner* with sample complexity $m_0 \in \mathbb{N}$ with respect to reference hypothesis class $\mathcal{F} \subseteq \{\pm 1\}^{\mathcal{X}}$ and base hypothesis class $\mathcal{H} \subseteq \{\pm 1\}^{\mathcal{X}}$ iff: For all $\mathcal{D} \in \Delta(\mathcal{X} \times \{\pm 1\})$, given sample $\boldsymbol{S} \sim \mathcal{D}^{m_0}$, with probability at least $1 - \delta_0$ over the randomness of $\boldsymbol{S}$, the hypothesis $\mathbf{v} = \mathcal{W}(\boldsymbol{S}) \in \mathcal{H}$ satisfies that

$$\mathrm{err}_{\mathcal{D}}(\mathbf{v}) \leq \inf_{f \in \mathcal{F}} \mathrm{err}_{\mathcal{D}}(f) + \varepsilon_0,$$

or, equivalently,

$$\begin{aligned}
\mathrm{corr}_{\mathcal{D}}(\mathbf{v}) &= 1 - 2\,\mathrm{err}_{\mathcal{D}}(\mathbf{v}) \\
&\geq 1 - 2 \inf_{f \in \mathcal{F}} \mathrm{err}_{\mathcal{D}}(f) - 2\varepsilon_0 \\
&= 1 + 2 \sup_{f \in \mathcal{F}} (-\mathrm{err}_{\mathcal{D}}(f)) - 2\varepsilon_0 \\
&= \sup_{f \in \mathcal{F}} \mathrm{corr}_{\mathcal{D}}(f) - 2\varepsilon_0.
\end{aligned}$$

This definition is quite close to that of Ghai and Singh [2024], and to ours. This becomes evident if one employs correlations instead of $\mathrm{err}$, weakens the required correlation between $\mathbf{v}$ and $\mathcal{F}$, and lets the weak learner output hypotheses with outputs in $[-1, 1]$.[5]

**Definition A.2** (Agnostic Weak Learner used in this article). Let $\gamma, \varepsilon_0, \delta_0 \in [0,1]$, $m_0 \in \mathbb{N}$, $\mathcal{F} \subseteq \{\pm 1\}^{\mathcal{X}}$, and $\mathcal{H} \subseteq [-1, 1]^{\mathcal{X}}$. A learning algorithm $\mathcal{W}: (\mathcal{X} \times \{\pm 1\})^* \to \mathcal{H}$ is an agnostic weak learner with advantage parameters $\gamma$ and $\varepsilon_0$, failure probability $\delta_0$, sample complexity $m_0$, reference hypothesis class $\mathcal{F}$, and base hypothesis class $\mathcal{H}$ iff: For any distribution $\mathcal{D}$ over $\mathcal{X} \times \{\pm 1\}$, given sample $\boldsymbol{S} \sim \mathcal{D}^{m_0}$, with probability at least $1 - \delta_0$ over the randomness of $\boldsymbol{S}$, the hypothesis $\mathbf{w} = \mathcal{W}(\boldsymbol{S})$ satisfies that

$$\mathrm{corr}_{\mathcal{D}}(\mathbf{w}) \geq \gamma \cdot \sup_{f \in \mathcal{F}} \mathrm{corr}_{\mathcal{D}}(f) - \varepsilon_0.$$

---

[5]It is not fully clear whether weak learners in Ghai and Singh [2024] are binary or real-valued.

If $\mathcal{H}$ is binary-valued $\mathcal{H} \subset \{\pm 1\}^{\mathcal{X}}$, the above is equivalent to

$$\begin{aligned}
\mathrm{err}_{\mathcal{D}}(\mathbf{v}) &= \frac{1}{2}(1 - \mathrm{corr}_{\mathcal{D}}(\mathbf{v})) \\
&\leq \frac{1}{2}(1 - \gamma \sup_{f \in \mathcal{F}} \mathrm{corr}_{\mathcal{D}}(f) + \varepsilon_0) \\
&= \frac{1}{2} - \frac{\gamma}{2} \sup_{f \in \mathcal{F}} \mathrm{corr}_{\mathcal{D}}(f) + \frac{\varepsilon_0}{2} \\
&= \frac{1}{2} + \gamma \inf_{f \in \mathcal{F}} \mathrm{err}_{\mathcal{D}}(f) - \frac{\gamma}{2} + \frac{\varepsilon_0}{2}.
\end{aligned}$$

This observation shows that the definition captures that of Ben-David et al. [2001] when $\gamma = 1$ and the base hypothesis class is binary.

## A.2 Distribution-Dependent Boosting

### A.2.1 Kalai and Kanade [2009]

Kalai and Kanade [2009] use the following definition of a weak learner, where we have made the sample complexity a specific parameter. This is an alternative to directly writing "over its random input" as in Kalai and Kanade [2009, Definition 1].

**Definition A.3** (Agnostic Weak Learner [Kalai and Kanade, 2009]). Given $\gamma, \varepsilon_0, \delta_0 \in (0, 1)$, and $\mathcal{D} \in \Delta(\mathcal{X})$, a learning algorithm $\mathcal{W} \colon (\mathcal{X} \times [-1, 1])^* \to [-1, 1]^{\mathcal{X}}$ is a $(\gamma, \varepsilon_0, \delta_0, \mathcal{D})$–*agnostic weak learner* with sample complexity $m_0 \in \mathbb{N}$ with respect to reference hypothesis class $\mathcal{F}$ and distribution $\mathcal{D}$ iff: For any re-labeling function $g \in [-1, 1]^{\mathcal{X}}$, given sample $\mathbf{S} \sim \mathcal{D}^m$ with $m \geq m_0$, with probability at least $1 - \delta_0$ over the randomness of $\mathbf{S}$ the hypothesis $\mathbf{v} = \mathcal{W}(\mathbf{S}) \in \mathcal{H}$ satisfies that

$$\mathbb{E}_{\mathbf{x} \sim \mathcal{D}}[g(\mathbf{x})\mathbf{v}(\mathbf{x})] \geq \gamma \cdot \sup_{f \in \mathcal{F}} \mathbb{E}_{\mathbf{x} \sim \mathcal{D}}[g(\mathbf{x})f(\mathbf{x})] - \varepsilon_0.$$

The authors mention that one can think of $m_0$ as being of the order of $1/\varepsilon_0^2$.

The above is a distribution-specific notion of weak learning in that the marginal over $\mathcal{X}$ is fixed, while the conditional distribution over the labels can vary. Theorem 1 of Kalai and Kanade [2009] states that there exists an algorithm $\mathcal{A}$ which, given access to a $(\gamma, \varepsilon_0, \delta, \mathcal{D})$–agnostic weak learner $\mathcal{W}$, produces, with probability at least $1 - 4\delta T$, a hypothesis $v \in \{\pm 1\}^{\mathcal{X}}$ such that $\mathbb{E}_{(\mathbf{x}, \mathbf{y}) \sim \mathcal{D}}[v(\mathbf{x})\mathbf{y}] \geq \sup_{f \in \mathcal{F}} \mathbb{E}_{(\mathbf{x}, \mathbf{y}) \sim \mathcal{D}}[f(\mathbf{x})y] - \varepsilon_0/\gamma - \varepsilon$. Doing so requires $T = 29/(\gamma^2 \varepsilon^2)$ calls to the weak learner, each requiring $m_0$ samples. Hence, the total sample complexity is $\mathcal{O}(m_0/(\gamma^2 \varepsilon^2))$, which for $m_0 = 1/\varepsilon_0^2$ is of order $\mathcal{O}(1/(\gamma^2 \varepsilon^2 \varepsilon_0^2))$.

### A.2.2 Feldman [2010]

Let $\mathcal{D}' \in \Delta(\mathcal{X})$, and $\phi \in [-1, 1]^{\mathcal{X}}$. Feldman [2010] defines a distribution $A = (\mathcal{D}', \phi)$ over examples $\mathcal{X} \times \{\pm 1\}$, in the following way: First, a point $x \in \mathcal{X}$ is drawn according to $\mathcal{D}'$, then, $x$ is labeled 1 with probability $(\phi(x) + 1)/2$, and labeled $-1$ otherwise. With this notation, Feldman [2010] employs the following definition: For $0 < \gamma \leq \alpha \leq 1/2$ and a distribution $A$, an algorithm $\mathcal{W}$ is an $(\alpha, \gamma)$–weak agnostic learner iff it produces a hypothesis $h \in \mathcal{H} \subseteq \{\pm 1\}^{\mathcal{X}}$ such that $\mathbb{P}_{(\mathbf{x}, \mathbf{y}) \sim A}[h(\mathbf{x}) \neq \mathbf{y}] \leq 1/2 - \gamma$ whenever $\inf_{f \in \mathcal{F}} \mathbb{P}_{(\mathbf{x}, \mathbf{y}) \sim A}[f(\mathbf{x}) \neq \mathbf{y}] \leq 1/2 - \alpha$, where $\mathcal{F}$ is a reference hypothesis class. The algorithm proposed by Feldman [2010] works by re-labeling, so it only requires a distribution-specific weak learning notion.

The weak learning notion used by Kalai and Kanade [2009] implies the definition of Feldman [2010] if, in the definition of Kalai and Kanade [2009], we have $\mathcal{H}$ and $\mathcal{F}$ consisting of binary-valued hypotheses, $\gamma > \varepsilon_0$ (the non-trivial case), and $\delta_0 = 0$. Specifically, for any $\alpha \in [0, 1/2]$, any $(\gamma, \varepsilon_0, 0, \mathcal{D})$–agnostic weak learner in the sense of Kalai and Kanade [2009] is a $(\alpha, \gamma\alpha - \varepsilon_0/2)$–agnostic learner in the sense of Feldman [2010]. Notice that such a learner is only better than random guessing when $\gamma\alpha - \varepsilon_0/2 > 0$, that is, when $\alpha > \varepsilon_0/(2\gamma)$. The reduction between definitions come from noticing that, for distribution $A = (\mathcal{D}, g)$ defined as in Feldman [2010] but using function $g$

from the definition of Kalai and Kanade [2009], and for any $h \in \{\pm 1\}^{\mathcal{X}}$, we have that

$$\mathbb{E}_{(\mathbf{x},\mathbf{y})\sim(\mathcal{D},g)}[h(\mathbf{x})\mathbf{y}] = \mathbb{E}_{\mathbf{x}\sim\mathcal{D}}\left[\left(\frac{g(\mathbf{x})+1}{2}\right)h(\mathbf{x}) - \left(1 - \frac{g(\mathbf{x})+1}{2}\right)h(\mathbf{x})\right]$$
$$= \mathbb{E}_{\mathbf{x}\sim\mathcal{D}}[g(\mathbf{x})h(\mathbf{x})].$$

Hence, as $\mathbb{E}_{(\mathbf{x},\mathbf{y})\sim(\mathcal{D},g)}[h(\mathbf{x})\mathbf{y}] = \mathrm{corr}_A(h) = 1 - 2\,\mathrm{err}_A(h)$ for binary-valued $h$, the condition of Kalai and Kanade [2009] implies that

$$\mathbb{E}_{\mathbf{x}\sim\mathcal{D}}[g(\mathbf{x})\mathbf{v}(\mathbf{x})] \geq \gamma \cdot \sup_{f\in\mathcal{F}} \mathbb{E}_{\mathbf{x}\sim\mathcal{D}}[g(\mathbf{x})f(\mathbf{x})] - \varepsilon_0,$$

so

$$1 - 2\,\mathrm{err}_A(\mathbf{v}) \geq \gamma \cdot (1 - 2\inf_{f\in\mathcal{F}}\mathrm{err}_A(f)) - \varepsilon_0,$$

thus

$$\mathrm{err}_A(\mathbf{v}) \leq 1/2 - \gamma/2 + \gamma\inf_{f\in\mathcal{F}}\mathrm{err}_A(f) + \varepsilon_0/2.$$

Hence, if $\inf_{f\in\mathcal{F}}\mathrm{err}_A(f) \leq 1/2 - \alpha$, then the weak learner returns a hypothesis $\mathbf{v}$ such that $\mathrm{err}_A(\mathbf{v}) \leq 1/2 + \varepsilon_0/2 - \gamma\alpha$, as claimed.

For any $0 < \varepsilon < 1$, algorithm $\mathcal{A}$ from Feldman [2010, Theorem 3.1], given access to a $(\alpha,\gamma)$–weak learner $\mathcal{W}$, produces a classifier $v \in \{\pm 1\}^{\mathcal{X}}$ (using at most $1/\gamma^2$ queries to $\mathcal{W}$) such that

$$\mathrm{err}_{\mathcal{D}}(v) = \inf_{f\in\mathcal{F}}\mathrm{err}_{\mathcal{D}}(f) + 2\alpha + \varepsilon.$$

That is, to obtain a strong learner (i.e., error at most $2\varepsilon$, and then re-scaling $\varepsilon$ to $\varepsilon/2$), one must have $\alpha = \varepsilon$.

As mentioned, the execution of $\mathcal{A}$ queries $\mathcal{W}$ at most $1/\gamma^2$ times to get an output hypothesis $h$ [Feldman, 2010, proof of Theorem 3.1]. After each such query, $\mathcal{A}$ checks whether $\mathbb{P}_{(\mathbf{x},\mathbf{y})\sim A_t}[h(\mathbf{x}) \neq \mathbf{y}] \leq 1/2 - \gamma$, where $A_t$ is the distribution at step $t$. To the best of our knowledge, this check requires $\Omega(1/\gamma^2)$ examples from $A_t$ since the threshold could be close to $1/2$. Furthermore, $\mathcal{A}$ also preforms at most $1/\varepsilon^2$ times a step called "balancing". Each balancing requires checking whether $\mathbb{P}_{(\mathbf{x},\mathbf{y})\sim A_t}[g(\mathbf{x}) \neq \mathbf{y}] \leq 1/2 - \varepsilon/2$ for a hypothesis $g$, so, by the same reasoning, it should require $\Omega(1/\varepsilon^2)$ samples. Thus, in total the algorithm needs $\mathcal{O}(1/\gamma^4)$ samples for checking the error of the weak learners output, and $\mathcal{O}(1/\varepsilon^4)$ samples for checking the error of the hypothesis returned by the balancing step, yielding a sample complexity of $\mathcal{O}(1/\varepsilon^4 + 1/\gamma^4)$. Here, we do not take into account the samples needed for the weak learner calls, which is $m_0/\gamma^2$ if the sample complexity of the weak learner is $m_0$.

Furthermore, if the algorithm by Feldman [2010] is given a $(\gamma, \varepsilon_0, 0, \mathcal{D})$–agnostic weak learner in the sense of Kalai and Kanade [2009], then, to get a strong learner with error at most $2\varepsilon$ one has to set $\alpha = \varepsilon$. So, the weak learner only gives a non-trivial guarantee when $\varepsilon > \varepsilon_0/(2\gamma)$, implying that one has to set $\varepsilon_0 = O(\varepsilon\gamma)$. For $\varepsilon_0 = \Theta(\varepsilon\gamma)$, this gives a $(\varepsilon, \gamma' = \Theta(\varepsilon\gamma))$–agnostic weak learner in the setting of Feldman [2010], which in turn implies that the sample complexity of the weak learner is $m_0 = 1/\varepsilon_0^2 = 1/(4\varepsilon^2\gamma^2)$ and that the total sample complexity becomes $\mathcal{O}(1/\varepsilon^4 + 1/(\gamma')^4 + m_0/\gamma'^2) = \mathcal{O}(1/(\gamma^4\varepsilon^4))$.

### A.2.3 Brukhim et al. [2020]

Brukhim et al. [2020] uses the following empirical notion of a agnostic weak learning algorithm.

**Definition A.4** (Agnostic Weak Learner of Brukhim et al. [2020, Defintion 6])**.** Let $\mathcal{F} \subseteq \{\pm 1\}^{\mathcal{X}}$ and let $X = (x_1, \ldots, x_m) \in \mathcal{X}^m$ denote an unlabeled sample. A learning algorithm $\mathcal{W}$ is a $(\gamma, \varepsilon_0, m_0)$–*agnostic weak learner* for $\mathcal{F}$ with respect to $X$ if for any labels $Y = (y_1, \ldots, y_m) \in \{\pm 1\}^m$,

$$\mathbb{E}_{\boldsymbol{S}'}\left[\frac{1}{m}\sum_{i=1}^{m} y_i\mathcal{W}(\boldsymbol{S}')(x_i)\right] \geq \gamma\sup_{f\in\mathcal{F}}\left[\frac{1}{m}\sum_{i=1}^{m} y_i f(x_i)\right] - \epsilon_0,$$

where $\boldsymbol{S}' = ((\mathbf{x}_1', \mathbf{y}_1'), \ldots, (\mathbf{x}_{m_0}', \mathbf{y}_{m_0}'))$ is an independent sample of size $m_0$ drawn from the uniform distribution over $S = ((x_1, y_1), \ldots, (x_m, y_m))$.

The following result states that the correlation of the output hypothesis is competitive with the best hypothesis in the reference class $\mathcal{F}$.

**Theorem A.5** (Brukhim et al. [2020], Theorem 7])**.** *There exists a boosting algorithm $\mathcal{A}$ that given a $(\gamma, \epsilon_0, m_0)$–agnostic weak learner $\mathcal{W}$ for $\mathcal{F} \subseteq \{\pm 1\}^{\mathcal{X}}$ and a sample $S = ((x_1, y_1), \ldots, (x_m, y_m))$ as input, runs for $T$ rounds and returns a hypothesis $\bar{f} \in \mathcal{F}$ satisfying that*

$$\mathbb{E}\left[\frac{1}{m}\sum_{i=1}^{m} y_i \bar{f}(x_i)\right] \geq \sup_{f \in \mathcal{F}} \mathbb{E}\left[\frac{1}{m}\sum_{i=1}^{m} y_i f(x_i)\right] - \left(\frac{\epsilon_0}{\gamma} + \mathcal{O}\left(\frac{1}{\gamma\sqrt{T}}\right)\right),$$

*where the expectation is taken over the randomness of $\mathcal{A}$ and $\mathcal{W}$, and the random samples given to $\mathcal{W}$.*[6]

In the paragraph following the above theorem (Theorem 7 in their text), Brukhim et al. [2020] argue that one can get a generalization bound up to an additive $\varepsilon$ term via sample compression, with a sample complexity of $m = \tilde{\mathcal{O}}(Tm_0/\varepsilon^2)$. To obtain a bound as in Kalai and Kanade [2009] (setting $T = 1/(\varepsilon^2\gamma^2)$ and assuming $m_0 = 1/\varepsilon_0^2$), the sample complexity becomes $m = \tilde{\mathcal{O}}(1/(\varepsilon^4\gamma^2\varepsilon_0^2))$.

### A.2.4    Ghai and Singh [2024]

Ghai and Singh [2024] employs a definition of agnostic weak learner quite close to ours. The only difference is that we assume our weak learner outputs a hypothesis $\mathcal{W} \in \mathcal{B}$ of range $[-1, 1]$ from a base hypothesis class $\mathcal{B}$. They prove the following.

**Theorem A.6** (Ghai and Singh [2024], Theorem 4])**.** *Fix $\epsilon, \delta > 0$. There exist a boosting algorithm $\mathcal{A}$ and a choice of $\eta, \sigma, T, \tau, S_0, S, m$ satisfying $T = \mathcal{O}(\log |\mathcal{B}|/(\gamma^2\epsilon^2)), \eta = \mathcal{O}(\gamma^2\epsilon/\log|\mathcal{B}|), \sigma = \eta/\gamma, \tau = \mathcal{O}(\gamma\epsilon), S = \mathcal{O}(1/(\gamma\epsilon)), S_0 = \mathcal{O}(1/\epsilon^2), m = \mathcal{O}(\log(|\mathcal{B}|/\delta)/\epsilon^2) + \mathcal{O}(1/(\gamma^2\epsilon^2))$ such that for any $\gamma$–agnostic weak learning oracle for hypothesis class $\mathcal{F}$ with finite base class $\mathcal{B}$, fixed tolerance $\epsilon_0$, and failure probability $\delta_0$, algorithm $\mathcal{A}$ outputs a hypothesis $\bar{f}$ such that with probability $1 - 10\delta_0 T - 10\delta T$,*

$$\mathrm{corr}_{\mathcal{D}}(\bar{f}) \geq \sup_{f \in \mathcal{F}} \mathrm{corr}_{\mathcal{D}}(f) - \frac{2\epsilon_0}{\gamma} - \epsilon,$$

*while making $T = \mathcal{O}(\log |\mathcal{B}|/(\gamma^2\epsilon^2))$ calls to the weak learning oracle, and sampling $TS + S_0 = \mathcal{O}((\log |\mathcal{B}|)/(\gamma^3\epsilon^3))$ labeled examples from $\mathcal{D}$.*

For infinite base hypothesis class $\mathcal{B}$, they obtain the analogous result with the VC dimension of $\mathcal{B}$ replacing $\log |\mathcal{B}|$ (Ghai and Singh [2024], Theorem 5]).

## B    AdaBoost Variation

In the following, we use a slight modification of ADABOOST that, in each boosting round, the weak learner will be run multiple times to enlarge the probability of it returning a hypothesis with the guaranteed correlation. The last part of the proof below follows directly from Schapire and Freund [2012, page 55, pages 111-112 and pages 277-278] and is included for completeness. Furthermore, we also follow much of the notation from Schapire and Freund [2012].

**Lemma B.1** (Restatement of 3.1)**.** *Let $\gamma', \delta_0 \in (0, 1)$, and given $m, m_0 \in \mathbb{N}$, let $S \in (\mathcal{X} \times \{\pm 1\})^m$. If a learning algorithm $\mathcal{W}: (\mathcal{X} \times \{\pm 1\})^* \to \mathcal{H} \subseteq [-1, 1]^{\mathcal{X}}$ is such that for any $\mathcal{Q} \in \Delta(S)$ with probability at least $1 - \delta_0$ over a sample $\mathbf{S}' \sim \mathcal{Q}^{m_0}$ the hypothesis $\mathbf{h} = \mathcal{W}(\mathbf{S}')$ satisfies $\mathrm{corr}_{\mathcal{Q}}(\mathbf{h}) \geq \gamma'$, then, for $T \geq \lceil 32 \ln(em)/\gamma'^2 \rceil$, running Algorithm 1 on input $(S, \mathcal{W}, m_0, T, \delta, \delta_0)$ yields a voting classifier $\mathbf{v} \in \mathrm{conv}(\mathcal{H})$ such that with probability at least $1 - \delta$ over the random draws from Line 5 it holds that $y\mathbf{v}(x) > \gamma'/8$ for all $(x, y) \in S$.*

*Proof.* Let $\mathbf{D}_1, \ldots, \mathbf{D}_T$, be the $T$, random distribution created over the $T$ rounds of boosting in Algorithm 1, where the randomness is over $\mathbf{S}_1^1, \mathbf{S}_1^2 \ldots, \mathbf{S}_T^k$, for shorten notation in the following we will for $t = 1, \ldots, T$ let $\mathbf{S}_t = (\mathbf{S}_t^1, \ldots, \mathbf{S}_t^k)$. We will show that for $t = 1, \ldots, T$, given any outcome

---

[6]We concluded the expectation to be over those sources of randomness from the proof of the theorem.

$D_1, \ldots D_t$ of $\mathbf{D}_1, \ldots, \mathbf{D}_t$ (which is given by a outcome $S_1 \ldots, S_{t-1}$ of $\mathbf{S}_1 \ldots, \mathbf{S}_{t-1}$) we have that the minimizer $\mathbf{h}_t$ between $\mathbf{h}_t^1, \ldots, \mathbf{h}_t^k$ is such that

$$\mathbb{E}_{(\mathbf{x},\mathbf{y}) \sim D_t}[\mathbf{y}\mathbf{h}_t(\mathbf{x})] \geq \gamma'$$

with probability at least $1 - \delta/T$ over $\mathbf{S}_t^1, \ldots \mathbf{S}_t^k$. Thus, by showing the above we get that

$$
\begin{aligned}
\mathbb{P}_{\mathbf{S}_1 \ldots, \mathbf{S}_T} &[\forall t \in \{1, \ldots, T\} \, \mathbb{E}_{(\mathbf{x},\mathbf{y}) \sim \mathbf{D}_t}[\mathbf{y}\mathbf{h}_t(\mathbf{x})] \geq \gamma'] \\
&= \mathbb{E}_{\mathbf{S}_1, \ldots, \mathbf{S}_T}\big[\mathbb{1}\{\forall t \in \{1, \ldots, T\} \, \mathbb{E}_{(\mathbf{x},\mathbf{y})}[\mathbf{y}\mathbf{h}_t(\mathbf{x})] \geq \gamma'\}\big] \\
&= \mathbb{E}_{\mathbf{S}_1, \ldots, \mathbf{S}_{T-1}}\big[\mathbb{E}_{\mathbf{S}_T}\big[\mathbb{1}\{\forall t \in \{1, \ldots, T\} \, \mathbb{E}_{(\mathbf{x},\mathbf{y})}[\mathbf{y}\mathbf{h}_t(\mathbf{x})] \geq \gamma'\}\big]\big] \\
&= \mathbb{E}_{\mathbf{S}_1, \ldots, \mathbf{S}_{T-1}}\big[\mathbb{E}_{\mathbf{S}_T}\big[\mathbb{1}\{\mathbb{E}_{(\mathbf{x},\mathbf{y})}[\mathbf{y}\mathbf{h}_T(\mathbf{x})] \geq \gamma'\}\big]\mathbb{1}\{\forall t \in \{1, \ldots, T-1\} \, \mathbb{E}_{(\mathbf{x},\mathbf{y})}[\mathbf{y}\mathbf{h}_t(\mathbf{x})] \geq \gamma'\}\big] \\
&= \mathbb{E}_{\mathbf{S}_1, \ldots, \mathbf{S}_{T-1}}\big[\mathbb{P}_{\mathbf{S}_T}\big[\mathbb{E}_{(\mathbf{x},\mathbf{y})}[\mathbf{y}\mathbf{h}_T(\mathbf{x})] \geq \gamma'\big]\mathbb{1}\{\forall t \in \{1, \ldots, T-1\} \, \mathbb{E}_{(\mathbf{x},\mathbf{y})}[\mathbf{y}\mathbf{h}_t(\mathbf{x})] \geq \gamma'\}\big] \\
&\geq (1 - \delta/T)\,\mathbb{E}_{\mathbf{S}_1, \ldots, \mathbf{S}_{T-1}}\big[\mathbb{1}\{\forall t \in \{1, \ldots, T-1\} \, \mathbb{E}_{(\mathbf{x},\mathbf{y})}[\mathbf{y}\mathbf{h}_t(\mathbf{x})] \geq \gamma'\}\big] \\
&= (1 - \delta/T)\,\mathbb{P}_{\mathbf{S}_1 \ldots, \mathbf{S}_{T-1}}\big[\forall t \in \{1, \ldots, T\} \, \mathbb{E}_{(\mathbf{x},\mathbf{y})}[\mathbf{y}\mathbf{h}_t(\mathbf{x})] \geq \gamma'\big] \\
&\geq (1 - \delta/T)^T \\
&\geq 1 - \delta,
\end{aligned}
$$

where the first inequality uses that given $\mathbf{S}_1, \ldots, \mathbf{S}_{t-1}$ (which determines $\mathbf{D}_t$), we have by the above claimed property that $\mathbb{P}_{\mathbf{S}_t}\big[\mathbb{E}_{(\mathbf{x},\mathbf{y}) \sim \mathbf{D}_t}[\mathbf{y}\mathbf{h}_t(\mathbf{x})] \geq \gamma'\big] \leq (1 - \delta/T)$ and the last inequality follows by Bernoulli's inequality.

Thus, we now show that for $t = 1, \ldots, T$, given any outcome $D_1, \ldots D_t$ of $\mathbf{D}_1, \ldots, \mathbf{D}_t$ we have that the minimizer $\mathbf{h}_t$ between $\mathbf{h}_t^1, \ldots, \mathbf{h}_t^k$ is such that

$$\mathbb{E}_{(\mathbf{x},\mathbf{y}) \sim D_t}[\mathbf{y}\mathbf{h}_t(\mathbf{x})] \geq \gamma'$$

with probability at least $1 - \delta/T$, over $\mathbf{S}_l$. We start with the former. To this end let $t \in \{1, \ldots, T\}$ and $D_1, \ldots, D_t$, be any outcome of $\mathbf{D}_1, \ldots, \mathbf{D}_t$, which only depends on $\mathbf{S}_1 \ldots, \mathbf{S}_{t-1}$. By Line 1 and Line 13 it follows that $D_t$ is such that $D_t(x, y) > 0$ only if $(x, y) \in S$. Thus, it holds for each $\ell = \{1 \ldots, k\}$ with probability at least $1 - \delta_0$ over $\mathbf{S}_t^\ell$ that $\mathbb{E}_{(\mathbf{x},\mathbf{y}) \sim D_t}[\mathbf{y}\mathbf{h}_t(\mathbf{x})] \geq \gamma'$. Now since $\mathbf{S}_t^1 \ldots, \mathbf{S}_t^k$ are sampled independently, it follows that the expected number of hypotheses with $\gamma'$ advantages is at least

$$\mu := \mathbb{E}_{\mathbf{S}_t^1, \ldots, \mathbf{S}_t^k}\left[\sum_{\ell=1}^k \mathbb{1}\{\mathbb{E}_{(\mathbf{x},\mathbf{y}) \sim D_t}[\mathbf{y}\mathbf{h}_t^\ell(\mathbf{x})] \leq \gamma'\}\right] \geq (1 - \delta_0)k,$$

and by the multiplicative Chernoff bound that

$$
\begin{aligned}
\mathbb{P}_{\mathbf{S}_t^1, \ldots, \mathbf{S}_t^k}\left[\sum_{\ell=1}^k \mathbb{1}\{\mathbb{E}_{(\mathbf{x},\mathbf{y}) \sim D_t}[\mathbf{y}\mathbf{h}_t^\ell(\mathbf{x})] \leq \gamma'\} \leq \mu/2\right] &\leq \exp\left(-\frac{\mu}{8}\right) \\
&\leq \exp\left(-\frac{(1 - \delta_0)k}{8}\right) \\
&\leq \frac{\delta}{T},
\end{aligned}
$$

where the last inequality follows by $k = \lceil 8\ln(eT/\delta)/(1 - \delta_0)\rceil$. This implies that with probability at least $1 - \delta/T$ over $\mathbf{S}_t$ we have that

$$
\begin{aligned}
\sum_{l=1}^k \mathbb{1}\{\mathbb{E}_{(\mathbf{x},\mathbf{y}) \sim D_t}[\mathbf{y}\mathbf{h}_t^i(\mathbf{x})] \leq \gamma'\} &> \frac{\mu}{2} \\
&\geq \frac{(1 - \delta_0)k}{2} \\
&\geq \frac{8\ln(eT/\delta)}{2} \\
&\geq 1,
\end{aligned}
$$

which implies that $\mathbf{h}_t$ satisfies

$$\mathbb{E}_{(\mathbf{x},\mathbf{y})\sim D_t}[\mathbf{y}\mathbf{h}_t(\mathbf{x})] \geq \gamma',$$

and concludes the first claim.

We for now consider outcomes $S_1, \ldots, S_T$ of $\mathbf{S}_1 \ldots, \mathbf{S}_T$ such that $r_t = \mathbb{E}_{(\mathbf{x},\mathbf{y})\sim\mathbf{D}_t}[\mathbf{y}\mathbf{h}_t(\mathbf{x})] \geq \gamma'$ for all $t \in [T]$. In the following, we closely follow Schapire and Freund [2012, page 55, page 111-112, and page 277-278]. We first notice that for any $(x,y) \in S$ we have that

$$
\begin{aligned}
D_{T+1}(x,y) &= \frac{D_T(x,y)\exp\left(-\boldsymbol{\alpha}_T y\mathbf{h}_T(x)\right)}{\mathbf{Z}_T} \\
&= \cdots \\
&= \frac{D_1(x,y)\exp\left(-\sum_{t=1}^{T}\boldsymbol{\alpha}_t y\mathbf{h}_t(x)\right)}{\prod_{t=1}^{T}\mathbf{Z}_t} \\
&= \frac{\exp\left(-\sum_{t=1}^{T}\boldsymbol{\alpha}_t y\mathbf{h}_t(x)\right)}{m\prod_{t=1}^{T}\mathbf{Z}_t},
\end{aligned}
$$

where the first equality follows from the definition of $D_{T+1}$, and similarly for the thirds equality using the definition of $D_T, \ldots, D_2$, and the last equality by $D_1 = 1/m$. Now using that $D_{T+1}$ is a probability distribution we get that

$$1 = \sum_{(x,y)\in S} \frac{\exp\left(-y\sum_{t=1}^{T}\boldsymbol{\alpha}_t \mathbf{h}_t(x)\right)}{m\prod_{t=1}^{T}\mathbf{Z}_t},$$

so

$$\prod_{t=1}^{T}\mathbf{Z}_t = \frac{1}{m}\sum_{(x,y)\in S}\exp\left(-y\sum_{t=1}^{T}\boldsymbol{\alpha}_t\mathbf{h}_t(x)\right). \tag{8}$$

Furthermore, we notice that for any $t \in \{1, \ldots, T\}$ we have that

$$
\begin{aligned}
\mathbf{Z}_t &= \sum_{(x,y)\in S} D_t(x,y)\exp\left(-\boldsymbol{\alpha}_t y\mathbf{h}_t(x)\right) \\
&= \sum_{(x,y)\in S} D_t(x,y)\exp\left(-\boldsymbol{\alpha}_t\frac{1+y\mathbf{h}_t(x)}{2} + \boldsymbol{\alpha}_t\frac{1-y\mathbf{h}_t(x)}{2}\right) \\
&\leq \sum_{(x,y)\in S} D_t(x,y)\left(\frac{1+y\mathbf{h}_t(x)}{2}\exp\left(-\boldsymbol{\alpha}_t\right) + \frac{1-y\mathbf{h}_t(x)}{2}\exp\left(\boldsymbol{\alpha}_t\right)\right) \\
&= \left(\frac{1+\mathbf{c}_t}{2}\right)\exp\left(-\boldsymbol{\alpha}_t\right) + \left(\frac{1-\mathbf{c}_t}{2}\right)\exp\left(\boldsymbol{\alpha}_t\right),
\end{aligned}
$$

where the inequality uses that $y\mathbf{h}_t(x) \in [-1,1]$, implying that $\frac{1+y\mathbf{h}_t(x)}{2}$ and $\frac{1-y\mathbf{h}_t(x)}{2}$ is scalars between $[0,1]$ summing to 1, and since the function $\exp(\cdot)$ is convex, the function value $\exp(\lambda(-\boldsymbol{\alpha}_t) + (1-\lambda)\boldsymbol{\alpha}_t)$ for any convex combination of $-\boldsymbol{\alpha}_t$ and $\boldsymbol{\alpha}_t$ ($1 \geq \lambda \geq 0$) is upper bounded by the convex combination $\lambda\exp\left(-\boldsymbol{\alpha}_t\right) + (1-\lambda)\exp\left(\boldsymbol{\alpha}_t\right)$. Furthermore, by inserting the value of $\boldsymbol{\alpha}_t = \frac{1}{2}\ln\left(\frac{1+\mathbf{c}_t}{1-\mathbf{c}_t}\right)$ in the above, we get that

$$
\begin{aligned}
\mathbf{Z}_t &\leq \left(\frac{1+\mathbf{c}_t}{2}\right)\exp\left(-\boldsymbol{\alpha}_t\right) + \left(\frac{1-\mathbf{c}_t}{2}\right)\exp\left(\boldsymbol{\alpha}_t\right) \\
&= \frac{1}{2}\sqrt{1-\mathbf{c}_t^2} + \frac{1}{2}\sqrt{1-\mathbf{c}_t^2} \\
&= \sqrt{1-\mathbf{c}_t^2}. \tag{9}
\end{aligned}
$$

Using the relation in Eq. (8) and Eq. (9) we get that, for $\beta = \gamma'/8$,

$$\frac{1}{m} \sum_{(x,y)\in S} \mathbb{1}\left\{ \frac{y \sum_{t=1}^{T} \boldsymbol{\alpha}_t \mathbf{h}_t(x)}{\sum_{t=1}^{T} \boldsymbol{\alpha}_t} \leq \beta \right\} \leq \frac{1}{m} \sum_{(x,y)\in S} \exp\left( \beta \sum_{t=1}^{T} \boldsymbol{\alpha}_t - y \sum_{t=1}^{T} \boldsymbol{\alpha}_t \mathbf{h}_t(x) \right)$$

$$\leq \exp\left( \beta \sum_{t=1}^{T} \boldsymbol{\alpha}_t \right) \cdot \prod_{t=1}^{T} \mathbf{Z}_t$$

$$= \prod_{t=1}^{T} \mathbf{Z}_t \exp\left( \beta \boldsymbol{\alpha}_t \right)$$

$$\leq \prod_{t=1}^{T} \sqrt{1 - \mathbf{c}_t^2} \left( \sqrt{\frac{1 + \mathbf{c}_t}{1 - \mathbf{c}_t}} \right)^{\beta}$$

$$= \prod_{t=1}^{T} \sqrt{(1 - \mathbf{c}_t)^{1-\beta}(1 + \mathbf{c}_t)^{1+\beta}}, \tag{10}$$

where the first inequality follows by $a \leq b$ implying that $1 \leq \exp(b - a)$, and the second by Eq. (8), the third inequality by Eq. (9) and the last equality by $1 - \mathbf{c}_t^2 = (1 - \mathbf{c}_t)(1 + \mathbf{c}_t)$. Now if $\mathbf{c}_t = 1$ then the above is 0 and we are done, so assume this is not the case for any $\mathbf{c}_t$. Now consider the function $f(x) = (1 - x)^{1-a}(1 + x)^{1+a}$ for $0 \leq x < 1$ which has derivative $\frac{2(a-x)}{(1-x)^a}(1 + x)^a$, thus is decreasing for $x \geq a$. Now we assumed that we considered a realization of $S_1, \ldots, S_T$ such that $\mathbf{c}_t \geq \gamma'$, for any $t \in \{1, \ldots, T\}$ and furthermore, we have that $\beta = \gamma'/8 < \gamma'$, whereby we conclude by the above argued monotonicity that $(1 - \mathbf{c}_t)^{1-\beta}(1 + \mathbf{c}_t)^{1+\beta} \leq (1 - \gamma')^{1-\beta}(1 + \gamma')^{1+\beta}$. Now plugging this into Eq. (10) we get that

$$\frac{1}{m} \sum_{(x,y)\in S} \mathbb{1}\{y\mathbf{v}(x) \leq \beta\} = \frac{1}{m} \sum_{(x,y)\in S} \mathbb{1}\left\{ \frac{y \sum_{t=1}^{T} \boldsymbol{\alpha}_t \mathbf{h}_t(x)}{\sum_{t=1}^{T} \boldsymbol{\alpha}_t} \leq \beta \right\}$$

$$\leq \left( \sqrt{(1 - \gamma')^{1-\beta}(1 + \gamma')^{1+\beta}} \right)^{T}$$

$$= \exp\left( T/2((1 - \beta)\ln(1 - \gamma') + (1 + \beta)\ln(1 + \gamma')) \right).$$

Furthermore, since we soon show that for $\beta = \gamma'/8$ it holds that

$$((1 - \beta)\ln(1 - \gamma') + (1 + \beta)\ln(1 + \gamma')) \leq -\gamma'^2/16, \tag{11}$$

we conclude that with probability at least $1 - \delta$ over $\mathbf{S}_1, \ldots, \mathbf{S}_T$ we have that

$$\sum_{(x,y)\in S} \mathbb{1}\{\mathbf{v}(x)y \leq \gamma'/8\} \leq m \exp\left( -T\gamma'^2/32 \right) < 1,$$

where the last inequality follows for $T \geq \lceil 32\ln(em)/\gamma'^2 \rceil$ and Eq. (11).

We now show that for $\beta = \gamma'/8$ it holds that $((1 - \beta)\ln(1 - \gamma') + (1 + \beta)\ln(1 + \gamma')) \leq -\gamma'^2/16$. To this end we consider the function $f(x) = x^2/16 + (1 - x/8)\ln(1 - x) + (1 + x/8)\ln(1 + x)$, for $0 \leq x < 1$. We first notice that

$$f'(x) = \frac{d}{dx}\left( \left(1 - \frac{x}{8}\right)\log(1 - x) + \left(1 + \frac{x}{8}\right)\log(1 + x) + \frac{x^2}{16} \right)$$

$$= \frac{1}{8}\left( \frac{x(x^2 + 13)}{x^2 - 1} - \ln(1 - x) + \ln(1 + x) \right),$$

and

$$f''(x) = \frac{d}{dx}\left( \frac{1}{8}\left( \frac{x(x^2 + 13)}{x^2 - 1} - \ln(1 - x) + \ln(1 + x) \right) \right)$$

$$= \frac{x^4 - 18x^2 - 11}{8(x^2 - 1)^2}.$$

Consider the values of $x$ in $[0, 1)$. We notice that $f''(x) < 0$, implying that $f'$ is a decreasing function. Thus, $0 = f'(0) \geq f'(x)$, so $f$ is a non-increasing function. Hence, $0 = f(0) \geq f(x)$, i.e., $f(x) = x^2/16 + (1 - x/8)\ln(1 - x) + (1 + x/8)\ln(1 + x) \leq 0$ for $0 \leq x < 1/2$, which implies that $(1 - x/8)\ln(1 - x) + (1 + x/8)\ln(1 + x) \leq -x^2/16+$ which give use that for $\beta = \gamma'/8$ it holds that $((1 - \beta)\ln(1 - \gamma') + (1 + \beta)\ln(1 + \gamma')) \leq -\gamma'^2/16$ since $0 < \gamma' < 1$. $\qquad\square$

## C  Margin bound

In this section we derive the following lemma.

**Lemma C.1** (Restatement of 3.2)**.** *There exist universal constants $C' \geq 1$ and $\hat{c} > 0$ for which the following holds. For all margin levels $0 \leq \gamma < \gamma' \leq 1$, hypothesis class $\mathcal{H} \subseteq [-1, 1]^{\mathcal{X}}$, and distribution $\mathcal{D} \in \Delta(\mathcal{X} \times \{\pm 1\})$, it holds with probability at least $1 - \delta$ over $\boldsymbol{S} \sim \mathcal{D}^m$ that for all $v \in \mathrm{conv}(\mathcal{H})$*

$$\mathcal{L}_{\mathcal{D}}^{\gamma}(v) \leq \mathcal{L}_{\boldsymbol{S}}^{\gamma'}(v) + C'\left(\sqrt{\mathcal{L}_{\boldsymbol{S}}^{\gamma'}(v) \cdot \frac{\beta}{m}} + \frac{\beta}{m}\right),$$

*where*

$$\beta = \frac{d}{(\gamma' - \gamma)^2}\mathrm{Ln}^{3/2}\left(\frac{(\gamma' - \gamma)^2 m}{d}\frac{\gamma'}{\gamma' - \gamma}\right) + \ln\frac{1}{\delta}, \qquad and \qquad d = \mathrm{fat}_{\hat{c}(\gamma' - \gamma)}(\mathcal{H}).$$

To show Lemma 3.2 we need the following two lemmas Lemma C.2 and Lemma C.3. We now present theses two lemmas and show how they imply Lemma 3.2, after we have shown how they imply Lemma 3.2 we give their proof.

**Lemma C.2.** *There exist universal constants $c = 128$ such that: For $0 \leq \gamma < \gamma' \leq 1$, $0 < \tau \leq 1$ distribution $\mathcal{D}$ over $\mathbf{X} \times \{\pm 1\}$, hypothesis class $\mathcal{H} \subseteq [-1, 1]^{\mathcal{X}}$, it holds with probability at least $1 - \sup_{X \in \mathcal{X}^{2m}} |N_{\infty}(X, \mathrm{conv}(\mathcal{H})_{\lceil 2\gamma' \rceil}, \frac{\gamma' - \gamma}{2})|\delta$ over $\mathbf{S} \sim \mathcal{D}^m$ that for all $v \in \mathrm{conv}(\mathcal{H})$ either*

$$\mathcal{L}_{\mathbf{S}}^{\gamma'}(v) > \tau$$

*or*

$$\mathcal{L}_{\mathcal{D}}^{\gamma}(v) \leq \tau + c\left(\sqrt{\tau\frac{\ln(e/\delta)}{m}} + \frac{\ln(e/\delta)}{m}\right).$$

**Lemma C.3.** *There exists universal constants $C \geq 1$, $\hat{C} \geq 1$ and $\hat{c} > 0$ such that: For margin levels $0 \leq \gamma < \gamma' \leq 1$, hypothesis class $\mathcal{H} \subseteq [-1, 1]^{\mathcal{X}}$ and $X \in \mathcal{X}^m$, where $m \geq \frac{\hat{C}\mathrm{fat}_{\hat{c}(\gamma' - \gamma)}(\mathcal{H})}{(\gamma' - \gamma)^2}$*

$$\ln\left(|N_{\infty}(X, \mathrm{conv}(\mathcal{H})_{\lceil 2\gamma' \rceil}, \frac{\gamma' - \gamma}{2})|\right) \leq \frac{C\hat{C}\mathrm{fat}_{\hat{c}(\gamma' - \gamma)}(\mathcal{H})}{(\gamma' - \gamma)^2}\ln^{3/2}\left(\frac{(\gamma' - \gamma)^2 m}{\hat{C}\mathrm{fat}_{\hat{c}(\gamma' - \gamma)}(\mathcal{H})}\frac{8\gamma'}{\gamma' - \gamma}\right).$$

With Lemma C.2 and Lemma C.3 stated, we now prove Lemma 3.2.

*Proof of Lemma 3.2.* Let $C, \hat{C} \geq 1$ and $\hat{c}$ be the universal constants from Lemma C.3 and $c \geq 1$ the universal constant from Lemma C.2. Moreover, let

$$\Delta_{\gamma} = \gamma' - \gamma.$$

In the following, we will show that with probability at least $1 - \delta$ over $\mathbf{S}$ it holds that: For all $v \in \mathrm{conv}(\mathcal{H})$

$$\mathcal{L}_{\mathcal{D}}^{\gamma}(v) \leq \mathcal{L}_{\mathbf{S}}^{\gamma'}(v) + c\left[\sqrt{\mathcal{L}_{\mathbf{S}}^{\gamma'}(v)\left(\frac{\ln(1/\delta)}{m} + \frac{3C\hat{C}\mathrm{fat}_{\hat{c}\Delta_{\gamma}}(\mathcal{H})\mathrm{Ln}^{3/2}\left(\frac{2\Delta_{\gamma}^2 m}{\hat{C}\mathrm{fat}_{\hat{c}\Delta_{\gamma}}(\mathcal{H})}\frac{8\gamma'}{\Delta_{\gamma}}\right)}{\Delta_{\gamma}^2 m}\right)} \right.$$

$$\left. + \frac{2\ln(1/\delta)}{m} + \frac{6C\hat{C}\mathrm{fat}_{\hat{c}\Delta_{\gamma}}(\mathcal{H})\mathrm{Ln}^{3/2}\left(\frac{2\Delta_{\gamma}^2 m}{\hat{C}\mathrm{fat}_{\hat{c}\Delta_{\gamma}}(\mathcal{H})}\frac{8\gamma'}{\Delta_{\gamma}}\right)}{\Delta_{\gamma}^2 m}\right].$$

We notice that we may assume that $m \geq \frac{C\hat{C}\operatorname{fat}_{\hat{c}\Delta_\gamma}(\mathcal{H})}{\Delta_\gamma^2}$, since else the right hand side of above is greater than 1 and the left hand side is at most 1. Furthermore, we may also assume that $m$ is so large that $\frac{1}{m} \cdot \frac{C\hat{C}\operatorname{fat}_{\hat{c}\Delta_\gamma}(\mathcal{H})}{\Delta_\gamma^2} \operatorname{Ln}^{3/2}\left(\frac{2\Delta_\gamma^2 m}{\hat{C}\operatorname{fat}_{\hat{c}\Delta_\gamma}(\mathcal{H})} \frac{8\gamma'}{\Delta_\gamma}\right) < 1$, where $m \geq \frac{C\hat{C}\operatorname{fat}_{\hat{c}\Delta_\gamma}(\mathcal{H})}{\Delta_\gamma^2}$, and $C \geq 1$ implies that the $\operatorname{Ln}^{3/2}$ term is equal to $\ln^{3/2}$. Now let $N = \exp\left(\frac{C\hat{C}\operatorname{fat}_{\hat{c}\Delta_\gamma}(\mathcal{H})}{\Delta_\gamma^2} \ln^{3/2}\left(\frac{2\Delta_\gamma^2 m}{\hat{C}\operatorname{fat}_{\hat{c}\Delta_\gamma}(\mathcal{H})} \frac{8\gamma'}{\Delta_\gamma}\right)\right) \geq 1$ (by the just argued size of $m$) and define $\tau_i = i\frac{\ln(N)}{m}$ for $i \in I = \{1, \ldots, \left\lfloor \frac{m}{\ln(N)} \right\rfloor, \frac{m}{\ln(N)}\}$. Noticing that the above conditions on $m$ imply that $\frac{m}{\ln(N)} \geq 1$, we have that $|I| \leq \left\lfloor \frac{m}{\ln(N)} \right\rfloor + 2 \leq 3\frac{m}{\ln(N)}$. Furthermore, let $\delta' = \delta \ln(N)/(3Nm)$. Now for each $i \in I$ we invoke Lemma 3.2 with $\tau_i$ and get by the union bound that with probability at least $1 - |I|\sup_{X \in \mathcal{X}^{2m}} |N_\infty(X, \operatorname{conv}(\mathcal{H})_{\lceil 2\gamma' \rceil}, \frac{\Delta_\gamma}{2})|\delta'$ over $\mathbf{S} \sim \mathcal{D}^m$ it holds for all $i \in I$ and all $v \in \operatorname{conv}(\mathcal{H})$ that

$$\mathcal{L}_{\mathbf{S}}^{\gamma'}(v) > \tau_i$$

or

$$\mathcal{L}_{\mathcal{D}}^{\gamma}(v) \leq \tau_i + c\left(\sqrt{\tau_i\frac{\ln(e/\delta')}{m}} + \frac{\ln(e/\delta')}{m}\right).$$

Now by Lemma C.3 we have

$$\sup_{X \in \mathcal{X}^{2m}} |N_\infty(X, \operatorname{conv}(\mathcal{H})_{\lceil 2\gamma' \rceil}, \frac{\Delta_\gamma}{2})| \leq \exp\left(\frac{C\hat{C}\operatorname{fat}_{\hat{c}\Delta_\gamma}(\mathcal{H})}{\Delta_\gamma^2} \ln^{3/2}\left(\frac{2\Delta_\gamma^2 m}{\hat{C}\operatorname{fat}_{\hat{c}\Delta_\gamma}(\mathcal{H})} \frac{8\gamma'}{\Delta_\gamma}\right)\right)$$
$$= N,$$

and we concluded earlier that $|I| \leq \frac{3m}{\ln(N)}$ thus by $\delta' = \delta \ln(N)/(3Nm)$, we get that

$$|I|\sup_{X \in \mathcal{X}^{2m}} |N_\infty(X, \operatorname{conv}(\mathcal{H})_{\lceil 2\gamma' \rceil}, \frac{\Delta_\gamma}{2})|\delta' \leq \delta,$$

whereby we conclude that probability at least $1 - \delta$ over $\mathbf{S} \sim \mathcal{D}^m$ it holds for all $i \in I$ and all $v \in \operatorname{conv}(\mathcal{H})$ that:

$$\mathcal{L}_{\mathbf{S}}^{\gamma'}(v) > \tau_i$$

or

$$\mathcal{L}_{\mathcal{D}}^{\gamma}(v) \leq \tau_i + c\left(\sqrt{\tau_i\frac{\ln(e/\delta')}{m}} + \frac{\ln(e/\delta')}{m}\right).$$

Now on this event, we notice that since $\mathcal{L}_{\mathbf{S}}^{\gamma'}(v) \in [0, 1]$ for any $v$ and $\cup_{i \in I}[\tau_i, \tau_i] = [\ln(N)/m, 1]$ it must be the case that for any $v \in \operatorname{conv}(\mathcal{H})$ there exists an largest $i \in I$ such that $\tau_{i-1} \leq \mathcal{L}_{\mathbf{S}}^{\gamma'} \leq \tau_i$, with $\tau_0 = 0$. Now for this $i$ it must be the case that

$$\mathcal{L}_{\mathcal{D}}^{\gamma}(v) \leq \tau_i + c\left(\sqrt{\tau_i\frac{\ln(e/\delta')}{m}} + \frac{\ln(e/\delta')}{m}\right).$$

and since $\tau_i \leq \tau_{i-1} + \frac{\ln(N)}{m} \leq \mathcal{L}_{\mathbf{S}}^{\gamma'}(v) + \frac{\ln(N)}{m}$, the above implies that

$$\mathcal{L}_{\mathcal{D}}^{\gamma}(v) \leq \tau_i + c\left(\sqrt{\tau_i\frac{\ln(e/\delta')}{m}} + \frac{\ln(e/\delta')}{m}\right)$$
$$\leq \mathcal{L}_{\mathbf{S}}^{\gamma'}(v) + \frac{\ln(N)}{m} + c\left(\sqrt{\mathcal{L}_{\mathbf{S}}^{\gamma'}(v)\frac{\ln(e/\delta')}{m}} + \frac{\sqrt{\ln(e/\delta')\ln(N)}}{m} + \frac{\ln(e/\delta')}{m}\right)$$
$$\leq \mathcal{L}_{\mathbf{S}}^{\gamma'}(v) + c\left(\sqrt{\mathcal{L}_{\mathbf{S}}^{\gamma'}(v)\frac{\ln(e/\delta')}{m}} + 2\frac{\ln(N) + \ln(e/\delta')}{m}\right), \tag{12}$$

where the second inequality follows from $\tau_{i-1} \leq \mathcal{L}_{\mathbf{S}}^{\gamma'}(v) + \frac{\ln(N)}{m}$, and by $\sqrt{a+b} \leq \sqrt{a} + \sqrt{b}$ for $a, b > 0$ and the third by $\sqrt{a \cdot b} \leq a + b$ for $a, b > 0$ and $c \geq 1$. Furthermore, as $\delta' = \delta \ln(N)/(3Nm)$, and $N = \exp\left(\frac{C\hat{C}\,\mathrm{fat}_{\hat{c}\Delta_\gamma}(\mathcal{H})}{\Delta_\gamma^2}\ln^{3/2}\left(\frac{2\Delta_\gamma^2 m}{\hat{C}\,\mathrm{fat}_{\hat{c}\Delta_\gamma}(\mathcal{H})}\frac{8\gamma'}{\Delta_\gamma}\right)\right)$ we get that

$$\frac{\ln(N) + \ln(e/\delta')}{m} = \frac{\ln(3/\delta) + \ln(em/\ln(N)) + 2\ln(N)}{m}$$

$$= \frac{\ln(3/\delta) + \ln\left(\frac{\Delta_\gamma^2 em}{C\hat{C}\,\mathrm{fat}_{\hat{c}\Delta_\gamma}(\mathcal{H})\ln^{3/2}\left(\frac{2\Delta_\gamma^2 m}{\hat{C}\,\mathrm{fat}_{\hat{c}\Delta_\gamma}(\mathcal{H})}\frac{8\gamma'}{\Delta_\gamma}\right)}\right) + 2\ln(N)}{m}$$

$$\leq \frac{\ln(3/\delta) + \mathrm{Ln}\left(\frac{\Delta_\gamma^2 em}{C\hat{C}\,\mathrm{fat}_{\hat{c}\Delta_\gamma}(\mathcal{H})}\right) + 2\frac{C\hat{C}\,\mathrm{fat}_{\hat{c}\Delta_\gamma}(\mathcal{H})}{\Delta_\gamma^2}\mathrm{Ln}^{3/2}\left(\frac{2\Delta_\gamma^2 m}{\hat{C}\,\mathrm{fat}_{\hat{c}\Delta_\gamma}(\mathcal{H})}\frac{8\gamma'}{\Delta_\gamma}\right)}{m}$$

$$\leq \frac{\ln(3/\delta)}{m} + \frac{3C\hat{C}\,\mathrm{fat}_{\hat{c}\Delta_\gamma}(\mathcal{H})\mathrm{Ln}^{3/2}\left(\frac{2\Delta_\gamma^2 m}{\hat{C}\,\mathrm{fat}_{\hat{c}\Delta_\gamma}(\mathcal{H})}\frac{8\gamma'}{\Delta_\gamma}\right)}{\Delta_\gamma^2 m}, \tag{13}$$

where the first inequality follows $\ln \leq \mathrm{Ln}$, and by us considering the case where $m$ is such that $\ln^{3/2}\left(\frac{2\Delta_\gamma^2 m}{\hat{C}\,\mathrm{fat}_{\hat{c}\Delta_\gamma}(\mathcal{H})}\frac{8\gamma'}{\Delta_\gamma}\right) \geq 1$, and $\mathrm{Ln}$ being an increasing function and the definition of $N$, the last inequality follows by $\frac{16\gamma'}{\Delta_\gamma} \geq 1$, $\frac{C\hat{C}\,\mathrm{fat}_{\hat{c}\Delta_\gamma}(\mathcal{H})}{(\Delta_\gamma)^2} \geq 1$ and $C \geq 1$ We now give the proof of Lemma C.2 and Lemma C.3, where we start with the former.

Now plugging the upper bound on $\ln(e/\delta')/m$ of Eq. (13) into Eq. (12) we conclude that

$$\mathcal{L}_\mathcal{D}^\gamma(v) \leq \mathcal{L}_{\mathbf{S}}^{\gamma'}(v) + c\left(\sqrt{\mathcal{L}_{\mathbf{S}}^{\gamma'}(v)\frac{\ln(e/\delta')}{m}} + 2\frac{\ln(N) + \ln(e/\delta')}{m}\right)$$

$$\leq \mathcal{L}_{\mathbf{S}}^{\gamma'}(v) + c\left(\sqrt{\mathcal{L}_{\mathbf{S}}^{\gamma'}(v)\left(\frac{\ln(3/\delta)}{m} + \frac{3C\hat{C}\,\mathrm{fat}_{\hat{c}\Delta_\gamma}(\mathcal{H})\mathrm{Ln}^{3/2}\left(\frac{2\Delta_\gamma^2 m}{\hat{C}\,\mathrm{fat}_{\hat{c}\Delta_\gamma}(\mathcal{H})}\frac{8\gamma'}{\Delta_\gamma}\right)}{\Delta_\gamma^2 m}\right)}\right.$$

$$\left. + \frac{2\ln(3/\delta)}{m} + \frac{6C\hat{C}\,\mathrm{fat}_{\hat{c}\Delta_\gamma}(\mathcal{H})\mathrm{Ln}^{3/2}\left(\frac{2\Delta_\gamma^2 m}{\hat{C}\,\mathrm{fat}_{\hat{c}\Delta_\gamma}(\mathcal{H})}\frac{8\gamma'}{\Delta_\gamma}\right)}{\Delta_\gamma^2 m}\right).$$

$\square$

*Proof of Lemma C.2.* In the following we consider the event $\exists v \in \mathrm{conv}(\mathcal{H})$ such that

$$\mathcal{L}_{\mathbf{S}}^{\gamma'}(v) \leq \tau$$

and

$$\mathcal{L}_\mathcal{D}^\gamma(v) > \tau + c\left(\sqrt{\tau\frac{\ln(e/\delta)}{m}} + \frac{\ln(e/\delta)}{m}\right),$$

and show that this happens with probability at most $\delta$. Let $\beta = \sqrt{\tau\ln(e/\delta)/m} + \ln(e/\delta)/m$ and $E = \{\exists v \in \mathrm{conv}(\mathcal{H}) : \mathcal{L}_{\mathbf{S}}^{\gamma'}(v) \leq \tau, \mathcal{L}_\mathcal{D}^\gamma > \tau + c\beta\}$ denote the above event. We notice that if $\frac{c\ln(e/\delta)}{m} \geq 1$ then the above holds with probability at most 0, since $\mathcal{L}_\mathcal{D}^\gamma(v) \leq 1$ for any $v \in \mathrm{conv}(\mathcal{H})$. Thus, we from now on consider the case that $\frac{c\ln(e/\delta)}{m} < 1$.

**Observation C.4.** In what follows we will use that for $a > 0$ we have that the function $x - \sqrt{ax}$ in $x$ is increasing for $x \geq a/4$, since it has derivative $1 - \frac{a}{2\sqrt{ax}}$.

We now show that

$$\mathbb{P}_{\mathbf{S} \sim \mathcal{D}^m}(E) \leq (1 - \delta/e) \mathbb{P}_{\mathbf{S},\mathbf{S}' \sim \mathcal{D}^m} \left( \exists v \in \mathrm{conv}(\mathcal{H}) : \mathcal{L}_{\mathbf{S}}^{\gamma'}(v) \leq \tau, \mathcal{L}_{\mathbf{S}'}^{\gamma}(v) \geq \tau + c\beta/2 \right).$$

Let $S$ be a realization of $\mathbf{S}$ in $E$ and let $v \in \mathrm{conv}(\mathcal{H})$ be a hypothesis realizing the above condition of the event $E$. Since $v$ is now fixed we conclude by the multiplicative Chernoff bound and $\mathcal{L}_{\mathcal{D}}^{\gamma}(v) \geq c \frac{\ln(e/\delta)}{m}$ for outcomes implying that $\frac{2\ln(e/\delta)}{\mathcal{L}_{\mathcal{D}}^{\gamma}(v)m} < 1$, since $c \geq 4$, we have that

$$\mathbb{P}_{\mathbf{S}' \sim \mathcal{D}^m} \left( \mathcal{L}_{\mathbf{S}'}^{\gamma}(v) \leq (1 - \sqrt{\frac{2\ln(e/\delta)}{\mathcal{L}_{\mathcal{D}}^{\gamma}(v)m}}) \mathcal{L}_{\mathcal{D}}^{\gamma}(v) \right) \leq \delta/e.$$

Thus with probability at least $1 - \delta/e$ we have that

$$\mathcal{L}_{\mathbf{S}'}^{\gamma}(v) \geq \mathcal{L}_{\mathcal{D}}^{\gamma}(v) - \sqrt{\frac{\mathcal{L}_{\mathcal{D}}^{\gamma}(v)2\ln(e/\delta)}{m}}$$

Now using that $\mathcal{L}_{\mathcal{D}}^{\gamma}(v) \geq \tau + c\left( \sqrt{\tau \frac{\ln(e/\delta)}{m}} + \frac{\ln(e/\delta)}{m} \right)$ it follows from Observation C.4 with $a = \frac{2\ln(e/\delta)}{m}$ and $x = \mathcal{L}_{\mathcal{D}}^{\gamma}(v)$ and $c \geq 1/2$ that,

$$\mathcal{L}_{\mathbf{S}'}^{\gamma}(v) \geq \mathcal{L}_{\mathcal{D}}^{\gamma}(v) - \sqrt{\frac{\mathcal{L}_{\mathcal{D}}^{\gamma}(v)2\ln(e/\delta)}{m}}$$

$$\geq \tau + c\left( \sqrt{\tau \frac{\ln(e/\delta)}{m}} + \frac{\ln(e/\delta)}{m} \right) - \sqrt{\frac{\left( \tau + c\left( \sqrt{\tau \frac{\ln(e/\delta)}{m}} + \frac{\ln(e/\delta)}{m} \right) \right) 2\ln(e/\delta)}{m}}.$$

(14)

Now using that $c \geq 1$, $a + b + \sqrt{ab} \leq 2(a + b)$ and $\sqrt{a + b} \leq \sqrt{a} + \sqrt{b}$ we get that the second term in the above is at most

$$\sqrt{\frac{\left( \tau + c\left( \sqrt{\tau \frac{\ln(e/\delta)}{m}} + \frac{\ln(e/\delta)}{m} \right) \right) 2\ln(e/\delta)}{m}} \leq \sqrt{c \frac{\left( \tau + \sqrt{\tau \frac{\ln(e/\delta)}{m}} + \frac{\ln(e/\delta)}{m} \right) 2\ln(e/\delta)}{m}}$$

$$\leq \sqrt{2c \frac{\left( \tau + \frac{\ln(e/\delta)}{m} \right) 2\ln(e/\delta)}{m}}$$

$$\leq \sqrt{c} 2 \left( \sqrt{\frac{\tau \ln(e/\delta)}{m}} + \frac{\ln(e/\delta)}{m} \right).$$

Thus, we conclude that Eq. (14) is lower bounded by

$$\mathcal{L}_{\mathbf{S}'}^{\gamma}(v) \geq \tau + (c - \sqrt{c}2) \left( \sqrt{\frac{\tau \ln(e/\delta)}{m}} + \frac{\ln(e/\delta)}{m} \right)$$

$$\geq \tau + c/2 \left( \sqrt{\frac{\tau \ln(e/\delta)}{m}} + \frac{\ln(e/\delta)}{m} \right)$$

$$= \tau + c\beta/2,$$

where the last inequality follows by $c \geq 164$ and $c - \sqrt{c}2 - c/2 \geq 0$ for $c > 16$. Thus we conclude by the law of total probability that

$$\mathbb{P}_{\mathbf{S},\mathbf{S}' \sim \mathcal{D}^m} \left( \exists v \in \mathrm{conv}(\mathcal{H}) : \mathcal{L}_{\mathbf{S}}^{\gamma'}(v) \leq \tau, \mathcal{L}_{\mathbf{S}'}^{\gamma}(v) \geq \tau + c\beta/2 \right)$$

$$\geq \mathbb{P}_{\mathbf{S},\mathbf{S}' \sim \mathcal{D}^m} \left( \exists v \in \mathrm{conv}(\mathcal{H}) : \mathcal{L}_{\mathbf{S}}^{\gamma'}(v) \leq \tau, \mathcal{L}_{\mathbf{S}'}^{\gamma}(v) \geq \tau + c\beta/2 \Big| E \right) \mathbb{P}_{\mathbf{S} \sim \mathcal{D}^m}(E)$$

$$\geq (1 - \frac{\delta}{e}) \mathbb{P}_{\mathbf{S} \sim \mathcal{D}^m}(E).$$

We now show that the term in the first line of the above inequalities is at most

$$\sup_{X \in \mathcal{X}^{2m}} |N_\infty(X, \text{conv}(\mathcal{H})_{\lceil 2\gamma' \rceil}, \frac{\gamma' - \gamma}{2})| \delta / e,$$

which implies that

$$\mathbb{P}_{\mathbf{S} \sim \mathcal{D}^m}(E) \leq |N_\infty(X, \text{conv}(\mathcal{H})_{\lceil 2\gamma' \rceil}, \frac{\gamma' - \gamma}{2})| \delta,$$

and would conclude the proof

.

To this end we notice that since $\mathbf{S}, \mathbf{S}' \sim \mathcal{D}^m$ are i.i.d. samples we may view them as drawn in the following way: First we draw $\tilde{\mathbf{S}} \sim \mathcal{D}^{2m}$, and then $\mathbf{S}$ is formed by drawing $m$ times without replacement from $\tilde{\mathbf{S}}$, and $\mathbf{S}'$ is set equal to the remaining elements in $\tilde{\mathbf{S}}$, $\mathbf{S}' = \tilde{\mathbf{S}} \backslash \mathbf{S}$. We will write drawing $\mathbf{S}$ and $\mathbf{S}'$ from $\tilde{\mathbf{S}}$ as $\mathbf{S}, \mathbf{S}' \sim \tilde{\mathbf{S}}$. We then have that

$$\mathbb{P}_{\mathbf{S}, \mathbf{S}' \sim \mathcal{D}^m}\left(\exists v \in \text{conv}(\mathcal{H}) : \mathcal{L}_{\mathbf{S}}^{\gamma'}(v) \leq \tau, \mathcal{L}_{\mathbf{S}'}^{\gamma}(v) \geq \tau + c\beta/2\right)$$

$$= \mathbb{E}_{\tilde{\mathbf{S}} \sim \mathcal{D}^{2m}}\left[\mathbb{P}_{\mathbf{S}, \mathbf{S}' \sim \tilde{\mathbf{S}}}\left(\exists v \in \text{conv}(\mathcal{H}) : \mathcal{L}_{\mathbf{S}}^{\gamma'}(v) \leq \tau, \mathcal{L}_{\mathbf{S}'}^{\gamma}(v) \geq \tau + c\beta/2\right)\right]$$

$$\leq \sup_{Z \in (\mathcal{X} \times \{\pm 1\})^{2m}} \mathbb{P}_{\mathbf{S}, \mathbf{S}' \sim Z}\left(\exists v \in \text{conv}(\mathcal{H}) : \mathcal{L}_{\mathbf{S}}^{\gamma'}(v) \leq \tau, \mathcal{L}_{\mathbf{S}'}^{\gamma}(v) \geq \tau + c\beta/2\right).$$

We now show that for any $Z \in (\mathcal{X} \times \{\pm 1\})^{2m}$ the probability over $\mathbf{S}, \mathbf{S}' \sim Z$ in the last line of the above is at most $|N_\infty(X, \text{conv}(\mathcal{H})_{\lceil 2\gamma' \rceil}, \frac{\gamma' - \gamma}{2})| \delta / e$, as claimed, which would conclude the proof. To this end let now $Z = (X, Y) \in (\mathcal{X} \times \{\pm 1\})^{2m}$, where $X \in \mathcal{X}^{2m}$ are the points in $Z$ and $Y \in \{\pm 1\}^m$ the labels in $Z$. We recall that for $v \in \text{conv}(\mathcal{H})$

$$v_{\lceil \alpha \rceil}(x) = \begin{cases} \alpha & \text{if } v(x) \geq \alpha \\ v(x) & \text{if } -\alpha < v(x) < \alpha \\ -\alpha & \text{if } v(x) \leq -\alpha \end{cases}$$

Furthermore, we notice that for $0 \leq \alpha < \alpha' < 1$, $v \in \text{conv}(\mathcal{H})$ and $(x, y)$ such that $v(x)y > \alpha'$ then we also have that $v_{\lceil 2\alpha' \rceil}(x)y > \alpha$ and $(x, y)$ such that $v(x)y \leq \alpha$ then we also have that $v_{\lceil 2\alpha' \rceil}(x)y \leq \alpha$. Thus, since $0 \leq \gamma < \gamma' < 1$ and $\text{conv}(\mathcal{H})_{\lceil 2\gamma' \rceil} = \{v' : v' = v_{\lceil 2\gamma' \rceil}, v \in \text{conv}(\mathcal{H})\}$ we conclude that

$$\mathbb{P}_{\mathbf{S}, \mathbf{S}' \sim Z}\left(\exists v \in \text{conv}(\mathcal{H}) : \mathcal{L}_{\mathbf{S}}^{\gamma'}(v) \leq \tau, \mathcal{L}_{\mathbf{S}'}^{\gamma}(v) \geq \tau + c\beta/2\right)$$

$$\leq \mathbb{P}_{\mathbf{S}, \mathbf{S}' \sim Z}\left(\exists v \in \text{conv}(\mathcal{H})_{\lceil 2\gamma' \rceil} : \mathcal{L}_{\mathbf{S}}^{\gamma'}(v) \leq \tau, \mathcal{L}_{\mathbf{S}'}^{\gamma}(v) \geq \tau + c\beta/2\right).$$

Let $N_\infty = N_\infty(X, \text{conv}(\mathcal{H})_{\lceil 2\gamma' \rceil}, \frac{\gamma' - \gamma}{2})$ be a $\frac{\gamma' - \gamma}{2}$-cover for $\text{conv}(\mathcal{H})_{\lceil 2\gamma' \rceil}$, in infinity norm on $X$ i.e., $\forall v \in \text{conv}(\mathcal{H})_{\lceil 2\gamma' \rceil}$ there exists $v' \in N_\infty$ such that $\max_{x \in X} |v(x) - v'(x)| \leq \frac{\gamma' - \gamma}{2}$. We now notice that for $v \in \text{conv}(\mathcal{H})_{\lceil 2\gamma' \rceil}$, $v' \in N_\infty$ the closest element in $N_\infty$ to $v$ in infinity norm and $(x, y) \in Z$ be such that $v(x)y > \gamma'$ then $v'(x)y = v(x)y + (v'(x) - v(x))y \geq v(x)y - \frac{\gamma' - \gamma}{2} > \frac{\gamma' + \gamma}{2}$. Furthermore, for $(x, y) \in Z$ such that $v(x)y \leq \gamma$ we have that $v'(x)y = v(x)y + (v'(x) - v(x))y \leq v(x)y + \frac{\gamma' - \gamma}{2} \leq \frac{\gamma' + \gamma}{2}$. Thus, we conclude that

$$\mathbb{P}_{\mathbf{S}, \mathbf{S}' \sim Z}\left(\exists v \in \text{conv}(\mathcal{H})_{\lceil 2\gamma' \rceil} : \mathcal{L}_{\mathbf{S}}^{\gamma'}(v) \leq \tau, \mathcal{L}_{\mathbf{S}'}^{\gamma}(v) \geq \tau + c\beta/2\right)$$

$$\leq \mathbb{P}_{\mathbf{S}, \mathbf{S}' \sim Z}\left(\exists v \in N_\infty : \mathcal{L}_{\mathbf{S}}^{\frac{\gamma' + \gamma}{2}}(v) \leq \tau, \mathcal{L}_{\mathbf{S}'}^{\frac{\gamma' + \gamma}{2}}(v) \geq \tau + c\beta/2\right)$$

$$\leq \sum_{v \in N_\infty} \mathbb{P}_{\mathbf{S}, \mathbf{S}' \sim Z}\left(\mathcal{L}_{\mathbf{S}}^{\frac{\gamma' + \gamma}{2}}(v) \leq \tau, \mathcal{L}_{\mathbf{S}'}^{\frac{\gamma' + \gamma}{2}}(v) \geq \tau + c/2\left(\sqrt{\frac{\tau \ln(e/\delta)}{m}} + \frac{\ln(e/\delta)}{m}\right)\right).$$

where the last inequality follows by the union bound over $N_\infty$, and the definition of $\beta$. We now show that each term in the sum over $v \in N_\infty$ is bounded by $\delta/e$ which would give that the above is at most $|N_\infty(X, \mathrm{conv}(\mathcal{H})_{\lceil 2\gamma'\rceil}, \frac{\gamma'-\gamma}{2})|\delta/e$, as claimed earlier and conclude the proof.

To this end consider $v \in N_\infty$, and let $\mu = (\mathcal{L}_{\mathbf{S}}^{\frac{\gamma'+\gamma}{2}}(v) + \mathcal{L}_{\mathbf{S}'}^{\frac{\gamma'+\gamma}{2}}(v))/2$, i.e., the fraction of points in $X$ that has less than $(\gamma'+\gamma)/2$-margin. We first notice that for $v$ in the above sum such that

$$2\mu = \mathcal{L}_{\mathbf{S}}^{\frac{\gamma'+\gamma}{2}}(v) + \mathcal{L}_{\mathbf{S}'}^{\frac{\gamma'+\gamma}{2}}(v) < \tau + c/2\left(\sqrt{\frac{\tau\ln(e/\delta)}{m}} + \frac{\ln(e/\delta)}{m}\right)$$

, the term is $0$. Thus, we consider for now $v$ being such that $2\mu = \mathcal{L}_{\mathbf{S}}^{\frac{\gamma'+\gamma}{2}}(v) + \mathcal{L}_{\mathbf{S}'}^{\frac{\gamma'+\gamma}{2}}(v) \geq \tau + c/2\left(\sqrt{\frac{\tau\ln(e/\delta)}{m}} + \frac{\ln(e/\delta)}{m}\right)$. We notice that $\mu$ is the expectation of $\mathcal{L}_{\mathbf{S}}^{\frac{\gamma'+\gamma}{2}}(v)$. Furthermore, since $\mathcal{L}_{\mathbf{S}}^{\frac{\gamma'+\gamma}{2}}$ is samples without replacement from $[\mathbb{1}\{v(x)y \leq \frac{\gamma'+\gamma}{2}\}]_{(x,y)\in Z}$ it follows by the multiplicative Chernoff bound without replacement [Hoeffding, 1963, Section 6] and $\mu \geq \frac{c\ln(e/\delta)}{4m} > \frac{2\ln(e/\delta)}{m}$ (since $c \geq 64$ ) that,

$$\mathbb{P}\left[\mathcal{L}_{\mathbf{S}}^{\frac{\gamma'+\gamma}{2}}(v) \leq \left(1 - \sqrt{\frac{2\ln(e/\delta)}{\mu m}}\right)\cdot\mu\right] \leq \frac{\delta}{e}.$$

Thus, we conclude that with probability at least $1 - \delta/e$ we have that,

$$\mathcal{L}_{\mathbf{S}}^{\frac{\gamma'+\gamma}{2}}(v) \geq \mu - \sqrt{\frac{2\mu\ln(e/\delta)}{m}} \tag{15}$$

which since $\mu = \left(\mathcal{L}_{\mathbf{S}}^{\frac{\gamma'+\gamma}{2}}(v) + \mathcal{L}_{\mathbf{S}'}^{\frac{\gamma'+\gamma}{2}}(v)\right)/2$ and that $\sqrt{a+b} \leq \sqrt{a} + \sqrt{b}$ gives that

$$\mathcal{L}_{\mathbf{S}}^{\frac{\gamma'+\gamma}{2}}(v) \geq \left(\mathcal{L}_{\mathbf{S}}^{\frac{\gamma'+\gamma}{2}}(v) + \mathcal{L}_{\mathbf{S}'}^{\frac{\gamma'+\gamma}{2}}(v)\right)/2 - \sqrt{\frac{1}{m}\mathcal{L}_{\mathbf{S}}^{\frac{\gamma'+\gamma}{2}}(v)\ln(e/\delta)} - \sqrt{\frac{1}{m}\mathcal{L}_{\mathbf{S}'}^{\frac{\gamma'+\gamma}{2}}(v)\ln(e/\delta)},$$

so

$$\mathcal{L}_{\mathbf{S}}^{\frac{\gamma'+\gamma}{2}}(v) + \sqrt{\frac{4}{m}\mathcal{L}_{\mathbf{S}}^{\frac{\gamma'+\gamma}{2}}(v)\ln(e/\delta)} \geq \mathcal{L}_{\mathbf{S}'}^{\frac{\gamma'+\gamma}{2}}(v) - \sqrt{\frac{4}{m}\mathcal{L}_{\mathbf{S}'}^{\frac{\gamma'+\gamma}{2}}(v)\ln(e/\delta)}.$$

We now show that for outcomes of $\mathbf{S}$ and $\mathbf{S}'$ such that $\mathcal{L}_{\mathbf{S}}^{\frac{\gamma'+\gamma}{2}}(v) \leq \tau$ and $\mathcal{L}_{\mathbf{S}'}^{\frac{\gamma'+\gamma}{2}}(v) \geq \tau + c/2\left(\sqrt{\frac{\tau\ln(e/\delta)}{m}} + \frac{\ln(e/\delta)}{m}\right)$ it holds that

$$\mathcal{L}_{\mathbf{S}}^{\frac{\gamma'+\gamma}{2}}(v) + \sqrt{\frac{4}{m}\mathcal{L}_{\mathbf{S}}^{\frac{\gamma'+\gamma}{2}}(v)\ln(e/\delta)} < \mathcal{L}_{\mathbf{S}'}^{\frac{\gamma'+\gamma}{2}}(v) - \sqrt{\frac{4}{m}\mathcal{L}_{\mathbf{S}'}^{\frac{\gamma'+\gamma}{2}}(v)\ln(e/\delta)},$$

which combined with the conclusion below Eq. (15) implies that $\mathbf{S}$ and $\mathbf{S}'$ such that $\mathcal{L}_{\mathbf{S}}^{\frac{\gamma'+\gamma}{2}}(v) \leq \tau$ and $\mathcal{L}_{\mathbf{S}'}^{\frac{\gamma'+\gamma}{2}}(v) \geq \tau + c/2\left(\sqrt{\frac{\tau\ln(e/\delta)}{m}} + \frac{\ln(e/\delta)}{m}\right)$ happens with probability at most $\delta/e$ concluding the proof.

Thus consider outcomes $S, S'$ of $\mathbf{S}$ and $\mathbf{S}'$ such that $\mathcal{L}_{S}^{\frac{\gamma'+\gamma}{2}}(v) \leq \tau$ and $\mathcal{L}_{S'}^{\frac{\gamma'+\gamma}{2}}(v) \geq \tau + c/2\left(\sqrt{\frac{\tau\ln(e/\delta)}{m}} + \frac{\ln(e/\delta)}{m}\right)$. We first notice that since for $a > 0$, $x + \sqrt{ax}$ is increasing in $x$ we have that

$$\mathcal{L}_{S}^{\frac{\gamma'+\gamma}{2}}(v) + \sqrt{\frac{4}{m}\mathcal{L}_{S}^{\frac{\gamma'+\gamma}{2}}(v)\ln(e/\delta)} \leq \tau + \sqrt{\frac{4}{m}\tau\ln(e/\delta)}. \tag{16}$$

Furthermore, since by Observation C.4 we have that $x - \sqrt{ax}$ is increasing for $x \geq a/4$, which $x = \mathcal{L}_{\mathbf{S}'}^{\frac{\gamma'+\gamma}{2}}(v)$ and $a = \frac{4\ln(e/\delta)}{m}$ and $\mathcal{L}_{\mathbf{S}'}^{\frac{\gamma'+\gamma}{2}}(v) \geq \tau + c/2\left(\sqrt{\frac{\tau \ln(e/\delta)}{m}} + \frac{\ln(e/\delta)}{m}\right)$, $c \geq 32$ we conclude that

$$\mathcal{L}_{\mathbf{S}'}^{\frac{\gamma'+\gamma}{2}}(v) - \sqrt{\frac{4}{m}\mathcal{L}_{\mathbf{S}'}^{\frac{\gamma'+\gamma}{2}}(v)\ln(e/\delta)}$$
$$\geq \tau + c/2\left(\sqrt{\frac{\tau\ln(e/\delta)}{m}} + \frac{\ln(e/\delta)}{m}\right) - \sqrt{\frac{4}{m}\left(\tau + c/2\left(\sqrt{\frac{\tau\ln(e/\delta)}{m}} + \frac{\ln(e/\delta)}{m}\right)\right)\ln(e/\delta)}. \tag{17}$$

Using that $c/2 \geq 1$ and that $a + b + \sqrt{ab} \leq 2(a+b)$ for $a, b > 0$ and that $\sqrt{a+b} \leq \sqrt{a} + \sqrt{b}$ we get that the last term in the above can be upper bounded by.

$$\sqrt{\frac{4}{m}\left(\tau + \frac{c}{2}\left(\sqrt{\frac{\tau}{m}\ln\left(\frac{e}{\delta}\right)} + \frac{\ln(e/\delta)}{m}\right)\right)\ln\left(\frac{e}{\delta}\right)} \leq \sqrt{\frac{2c}{m}\left(\tau + \sqrt{\frac{\tau\ln(e/\delta)}{m}} + \frac{\ln(e/\delta)}{m}\right)\ln\left(\frac{e}{\delta}\right)}$$
$$\leq \sqrt{\frac{4c}{m}\left(\tau + \frac{\ln(e/\delta)}{m}\right)\ln\left(\frac{e}{\delta}\right)}$$
$$\leq 2\sqrt{c}\left(\sqrt{\frac{\tau\ln(e/\delta)}{m}} + \frac{\ln(e/\delta)}{m}\right).$$

Thus, plugging back into Eq. (17) we conclude that

$$\mathcal{L}_{\mathbf{S}'}^{\frac{\gamma'+\gamma}{2}}(v) - \sqrt{\frac{4}{m}\mathcal{L}_{\mathbf{S}'}^{\frac{\gamma'+\gamma}{2}}(v)\ln\left(\frac{e}{\delta}\right)} \geq \tau + (c/2 - 2\sqrt{c})\left(\sqrt{\frac{\tau\ln(e/\delta)}{m}} + \frac{\ln(e/\delta)}{m}\right)$$
$$\geq \tau + c/4\left(\sqrt{\frac{\tau\ln(e/\delta)}{m}} + \frac{\ln(e/\delta)}{m}\right), \tag{18}$$

where the last inequality follows by $c/2 - 2\sqrt{c} \geq c/4$ since $c \geq 128$. Furthermore since $c/4 \geq 32$ we conclude by the above Eq. (18) and Eq. (16) that

$$\mathcal{L}_{\mathbf{S}}^{\frac{\gamma'+\gamma}{2}}(v) + \sqrt{\frac{4}{m}\mathcal{L}_{\mathbf{S}}^{\frac{\gamma'+\gamma}{2}}(v)\ln\left(\frac{e}{\delta}\right)} < \mathcal{L}_{\mathbf{S}'}^{\frac{\gamma'+\gamma}{2}}(v) - \sqrt{\frac{4}{m}\mathcal{L}_{\mathbf{S}'}^{\frac{\gamma'+\gamma}{2}}(v)\ln\left(\frac{e}{\delta}\right)}.$$

$\square$

We now move on to show Lemma C.3. For that, we need Rudelson and Vershynin [2006, Theorem 4.4] bounding the minimal infinity cover of a function class in terms of its fat shattering dimension

**Lemma C.5.** *There exists universal constants $C \geq 1$ and $c > 0$ such that: For a function class $\mathcal{F}$ and a point set $X = \{x_1, \ldots, x_m\}$ of size $m$, such that for any $v \in \mathcal{F}$ it holds that $\sum_{x \in X} |v(x)|/m \leq 1$. Then for $0 < \varepsilon < 1$, and $0 < \alpha < 1/2$ it holds that for $d = \mathrm{fat}_{c\varepsilon\alpha}(\mathcal{F})$ that*

$$\ln(|N_\infty(X, \mathcal{F}, \alpha)|) \leq Cd\ln\left(\frac{m}{d\alpha}\right)\ln^\varepsilon\left(\frac{2m}{d}\right).$$

Furthermore, to show Lemma C.3 we need the following lemma upper bounding the fat shattering dimension of convex combinations of a hypothesis class $\mathcal{H}$, truncated to $\lceil\gamma\rceil$, by the fat shattering dimension of the hypothesis class $\mathcal{H}$.

**Lemma C.6.** *There exists universal constants $C' \geq 1$ and $1 \geq c' > 0$ such that: For hypothesis class $\mathcal{H} \subseteq [-1,1]^{\mathcal{X}}$, $\gamma > 0$ and $\alpha > 0$ we have that*

$$\mathrm{fat}_\alpha(\mathrm{conv}(\mathcal{H})_{\lceil\gamma\rceil}) \leq \frac{C'\,\mathrm{fat}_{c'\alpha}(\mathcal{H})}{\alpha^2}.$$

We now show how the above two lemmas combined give Lemma C.3.

*Proof of Lemma C.3.* Let in the following $C \geq 1$ and $c > 0$ denote the universal constants of Lemma C.5, Furthermore let $C' > 1$ and $c' > 0$ denote the universal constants of Lemma C.6, and lastly let $\hat{c} = \frac{c'c}{4}$ and $\hat{C} = \max(1, \frac{16C'}{c^2})$.

We first consider the function class $\operatorname{conv}(\mathcal{H})_{\lceil 2\gamma' \rceil} / (2\gamma') = \{f' = f/(2\gamma') : f \in \operatorname{conv}(\mathcal{H})_{\lceil 2\gamma' \rceil}\}$, i.e., the functions in $\operatorname{conv}(\mathcal{H})_{\lceil 2\gamma' \rceil}$ scaled by $1/(2\gamma')$. We notice that the functions $v \in \operatorname{conv}(\mathcal{H})_{\lceil 2\gamma' \rceil} / (2\gamma')$, has absolute value at most 1, thus it if we consider a minimal $\frac{\gamma'-\gamma}{4\gamma'}$-cover in infinity norm of $\operatorname{conv}(\mathcal{H})_{\lceil 2\gamma' \rceil} / (2\gamma')$, denote it $N_\infty = N_\infty(X, \operatorname{conv}(\mathcal{H})_{\lceil 2\gamma' \rceil} / (2\gamma'), \frac{\gamma'-\gamma}{4\gamma'})$, i.e., for all $v \in \operatorname{conv}(\mathcal{H})_{\lceil 2\gamma' \rceil} / (2\gamma')$ there exists $\hat{v} \in N_\infty$ such that for

$$\max_{x \in X} |v(x) - \hat{v}(x)| \leq \frac{\gamma' - \gamma}{4\gamma'}.$$

and any other cover with this property has size less than or equal to $N_\infty$. We now notice that since for any $v \in \operatorname{conv}(\mathcal{H})_{\lceil 2\gamma' \rceil}$ we have that $v/(2\gamma') \in \operatorname{conv}(\mathcal{H})_{\lceil 2\gamma' \rceil} / (2\gamma')$, we have that there exists $\hat{v} \in N_\infty$ such that

$$\max_{x \in X} |v(x)/(2\gamma') - \hat{v}(x)| \leq \frac{\gamma' - \gamma}{4\gamma'}$$

which further implies that

$$\max_{x \in X} |v(x) - (2\gamma')\hat{v}(x)| \leq \frac{\gamma' - \gamma}{2},$$

whereby we conclude that $(2\gamma')N_\infty = \{v' = (2\gamma')v : v \in N_\infty\}$, the functions in $N_\infty$ scaled by $(2\gamma')$, is a $\frac{\gamma'-\gamma}{2}$- cover for $\operatorname{conv}(\mathcal{H})_{\lceil 2\gamma' \rceil}$. Thus, if we can bound that size of $N_\infty$ we also get an upper bound on the size of a minimal $\frac{\gamma'-\gamma}{2}$- cover for $\operatorname{conv}(\mathcal{H})_{\lceil 2\gamma' \rceil}$, where we denote such a minimal cover $N_\infty(X, \operatorname{conv}(\mathcal{H})_{\lceil 2\gamma' \rceil}, \frac{\gamma'-\gamma}{2})$.

We notice that $\operatorname{fat}_{\frac{c(\gamma'-\gamma)}{8\gamma'}} (\operatorname{conv}(\mathcal{H})_{\lceil 2\gamma' \rceil} / (2\gamma')) = \operatorname{fat}_{\frac{c(\gamma'-\gamma)}{4}} (\operatorname{conv}(\mathcal{H})_{\lceil 2\gamma' \rceil})$, where we have used that for scalars $a, b > 0$ and a function class $\mathcal{F}$ we have that $\operatorname{fat}_a(b \cdot \mathcal{F}) = \operatorname{fat}_{a/b}(\mathcal{F})$, where $b \cdot \mathcal{F}$ is the function class obtained from $\mathcal{F}$ by scaling all the functions in $\mathcal{F}$ by $b$. Furthermore by Lemma C.6 we have that $\operatorname{fat}_{\frac{c(\gamma'-\gamma)}{8\gamma'}} (\operatorname{conv}(\mathcal{H})_{\lceil 2\gamma' \rceil} / (2\gamma')) = \operatorname{fat}_{\frac{c(\gamma'-\gamma)}{4}} (\operatorname{conv}(\mathcal{H})_{\lceil 2\gamma' \rceil}) \leq$

$\frac{C' 16 \operatorname{fat}_{\frac{c'c(\gamma'-\gamma)}{4}} (\mathcal{H})}{c^2 (\gamma'-\gamma)^2} \leq \frac{\hat{C} \operatorname{fat}_{\hat{c}(\gamma'-\gamma)}(\mathcal{H})}{(\gamma'-\gamma)^2} \leq m$, by $m \geq \frac{\hat{C} \operatorname{fat}_{\hat{c}(\gamma'-\gamma)}(\mathcal{H})}{(\gamma'-\gamma)^2}$, $\hat{c} = \frac{c'c}{4}$ and $\hat{C} = \max(1, \frac{16C'}{c^2})$ i.e., $1 \leq m/\operatorname{fat}_{\frac{c(\gamma'-\gamma)}{8\gamma'}} (\operatorname{conv}(\mathcal{H})_{\lceil 2\gamma' \rceil} / (2\gamma'))$ and we may thus invoke Lemma C.5 with the function class $\operatorname{conv}(\mathcal{H})_{\lceil 2\gamma' \rceil} / (2\gamma')$, $\varepsilon = 1/2$ and $\alpha = \frac{\gamma'-\gamma}{4\gamma'}$ (which is less than 1/2) to get that

$$\ln\left( |N_\infty(X, \operatorname{conv}(\mathcal{H})_{\lceil 2\gamma' \rceil}, \frac{\gamma'-\gamma}{2})| \right)$$

$$\leq \ln\left( |N_\infty(X, \operatorname{conv}(\mathcal{H})_{\lceil 2\gamma' \rceil} / (2\gamma'), \frac{\gamma'-\gamma}{4\gamma'})| \right)$$

$$\leq C \operatorname{fat}_{\frac{c(\gamma'-\gamma)}{8\gamma'}} (\operatorname{conv}(\mathcal{H})_{\lceil 2\gamma' \rceil} / (2\gamma')) \ln\left( \frac{m}{\operatorname{fat}_{\frac{c(\gamma'-\gamma)}{8\gamma'}} (\operatorname{conv}(\mathcal{H})_{\lceil 2\gamma' \rceil} / (2\gamma'))} \frac{4\gamma'}{\gamma'-\gamma} \right)$$

$$\cdot \ln^{1/2}\left( \frac{2m}{\operatorname{fat}_{\frac{c(\gamma'-\gamma)}{8\gamma'}} (\operatorname{conv}(\mathcal{H})_{\lceil 2\gamma' \rceil} / (2\gamma'))} \right)$$

$$\leq C \operatorname{fat}_{\frac{c(\gamma'-\gamma)}{8\gamma'}} (\operatorname{conv}(\mathcal{H})_{\lceil 2\gamma' \rceil} / (2\gamma')) \ln^{3/2}\left( \frac{m}{\operatorname{fat}_{\frac{c(\gamma'-\gamma)}{8\gamma'}} (\operatorname{conv}(\mathcal{H})_{\lceil 2\gamma' \rceil} / (2\gamma'))} \frac{4\gamma'}{\gamma'-\gamma} \right)$$

$$\leq C \operatorname{fat}_{\frac{c(\gamma'-\gamma)}{8\gamma'}} (\operatorname{conv}(\mathcal{H})_{\lceil 2\gamma' \rceil} / (2\gamma')) \ln^{3/2}\left( \frac{m}{\operatorname{fat}_{\frac{c(\gamma'-\gamma)}{8\gamma'}} (\operatorname{conv}(\mathcal{H})_{\lceil 2\gamma' \rceil} / (2\gamma'))} \frac{8\gamma'}{\gamma'-\gamma} \right)$$

where we in the second to last inequality have used that $\frac{4\gamma'}{\gamma'-\gamma} \geq 2$, and in the last we make an upper bound need in the following to argue for the monotonicity of a function. To this end consider a number $a > 0$ and the function $f(x) = x \ln^{3/2}(a/x)$, for $a/e > x$. We notice that $f$ has derivative $f'(x) = \frac{1}{2}\sqrt{\ln\left(\frac{a}{x}\right)}(2\ln\left(\frac{a}{x}\right) - 3)$, which is non-negative when $2\ln\left(\frac{a}{x}\right) - 3 > 0$ our equivalently $\frac{a}{\exp\left(\frac{3}{2}\right)} > x$, thus increasing for such values. Now consider $a = m\frac{8\gamma'}{\gamma'-\gamma}$. We have that $m \geq \frac{\hat{C}\operatorname{fat}_{\hat{c}(\gamma'-\gamma)}(\mathcal{H})}{(\gamma'-\gamma)^2} \geq \operatorname{fat}_{\frac{c(\gamma'-\gamma)}{8\gamma'}}(\operatorname{conv}(\mathcal{H})_{\lceil 2\gamma'\rceil}/(2\gamma'))$, implying that $m\frac{8\gamma'}{\exp\left(\frac{3}{2}\right)(\gamma'-\gamma)} > \frac{\hat{C}\operatorname{fat}_{\hat{c}(\gamma'-\gamma)}(\mathcal{H})}{(\gamma'-\gamma)^2} \geq \operatorname{fat}_{\frac{c(\gamma'-\gamma)}{8\gamma'}}(\operatorname{conv}(\mathcal{H})_{\lceil 2\gamma'\rceil}/(2\gamma'))$, since $\frac{8\gamma'}{\gamma'-\gamma} > \exp\left(\frac{3}{2}\right)$. Thus using this observation, with the above argued monotonicity of $x\ln^{3/2}(a/x)$ for $\frac{a}{\exp\left(\frac{3}{2}\right)} > x$, where $a = m\frac{8\gamma'}{\gamma'-\gamma}$ and $x = \operatorname{fat}_{\frac{c(\gamma'-\gamma)}{8\gamma'}}(\operatorname{conv}(\mathcal{H})_{\lceil 2\gamma'\rceil}/(2\gamma'))$ we conclude that

$$\ln\left(|N_\infty(X, \operatorname{conv}(\mathcal{H})_{\lceil 2\gamma'\rceil}, \frac{\gamma'-\gamma}{2})|\right)$$

$$\leq C\operatorname{fat}_{\frac{c(\gamma'-\gamma)}{8\gamma'}}(\operatorname{conv}(\mathcal{H})_{\lceil 2\gamma'\rceil}/(2\gamma'))\ln^{3/2}\left(\frac{m}{\operatorname{fat}_{\frac{c(\gamma'-\gamma)}{8\gamma'}}(\operatorname{conv}(\mathcal{H})_{\lceil 2\gamma'\rceil}/(2\gamma'))}\frac{8\gamma'}{\gamma'-\gamma}\right)$$

$$\leq \frac{C\hat{C}\operatorname{fat}_{\hat{c}(\gamma'-\gamma)}(\mathcal{H})}{(\gamma'-\gamma)^2}\ln^{3/2}\left(\frac{(\gamma'-\gamma)^2 m}{\hat{C}\operatorname{fat}_{\hat{c}(\gamma'-\gamma)}(\mathcal{H})}\frac{8\gamma'}{\gamma'-\gamma}\right).$$

$\square$

To prove Lemma C.6, we will drive a lower and a upper bound on the Rademacher complexity in terms of the fat shattering dimension of $\operatorname{conv}(\mathcal{H})$ and $\mathcal{H}$ . From this relation we can bound the fat shattering dimension of $\operatorname{conv}(\mathcal{H})$ in terms of $\mathcal{H}$. To the end of showing the upper bound we need the following two lemmas. The first results gives a bound on the Rademacher complexity of a function class $\mathcal{F}$ in terms the size of a minimal $\varepsilon$-cover of $\mathcal{F} \subset \mathbb{R}^{\mathcal{X}}$, over a point set $S = \{x_1, \ldots, x_m\}$ in $L_2$. To this end we let $N_2(S, \mathcal{F}, \varepsilon)$, denote the size of the smallest set of functions $N \subseteq \mathbb{R}^{\mathcal{X}}$ with the property that for any $f \in \mathcal{F}$, there exists $f' \in N$ such that $\sqrt{\sum_{i=1}^m (f(x_i) - f'(x_i))^2/m} \leq \varepsilon$.

**Lemma C.7** (Dudley's Entropy Integral Bound. E.g., Rebeschini [2021, Proposition 5.3]). *Let $\mathcal{F}$ be a class of real-valued functions, $S = \{x_1, \ldots, x_m\}$ be a point set of $m$ points, and $N_2(S, \mathcal{F}, \varepsilon)$ be the size of minimal $\epsilon$-cover of $\mathcal{F}$. Assuming $\sup_{f\in\mathcal{F}}\left(\frac{1}{m}\sum_{i=1}^m f^2(x_i)\right)^{1/2} \leq c$, then we have*

$$\mathbb{E}_{\boldsymbol{\sigma}\sim\{\pm1\}}\left[\frac{1}{m}\sup_{f\in\mathcal{F}}\sum_{i=1}^m \boldsymbol{\sigma}_i f(x_i)\right] \leq \inf_{\varepsilon\in[0,c/2]}\left(4\varepsilon + \frac{12}{\sqrt{m}}\int_\varepsilon^{c/2}\sqrt{\ln\left(|N_2(S, \mathcal{F}, \nu)|\right)}\,d\nu\right),$$

From the above lemma we see that having a bound on $N_2(S, \mathcal{F}, \varepsilon)$, implies an upper bound on the Rademacher complexity. To the end of bounding $N_2(S, \mathcal{F}, \varepsilon)$, we present the following lemma (it is a special case of Rudelson and Vershynin [2006, Corollary 5.4] with for instance $p = 2$ and $q = 3$.).

**Lemma C.8.** *Let $\mathcal{F}$ be a hypothesis set bounded in absolute value by 1. Let $S = \{x_1, \ldots, x_m\}$ be a set of $m$ points. There exists universal constants $C > 0$ and $0 < c \leq 1$ such that for any $0 < \epsilon < 1/2$, we have that*

$$\ln\left(|N_2(S, \mathcal{F}, \varepsilon)|\right) \leq C\operatorname{fat}_{c\varepsilon}(\mathcal{F})\ln\left(1/(c\varepsilon)\right).$$

We now combine the above lemmas to derive an upper bound on the Rademacher complexity in terms of the fat shattering dimension of $\operatorname{conv}(\mathcal{H})$ and $\mathcal{H}$. Furthermore, using the definition of fat shattering dimension we also derive and lower bound on the Rademacher complexity. Solving for the fat shattering dimension of $\operatorname{conv}(\mathcal{H})$ in this relation give the claim of Lemma C.6.

*Proof of Lemma C.6.*
**Shattering of $\operatorname{conv}(\mathcal{H})_{\lceil\gamma\rceil}$ implies shattering of $\operatorname{conv}(\mathcal{H})$:** We first recall the definition of

$\mathrm{conv}(\mathcal{H})_{\lceil\gamma\rceil} = \{v_{\lceil\gamma\rceil} : v \in \mathrm{conv}(\mathcal{H})\}$, where the operation $(\cdot)_{\lceil\gamma\rceil}$ was

$$v_{\lceil\gamma\rceil}(x) = \begin{cases} \gamma \text{ if } v(x) \geq \gamma \\ v(x) \text{ if } -\gamma < v(x) < \gamma \\ -\gamma \text{ if } v(x) \leq -\gamma \end{cases}$$

We notice that by this definition we always have that $\gamma > v_{\lceil\gamma\rceil}(x)$ implies $v_{\lceil\gamma\rceil} \geq v(x)$ and $-\gamma < v_{\lceil\gamma\rceil}(x)$ implies $v_{\lceil\gamma\rceil} \leq v(x)$.

Now consider a sequence of points $x_1, \ldots, x_d$ and levels $r_1, \ldots, r_d$ which is $\alpha$ shattered by $\mathrm{conv}(\mathcal{H})_{\lceil\gamma\rceil}$, i.e., we have that for any $b \in \{\pm 1\}^d$, there exists $v_{\lceil\gamma\rceil} \in \mathrm{conv}(\mathcal{H})_{\lceil\gamma\rceil}$, where $v \in \mathrm{conv}(\mathcal{H})$, such that for $i \in [d]$ it holds that

$$v_{\lceil\gamma\rceil}(x_i) \geq r_i + \alpha \text{ if } b_i = 1$$
$$v_{\lceil\gamma\rceil}(x_i) \leq r_i - \alpha \text{ if } b_i = -1.$$

We notice that since $v_{\lceil\gamma\rceil}$ only attains values in $[-\gamma, \gamma]$ it must be the case that $\alpha \leq \gamma$ for $d$ not to be 0(and $\alpha \leq 1$ since $\mathrm{conv}(\mathcal{H})_{\lceil\gamma\rceil}$ is bounded in absolute value by 1) where by the claim holds. Thus, we assume from now on that $\alpha \leq \gamma$ and $\alpha \leq 1$ We further notice, again by $v_{\lceil\gamma\rceil}$ attaining values in $[-\gamma, \gamma]$, that it must be the case that $r_i \in [\alpha - \gamma, \gamma - \alpha]$, stated equivalently that $-\gamma \leq r_i - \alpha, r_i + \alpha \leq \gamma$, otherwise no function $v_{\lceil\gamma\rceil} \in \mathrm{conv}(\mathcal{H})_{\lceil\gamma\rceil}$ in can either be $\alpha$ above or $\alpha$ below $r_i$ since in this case either $r_i - \alpha < -\gamma$ or $r_i + \alpha > \gamma$.

Now for $r_i \in [\alpha - \gamma, \gamma - \alpha]$ we notice that have that $-\gamma < r_i + \alpha$ and that $\gamma > r_i - \alpha$ thus since we earlier conclude that $\gamma > v_{\lceil\gamma\rceil}(x)$ implies $v_{\lceil\gamma\rceil} \geq v(x)$ and $-\gamma < v_{\lceil\gamma\rceil}(x)$ implies $v_{\lceil\gamma\rceil} \leq v(x)$, we get that if $r_i - \alpha \geq v_{\lceil\gamma\rceil}(x_i)$ then we also have that $r_i - \alpha \geq v(x_i)$, and if $r_i + \alpha \leq v_{\lceil\gamma\rceil}(x)$ then we also have that $r_i + \alpha \leq v(x_i)$. This shows, by $x_1, \ldots, x_d$ and $r_1, \ldots, r_d$, $\alpha$ shattering $\mathrm{conv}(\mathcal{H})_{\lceil\gamma\rceil}$, that $x_1, \ldots, x_d$ and $r_1, \ldots, r_d$, is also $\alpha$-shattering $\mathrm{conv}(\mathcal{H})$.

**Bounds on the Rademacher complexity of** $\mathrm{conv}(\mathcal{H})$ **in terms of** $\mathrm{fat}(\mathcal{H})$**,** $d$**, and** $\alpha$**:** Since $\mathrm{conv}(\mathcal{H})$ is $\alpha$-shattered by $x_1, \ldots, x_d$ and $r_1, \ldots, r_d$ this implies that for any $b \in \{\pm 1\}$ we have that there exists $v \in \mathrm{conv}(\mathcal{H})$ such that $b_i(v(x_i) - r_i) \geq \alpha$. Thus, we conclude by the expectation of $\mathbb{E}_{\boldsymbol{\sigma} \sim \{\pm 1\}} \left[ \sum_{i=1}^{d} \boldsymbol{\sigma}_i r_i \right] = 0$ that the Rademacher complexity of $\mathrm{conv}(\mathcal{H})$ on $x_1, \ldots, x_d$ can be lower bounded as follows

$$\mathbb{E}_{\boldsymbol{\sigma} \sim \{\pm 1\}} \left[ \sup_{v \in \mathrm{conv}(\mathcal{H})} \sum_{i=1}^{d} \boldsymbol{\sigma}_i v(x_i)/d \right] = \mathbb{E}_{\boldsymbol{\sigma} \sim \{\pm 1\}} \left[ \sup_{v \in \mathrm{conv}(\mathcal{H})} \sum_{i=1}^{d} \boldsymbol{\sigma}_i (v(x_i) - r_i)/d \right] \geq \alpha \quad (19)$$

Furthermore, we notice that the Rademacher complexity of $\mathrm{conv}(\mathcal{H})$, is the same as the Rademacher complexity of $\mathcal{H}$, since $\mathrm{conv}(\mathcal{H})$ are convex combinations of hypothesis in $\mathcal{H}$. To see this consider a realization $\sigma$ of $\boldsymbol{\sigma}$, then for any $v \in \mathrm{conv}(\mathcal{H})$, which can be written as $v = \sum_{h \in \mathcal{H}} \alpha_h h$ where $\sum_{h \in \mathcal{H}} \alpha_h = 1$ and $\alpha_h \geq 0$ we have that $\sum_{i=1}^{d} \sigma_i v(x_i) = \sum_{h \in \mathcal{H}} \alpha_h \sum_{i=1}^{d} \sigma_i h(x_i) \leq \sup_{h \in \mathcal{H}} \sum_{i=1}^{d} \sigma_i h(x_i)$, where the last inequality follows by $\sum_{h \in \mathcal{H}} \alpha_h = 1$. The opposite direction of the inequality follows from $\mathcal{H} \subseteq \mathrm{conv}(\mathcal{H})$ thus we have that

$$\mathbb{E}_{\boldsymbol{\sigma} \sim \{\pm 1\}} \left[ \sup_{v \in \mathrm{conv}(\mathcal{H})} \sum_{i=1}^{d} \boldsymbol{\sigma}_i v(x_i) \right] = \mathbb{E}_{\boldsymbol{\sigma} \sim \{\pm 1\}} \left[ \sup_{v \in \mathcal{H}} \sum_{i=1}^{d} \boldsymbol{\sigma}_i v(x_i) \right]. \quad (20)$$

Now since $v \in \mathrm{conv}(\mathcal{H})$ is bounded in absolute value by 1 it follows by Applying Lemma C.7 yields

$$\mathbb{E}_{\boldsymbol{\sigma} \sim \{\pm 1\}} \left[ \sup_{v \in \mathcal{H}} \sum_{i=1}^{d} \boldsymbol{\sigma}_i v(x_i) \right] \leq \inf_{\varepsilon \in [0, 1/2]} \left( 4\varepsilon + \frac{12}{\sqrt{d}} \int_{\varepsilon}^{1/2} \sqrt{\ln\left(|N_2(X, \mathcal{H}, \nu)|\right)} \, d\nu \right), \quad (21)$$

where $|N_2(X, \mathcal{H}, \nu)|$ is the size of a minimal $||\cdot||_2$-cover of $\mathcal{H}$ on $X$ that is, for any $h \in \mathcal{H}$, there exists an $\hat{h} \in N_2(X, \mathcal{H}, \nu)$ such that $\sqrt{\sum_{i=1}^{d}(h(x_i) - \hat{h}(x_i))^2/d} \leq \nu$. Now applying Lemma C.8 yields $\ln\left(|N_2(X, \mathcal{H}, \varepsilon)|\right) \leq C \mathrm{fat}_{c\varepsilon}(\mathcal{H}) \ln\left(1/(c\varepsilon)\right)$ for universal constants $C \geq 1$ and $1 \geq c > 0$.

Now setting $\varepsilon = \alpha/8$ in Eq. (21) (recall we are in the case that $\alpha \le 1$) and plugging in the above bound on $\ln\left(|N_2(X, \mathcal{H}, \varepsilon)|\right)$

$$\mathbb{E}_{\boldsymbol{\sigma} \sim \{\pm 1\}}\left[\sup_{v \in \mathcal{H}} \sum_{i=1}^{d} \boldsymbol{\sigma}_i v(x_i)\right] \le \alpha/2 + \frac{12}{\sqrt{d}} \int_{\alpha/8}^{1/2} \sqrt{C \operatorname{fat}_{c\varepsilon}(\mathcal{H}) \ln\left(1/(c\varepsilon)\right)}\, d\varepsilon,$$

$$\le \alpha/2 + \frac{12\sqrt{C \operatorname{fat}_{c\alpha/8}(\mathcal{H})}}{c\sqrt{d}} \int_{c\alpha/8}^{c/2} \sqrt{\ln\left(1/\varepsilon'\right)}\, d\varepsilon'$$

$$\le \alpha/2 + \frac{12\sqrt{C \operatorname{fat}_{c\alpha/8}(\mathcal{H})}}{c\sqrt{d}} \tag{22}$$

where the second inequality follows from integration by substitution with $c\varepsilon = \varepsilon'$, and the last by $\int_0^1 \sqrt{\ln\left(1/\varepsilon'\right)}\, d\varepsilon' \le 1$. Now combining the upper bound of Eq. (22), lower bound of Eq. (19) and the relation Eq. (20) we get that

$$\alpha \le \alpha/2 + \frac{12\sqrt{C \operatorname{fat}_{c\alpha/8}(\mathcal{H})}}{c\sqrt{d}}$$

which implies that

$$d \le \frac{24^2 C \operatorname{fat}_{c\alpha/8}(\mathcal{H})}{c^2 \alpha^2}.$$

Thus, we conclude that $\operatorname{conv}(\mathcal{H})_{\lceil \gamma \rceil}$ can not be $\alpha$-shattered by a point set $X = \{x_1, \ldots, x_d\}$ and level sets $r_1, \ldots, r_d$ of more than $\frac{24^2 C \operatorname{fat}_{c\alpha/8}(\mathcal{H})}{c^2 \alpha^2}$ points, and we can conclude that $\operatorname{fat}_\alpha(\operatorname{conv}(\mathcal{H})_{\lceil \gamma \rceil}) \le \frac{24^2 C \operatorname{fat}_{c\alpha/8}(\mathcal{H})}{c^2 \alpha^2}$ and setting $C' = \max\left(1, \frac{24^2 C}{c^2}\right)$ and $c' = c/8$ this concludes the proof. $\square$

## D Agnostic boosting proof

In this section we give the proof of Theorem 3.4, which impies Theorem 1.3. To keep notation concise in the following we will let $\theta = \gamma - \varepsilon_0$.

**Lemma D.1** (Shalev-Shwartz and Ben-David [2014, Lemma B.10])**.** *Let $v \in [-1, 1]^{\mathcal{X}}$, $\gamma' \in [-1, 1]$, $\delta \in (0, 1)$, $m \in \mathbb{N}$, and $\mathcal{D} \in \Delta(\mathcal{X} \times \{\pm 1\})$. Then,*

$$\mathbb{P}_{\boldsymbol{S}}\left[\mathcal{L}_{\mathcal{D}}^{\gamma'}(v) \le \mathcal{L}_{\boldsymbol{S}}^{\gamma'}(v) + \sqrt{\frac{2\,\mathcal{L}_{\boldsymbol{S}}^{\gamma'}(v) \ln(1/\delta)}{m}} + \frac{4 \ln(1/\delta)}{m}\right] \ge 1 - \delta,$$

*and*

$$\mathbb{P}_{\boldsymbol{S}}\left[\mathcal{L}_{\boldsymbol{S}}^{\gamma'}(v) \le \mathcal{L}_{\mathcal{D}}^{\gamma'}(v) + \sqrt{\frac{2\,\mathcal{L}_{\mathcal{D}}^{\gamma'}(v) \ln(1/\delta)}{3m}} + \frac{2 \ln(1/\delta)}{m}\right] \ge 1 - \delta.$$

**Lemma D.2.** *There exists universal constants $C' \ge 1$ and $\hat{c}$ such that: Letting $d = \operatorname{fat}_{\hat{c}(\gamma-\varepsilon_0)/16}$, after the **for** loop starting at Line 3, with probability at least $1 - \frac{2\delta}{10}$ over $\mathbf{S}_1$ and randomness used in Algorithm 1, $\mathcal{B}_1$ contains a voting classifier $v_g$ such that*

$$\mathcal{L}_{\mathcal{D}_{f^\star}}^{\theta/16}(v_g) \le \frac{3C'}{m} \cdot \left[\frac{16^2 d}{\theta^2} \cdot \operatorname{Ln}^{3/2}\left(\frac{2\theta^2 m}{3 \cdot 16^2 d}\right) + \ln \frac{10e}{\delta}\right]. \tag{23}$$

*Proof.* We first notice that we may assume that

$$m \ge 3C' \ln \frac{10e}{\delta} \quad \text{and} \quad m \ge \frac{3 \cdot 16^2 C' d}{\theta^2} \tag{24}$$

as otherwise the right hand side of Eq. (23) is greater than 1 and the result follows by noting that $\mathcal{L}_{\mathcal{D}_{f^\star}}^{\theta/16}(v_g) \le 1$.

Without loss of generality, let

$$f^\star = \arg\max_{f \in \mathcal{F}} \operatorname{corr}_{\mathcal{D}}(f).$$

To the end of making an observation about Algorithm 2 let $S_1$ be a realization of $\mathbf{S}_1$. The execution of Algorithm 2 runs Algorithm 1 with all possible labelings of $S_1 = ((x_1, y_1), \ldots, (x_{m/3}, y_{m/3}))$. In particular, letting

$$S_{1,f^\star} := \big((x_i, f^\star(x_i))\big)_{i=1}^{m/3},$$

and denoting Algorithm 1 by $\mathcal{A}$, it must be the case that $\mathcal{A}(S_{1,f^\star}) \in \mathcal{B}_1$. Moreover, as $f^\star$ correctly classifies all points in $S_{1,f^\star}$, and $f^\star \in \mathcal{F}$, we have that $\sup_{f \in \mathcal{F}} \operatorname{corr}_{\mathcal{D}'}(f) = 1$ for any distribution $\mathcal{D}'$ over $S_{1,f^\star}$. Therefore, for such $\mathcal{D}'$, the weak-learning guarantee becomes that with probability at least $1 - \delta_0$ over $\mathbf{S}' \sim (\mathcal{D}')^{m_0}$ it holds that

$$\operatorname{corr}_{\mathcal{D}'}\big(\mathcal{W}(\mathbf{S}')\big) \geq \gamma \sup_{f \in \mathcal{F}} \operatorname{corr}_{\mathcal{D}'}(f) - \varepsilon_0$$

$$= \gamma - \varepsilon_0 = \theta.$$

Accordingly, to leverage Lemma 3.1, let

$$E_1(S_{1,f^\star}) := \Big\{ \mathcal{L}^{\theta/8}_{S_{1,f^\star}}(\mathcal{A}(S_{1,f^\star})) = 0 \Big\},$$

be an event over the randomness used in Algorithm 1, and notice that Line 6 runs $\mathcal{A}$ on $S_{1,f^\star}$ for

$$T = \lceil 32 m \ln(em) \rceil$$

$$\geq \left\lceil \frac{32 \ln(em)}{\theta^2} \right\rceil \qquad \text{(by Eq. (24))}$$

with $k = \lceil 8 \ln(10 e T/\delta)/(1 - \delta_0) \rceil$. Thus, applying Lemma 3.1 with $\gamma' = \theta$ ensures that

$$\mathbb{P}[E_1(S_{1,f^\star})] \geq 1 - \frac{\delta}{10}.$$

Notice we showed the above for any realization $S_1$ of $\mathbf{S}_1$. Let in the following $E_1 = E_1(\mathbf{S}_{1,f^\star})$.

Invoking Lemma 3.2 with $\mathbf{S}_{1,f^\star} \sim \mathcal{D}_{f^\star}^{m/3}$ and margin levels $\theta/16 < \theta/8$ ensures that, with probability at least $1 - \delta/10$, the event (over $\mathbf{S}_{1,f^\star}$)

$$E_2 := \left\{ \forall v \in \operatorname{conv}(\mathcal{H}) : \mathcal{L}^{\theta/16}_{\mathcal{D}_{f^\star}}(v) \leq \mathcal{L}^{\theta/8}_{\mathbf{S}_{1,f^\star}}(v) + C'\left( \sqrt{\mathcal{L}^{\theta/8}_{\mathbf{S}_{1,f^\star}}(v) \cdot \beta_{m/3,\theta}} + \beta_{m/3,\theta} \right) \right\}$$

holds, where $d = \operatorname{fat}_{\hat{c}\theta/16}(\mathcal{H})$ and

$$\beta_{n,\lambda} := \frac{1}{n} \cdot \left[ \frac{16^2 d}{\lambda^2} \cdot \operatorname{Ln}^{3/2}\left( \frac{2n\lambda^2}{16^2 d} \right) + \ln \frac{10e}{\delta} \right].$$

In the following, we will use $\mathbf{A}$ to refer to the randomness used in all calls to Algorithm 1 during the execution Algorithm 2. Namely, the randomness used in Line 6 to draw from $D_t$, where we assume all the $(2^{|\mathbf{S}_1|} - 1) \cdot k$ draws to be mutually independent.

Compiling the above, we have that

$$\mathbb{P}_{\mathbf{S}_1, \mathbf{A}}\Big[ \exists v \in \mathcal{B}_1 : \mathcal{L}^{\theta/16}_{\mathcal{D}_{f^\star}}(v) \leq C' \beta_{m/3,\theta} \Big]$$

$$\geq \mathbb{P}_{\mathbf{S}_1, \mathbf{A}}\Big[ \mathcal{L}^{\theta/16}_{\mathcal{D}_{f^\star}}(\mathcal{A}(\mathbf{S}_{1,f^\star})) \leq C' \beta_{m/3,\theta} \Big] \qquad \text{(as } \mathcal{A}(\mathbf{S}_{1,f^\star}) \in \mathcal{B}_1\text{)}$$

$$= \mathbb{E}_{\mathbf{S}_1}\Big[ \mathbb{E}_{\mathbf{A}}\Big[ \mathbb{1}\big\{ \mathcal{L}^{\theta/16}_{\mathcal{D}_{f^\star}}(\mathcal{A}(\mathbf{S}_{1,f^\star})) \leq C' \beta_{m/3,\theta} \big\} \Big] \Big] \qquad \text{(by independence of } \mathbf{S}_1 \text{ and } \mathcal{A}\text{)}$$

$$\geq \mathbb{E}_{\mathbf{S}_1}\Big[ \mathbb{E}_{\mathbf{A}}\Big[ \mathbb{1}\big\{ \mathcal{L}^{\theta/16}_{\mathcal{D}_{f^\star}}(\mathcal{A}(\mathbf{S}_{1,f^\star})) \leq C' \beta_{m/3,\theta} \big\} \cdot \mathbb{1}\{E_1\} \Big] \cdot \mathbb{1}\{E_2\} \Big]$$

$$\geq (1 - \delta/10)^2 \qquad (25)$$

$$\geq 1 - 2\delta/10, \qquad \text{(by Bernoulli's inequality)}$$

where Eq. (25) holds as the events $E_1$ and $E_2$ each hold with probability at least $1 - \delta/10$ and their simultaneous occurence implying that $\mathcal{L}^{\theta/16}_{\mathcal{D}_{f^\star}}(\mathcal{A}(\mathbf{S}_{1,f^\star})) \leq C' \beta_{m/3,\theta}$. $\qquad \square$

**Lemma D.3** (Restatement of 3.3). *There exists universal constants $C \geq 1$ and $\hat{c} > 0$ such that: Letting $\hat{d} = \mathrm{fat}_{\hat{c}(\gamma - \varepsilon_0)/32}(\mathcal{H})$, after the **for** loop starting at Line 8 of Algorithm 2, with probability at least $1 - \delta/2$ over $\mathbf{S}_1, \mathbf{S}_2$ and randomness used in Algorithm 1, that $\mathcal{B}_2$ contains a voting classifier $v_g$ such that*

$$\mathcal{L}_{\mathcal{D}}(v_g) \leq \mathrm{err}_{\mathcal{D}}(f^\star) + \sqrt{\frac{C\,\mathrm{err}_{\mathcal{D}}(f^\star)}{m} \cdot \left[\frac{\hat{d}}{\theta^2} \cdot \mathrm{Ln}^{3/2}\!\left(\frac{\theta^2 m}{\hat{d}}\right) + \ln\frac{10}{\delta}\right]}$$

$$+ \frac{C}{m}\left[\frac{\hat{d}}{\theta^2} \cdot \mathrm{Ln}^{3/2}\!\left(\frac{\theta^2 m}{\hat{d}}\right) + \ln\frac{10}{\delta}\right]. \tag{26}$$

*Proof.* It will be useful to consider the function $\zeta \colon \mathbb{R}_{>0} \to \mathbb{R}_{>0}$ given by

$$\zeta(x) = x^{-1} \cdot \ln^{3/2}\!\big(\max\{2x, e^2\}\big), \qquad \text{which is decreasing for any } x > 0.$$

To see this, let $f(x) = x^{-1} \ln^{3/2}(2x)$ for $x > 1/2$, so that $f'(x) = \frac{(3 - 2\ln 2x)\sqrt{\ln 2x}}{2x^2}$, thus $f(x)$ is decreasing for $x > \exp(3/2)/2$. As $1/x$ is decreasing for $x > 0$, we conclude that $x^{-1}\ln^{3/2}(\max\{2x, e^2\})$ is decreasing for $x > 0$. We shall also implicitly use that $\mathrm{Ln}(x) := \ln(\max\{x, e\}) \leq \ln(\max\{x, e^2\})$.

We will prove Eq. (26) for $C \geq 3072e^2$. Thus, we may assume that

$$m \geq \frac{3072e^2\hat{d}}{\theta^2}$$

as otherwise the right hand side of Eq. (26) is greater than 1 and the result follows trivially.

By Lemma D.2, with probability at least $1 - \frac{2\delta}{10}$ over $\mathbf{S}_1 \sim \mathcal{D}^{m/3}$ and the randomness used in Algorithm 1 there exists a voting classifier $\mathbf{v}_g \in \mathcal{B}_1$ such that

$$\mathcal{L}_{\mathcal{D}_{f^\star}}^{\theta/16}(\mathbf{v}_g) \leq C'\left[\zeta\!\left(\frac{\theta^2 m}{3 \cdot 16^2 d}\right) + \frac{3}{m}\ln\frac{10}{\delta}\right], \tag{27}$$

with $d = \mathrm{fat}_{\hat{c}\theta/16}$ and $C' \geq 1$. Since the fat-shattering dimension is decreasing in its level parameter, $\mathrm{fat}_{\hat{c}\theta/16} \leq \mathrm{fat}_{\hat{c}\theta/32} = \hat{d}$, thus, by the monotonic decrease of $\zeta$ and the above,

$$\mathcal{L}_{\mathcal{D}_{f^\star}}^{\theta/16}(\mathbf{v}_g) \leq C'\left[\zeta\!\left(\frac{\theta^2 m}{3 \cdot 16^2 \hat{d}}\right) + \frac{3}{m}\ln\frac{10}{\delta}\right]. \tag{28}$$

Consider a realization $S_1$ of $\mathbf{S}_1$ and the randomness of Algorithm 1 for which the above holds, and let $v_g \in \mathcal{B}_1$ be the associated classifier. Then, by Lemma D.1, with probability at least $1 - \frac{\delta}{10}$ over $\mathbf{S}_2$,

$$\mathcal{L}_{\mathbf{S}_2}^{\theta/16}(v_g) \leq \mathcal{L}_{\mathcal{D}}^{\theta/16}(v_g) + \sqrt{\frac{2\,\mathcal{L}_{\mathcal{D}}^{\theta/16}(v_g)}{m}\ln\frac{10}{\delta}} + \frac{6}{m}\ln\frac{10}{\delta}. \tag{29}$$

Let $i_g$ be the natural number such that $\theta/16 \in [2^{-i_g}, 2^{-i_g+1})$, and let $\gamma_g' = 2^{-i_g}$. To see that $\gamma_g' \in \{1, 1/2, 1/4, \ldots, 1/2^{\lceil \log_2(\sqrt{m})\rceil}\}$ so that it is considered in Line 6 of Algorithm 2, recall that we are in the case $m \geq 3072e^2 d/\theta^2$, thus $\theta/16 \geq \sqrt{24/m}$.

Letting $d' = \mathrm{fat}_{\hat{c}\gamma_g'}(\mathcal{H})$, Lemma 3.2 with sample $\mathbf{S}_2$ and margin levels 0 and $\gamma_g'$ ensures that, with probability at least $1 - \delta/10$ over $\mathbf{S}_2 \sim \mathcal{D}^{m/3}$ it holds that for all $v \in \mathrm{conv}(\mathcal{H})$

$$\mathcal{L}_{\mathcal{D}}(v) \leq \mathcal{L}_{\mathbf{S}_2}^{\gamma_g'}(v) + C'\left[\sqrt{\mathcal{L}_{\mathbf{S}_2}^{\gamma_g'}(v)\left[\zeta\!\left(\frac{(\gamma_g')^2 m}{3d'}\right) + \frac{3}{m}\ln\frac{10}{\delta}\right]} + \zeta\!\left(\frac{(\gamma_g')^2 m}{3d'}\right) + \frac{3}{m}\ln\frac{10}{\delta}\right].$$

Furthermore, the choice of $\gamma_g'$ implies that $\gamma_g' > \theta/32$, and by the fat-shattering dimension being decreasing in its level parameter implies that $d' = \mathrm{fat}_{\hat{c}\gamma_g'}(\mathcal{H}) \leq \mathrm{fat}_{\hat{c}\theta/32}(\mathcal{H}) = \hat{d}$. Applying this in

the inequality above, combined with the monotonic decrease of $\zeta$ yields that, with probabilities at least $1 - \delta/10$ over $\mathbf{S}_2$,

$$\mathcal{L}_{\mathcal{D}}(v) \le \mathcal{L}_{\mathbf{S}_2}^{\gamma_g'}(v) + C'\left[\sqrt{\mathcal{L}_{\mathbf{S}_2}^{\gamma_g'}(v)\left[\zeta\Big(\frac{\theta^2 m}{3 \cdot 32^2 \hat{d}}\Big) + \frac{3}{m}\ln\frac{10}{\delta}\right]} + \zeta\Big(\frac{\theta^2 m}{3 \cdot 32^2 \hat{d}}\Big) + \frac{3}{m}\ln\frac{10}{\delta}\right],$$

for all $v \in \operatorname{conv}(\mathcal{H})$. In particular, the above holds for $v_g' \coloneqq \arg\min_{v \in \mathcal{B}_1} \mathcal{L}_{\mathbf{S}_2}^{\gamma_g'}(v)$. With that, as $v_g \in \mathcal{B}_1$, we have that $\mathcal{L}_{\mathbf{S}_2}^{\gamma_g'}(v_g') \le \mathcal{L}_{\mathbf{S}_2}^{\gamma_g'}(v_g)$. Additionally, as $\gamma_g' \le \theta/16$, it must be that $\mathcal{L}_{\mathbf{S}_2}^{\gamma_g'}(v_g) \le \mathcal{L}_{\mathbf{S}_2}^{\theta/16}(v_g)$. Altogether, we obtain that with probability at least $1 - \delta/10$ over $\mathbf{S}_2$,

$$\mathcal{L}_{\mathcal{D}}(v_g') \le \mathcal{L}_{\mathbf{S}_2}^{\theta/16}(v_g) + C'\sqrt{\mathcal{L}_{\mathbf{S}_2}^{\theta/16}(v_g)\left[\zeta\Big(\frac{\theta^2 m}{3 \cdot 32^2 \hat{d}}\Big) + \frac{3}{m}\ln\frac{10}{\delta}\right]}$$
$$+ C'\left[\zeta\Big(\frac{\theta^2 m}{3 \cdot 32^2 \hat{d}}\Big) + \frac{3}{m}\ln\frac{10}{\delta}\right].$$

Using the union bound to also have Eq. (29) hold, we obtain that with probability at least $1 - 2\delta/10$ over $\mathbf{S}_2$,

$$\mathcal{L}_{\mathcal{D}}(v_g') \le \mathcal{L}_{\mathcal{D}}^{\theta/16}(v_g) + \sqrt{\frac{2\,\mathcal{L}_{\mathcal{D}}^{\theta/16}(v_g)}{m}\ln\frac{10}{\delta}} + \frac{6}{m}\ln\frac{10}{\delta} + C'\left[\zeta\Big(\frac{\theta^2 m}{3 \cdot 32^2 \hat{d}}\Big) + \frac{3}{m}\ln\frac{10}{\delta}\right]$$
$$+ C'\sqrt{\left[\mathcal{L}_{\mathcal{D}}^{\theta/16}(v_g) + \sqrt{\frac{2\,\mathcal{L}_{\mathcal{D}}^{\theta/16}(v_g)}{m}\ln\frac{10}{\delta}} + \frac{6}{m}\ln\frac{10}{\delta}\right]\left[\zeta\Big(\frac{\theta^2 m}{3 \cdot 32^2 \hat{d}}\Big) + \frac{3}{m}\ln\frac{10}{\delta}\right]}.$$
$$(30)$$

To bound $\mathcal{L}_{\mathcal{D}}^{\theta/16}(v_g)$, we make the following observation. Given function $f \in \{\pm 1\}^{\mathcal{X}}$, example $(x, y) \in \mathcal{X} \times \{\pm 1\}$ and voting classifier $v \in \operatorname{conv}(\mathcal{H})$, if $y \cdot v(x) \le \theta/16$, then either $f(x) = y$, so that $f(x) \cdot v(x) \le \theta/16$; or $f(x) = -y$, so that $y \cdot f(x) \le \theta/16$. Applying this for $f^\star$ and $v_g$, we conclude that

$$\mathcal{L}_{\mathcal{D}}^{\theta/16}(v_g) \le \operatorname{err}_{\mathcal{D}}(f^\star) + \mathcal{L}_{\mathcal{D}_{f^\star}}^{\theta/16}(v_g), \tag{31}$$

where we have used the definition of $\mathcal{D}_{f^\star}$. With Eq. (30) in mind, Eq. (31) yields that

$$\mathcal{L}_{\mathcal{D}}^{\theta/16}(v_g) + \sqrt{\frac{2\,\mathcal{L}_{\mathcal{D}}^{\theta/16}(v_g)}{m}\ln\frac{10}{\delta}} + \frac{6}{m}\ln\frac{10}{\delta}$$
$$\le \operatorname{err}_{\mathcal{D}}(f^\star) + \sqrt{\frac{2\operatorname{err}_{\mathcal{D}}(f^\star)}{m}\ln\frac{10}{\delta}} + \mathcal{L}_{\mathcal{D}_{f^\star}}^{\theta/16}(v_g) + \sqrt{\frac{2\,\mathcal{L}_{\mathcal{D}_{f^\star}}^{\theta/16}(v_g)}{m}\ln\frac{10}{\delta}} + \frac{6}{m}\ln\frac{10}{\delta}$$
$$\text{(as } \sqrt{a + b} \le \sqrt{a} + \sqrt{b} \text{ for } a, b > 0\text{)}$$
$$\le \operatorname{err}_{\mathcal{D}}(f^\star) + \sqrt{\frac{2\operatorname{err}_{\mathcal{D}}(f^\star)}{m}\ln\frac{10}{\delta}} + 2\,\mathcal{L}_{\mathcal{D}_{f^\star}}^{\theta/16}(v_g) + \Big(\frac{1}{2m} + \frac{6}{m}\Big)\ln\frac{10}{\delta} \tag{32}$$
$$\text{(as } 2\sqrt{ab} \le a + b \text{ for } a, b > 0 \text{ (AM–GM inequality))}$$
$$\le 2\operatorname{err}_{\mathcal{D}}(f^\star) + 2\,\mathcal{L}_{\mathcal{D}_{f^\star}}^{\theta/16}(v_g) + \frac{7}{m}\ln\frac{10}{\delta}, \tag{33}$$

where the last inequality follows again from the AM–GM inequality. Using Eq. (28) and that $C' \ge 1$, the two last inequalities (Eq. (33) and Eq. (32)) yield the two upper bounds, respectively:

$$\mathcal{L}_{\mathcal{D}}^{\theta/16}(v_g) + \sqrt{\frac{2\,\mathcal{L}_{\mathcal{D}}^{\theta/16}(v_g)}{m}\ln\frac{10}{\delta}} + \frac{6}{m}\ln\frac{10}{\delta}$$
$$\le \operatorname{err}_{\mathcal{D}}(f^\star) + \sqrt{\frac{2\operatorname{err}_{\mathcal{D}}(f^\star)}{m}\ln\frac{10}{\delta}} + 2C'\left[\zeta\Big(\frac{\theta^2 m}{3 \cdot 16^2 \hat{d}}\Big) + \frac{7}{m}\ln\frac{10}{\delta}\right],$$

and

$$\mathcal{L}_{\mathcal{D}}^{\theta/16}(v_g) + \sqrt{\frac{2\,\mathcal{L}_{\mathcal{D}}^{\theta/16}(v_g)}{m}\ln\frac{10}{\delta}} + \frac{6}{m}\ln\frac{10}{\delta}$$

$$\leq 2\operatorname{err}_{\mathcal{D}}(f^\star) + 2C'\left[\zeta\Big(\frac{\theta^2 m}{3\cdot 16^2\hat{d}}\Big) + \frac{7}{m}\ln\frac{10}{\delta}\right].$$

Applying both of these to Eq. (30), we conclude that

$$
\begin{aligned}
\mathcal{L}_{\mathcal{D}}(v_g') \leq{}& \operatorname{err}_{\mathcal{D}}(f^\star) + \sqrt{\frac{2\operatorname{err}_{\mathcal{D}}(f^\star)}{m}\ln\frac{10}{\delta}} \\
&+ 2C'\left[\zeta\Big(\frac{\theta^2 m}{3\cdot 16^2\hat{d}}\Big) + \frac{7}{m}\ln\frac{10}{\delta}\right] + C'\left[\zeta\Big(\frac{\theta^2 m}{3\cdot 32^2\hat{d}}\Big) + \frac{3}{m}\ln\frac{10}{\delta}\right] \\
&+ C'\sqrt{\left[2\operatorname{err}_{\mathcal{D}}(f^\star) + 2C'\Big(\zeta\Big(\frac{\theta^2 m}{3\cdot 16^2\hat{d}}\Big) + \frac{7}{m}\ln\frac{10}{\delta}\Big)\right]\left[\zeta\Big(\frac{\theta^2 m}{3\cdot 32^2\hat{d}}\Big) + \frac{3}{m}\ln\frac{10}{\delta}\right]} \\
\leq{}& \operatorname{err}_{\mathcal{D}}(f^\star) + \sqrt{\frac{2\operatorname{err}_{\mathcal{D}}(f^\star)}{m}\ln\frac{10}{\delta}} + 3C'\left[\zeta\Big(\frac{\theta^2 m}{3\cdot 32^2\hat{d}}\Big) + \frac{7}{m}\ln\frac{10}{\delta}\right] \\
&+ C'\sqrt{2\operatorname{err}_{\mathcal{D}}(f^\star)\left[\zeta\Big(\frac{\theta^2 m}{3\cdot 32^2\hat{d}}\Big) + \frac{3}{m}\ln\frac{10}{\delta}\right]} \\
&+ C'\sqrt{2C'}\cdot\left[\zeta\Big(\frac{\theta^2 m}{3\cdot 32^2\hat{d}}\Big) + \frac{7}{m}\ln\frac{10}{\delta}\right] \quad\quad (34)\\
\leq{}& \operatorname{err}_{\mathcal{D}}(f^\star) + (1+C')\sqrt{2\operatorname{err}_{\mathcal{D}}(f^\star)\left[\zeta\Big(\frac{\theta^2 m}{3\cdot 32^2\hat{d}}\Big) + \frac{3}{m}\ln\frac{10}{\delta}\right]} \\
&+ (3C' + C'\sqrt{2C'})\cdot\left[\zeta\Big(\frac{\theta^2 m}{3\cdot 32^2\hat{d}}\Big) + \frac{7}{m}\ln\frac{10}{\delta}\right] \\
\leq{}& \operatorname{err}_{\mathcal{D}}(f^\star) + (1+C')\sqrt{2\operatorname{err}_{\mathcal{D}}(f^\star)\left[\zeta\Big(\frac{\theta^2 m}{3\cdot 32^2\hat{d}}\Big) + \frac{3}{m}\ln\frac{10}{\delta}\right]} \\
&+ (3C' + C'\sqrt{2C'})\cdot\left[\zeta\Big(\frac{\theta^2 m}{3\cdot 32^2\hat{d}}\Big) + \frac{7}{m}\ln\frac{10}{\delta}\right] \\
\leq{}& \operatorname{err}_{\mathcal{D}}(f^\star) + \sqrt{\frac{C\operatorname{err}_{\mathcal{D}}(f^\star)}{m}\left[\frac{\hat{d}}{\theta^2}\ln^{3/2}\Big(\frac{\theta^2 m}{\hat{d}}\Big) + \ln\frac{10}{\delta}\right]} \\
&+ \frac{C}{m}\left[\frac{\hat{d}}{\theta^2}\ln^{3/2}\Big(\frac{\theta^2 m}{\hat{d}}\Big) + \ln\frac{10}{\delta}\right],
\end{aligned}
$$

where in Eq. (34) we used that $\sqrt{a+b} \leq \sqrt{a} + \sqrt{b}$ for $a, b > 0$, that $C' \geq 1$, and that $\zeta$ is decreasing, and in the last inequality we used the definition of $\zeta$ and that $m \geq 3072e^2 d/\theta^2$ such that $\theta^2 m/(3\cdot 32^2\hat{d}) \geq e^2$. Finally, since we show the above with probability at least $1 - \frac{2\delta}{10}$ over $\mathbf{S}_2$ and for any realization $S_1$ of $\mathbf{S}_1$ and the randomness of Algorithm 1 satisfying Eq. (27), which happens with probability at least $1 - \frac{2\delta}{10}$, it follows by independence that the bound on $v_g'$ holds with probability at least $1 - 4\delta/10 = 1 - \delta/2$. Since $v_g' \in \mathcal{B}_2$, this concludes the proof. $\qquad\square$

**Theorem D.4** (Restatement of 3.4)**.** *There exist universal constants $C, c > 0$ such that the following holds. Let $\mathcal{W}$ be a $(\gamma, \varepsilon_0, \delta_0, m_0, \mathcal{F}, \mathcal{H})$ agnostic weak learner. If $\gamma > \varepsilon_0$ and $\delta_0 < 1$, then, for all $\delta \in (0,1)$, $m \in \mathbb{N}$, and $\mathcal{D} \in \Delta(\mathcal{X} \times \{\pm 1\})$, given training sequence $\mathbf{S} \sim \mathcal{D}^m$, we have that Algorithm 2 on inputs $(\mathbf{S}, \mathcal{W}, \delta, \delta_0, m_0)$ returns, with probability at least $1 - \delta$ over $\mathbf{S}$ and the internal randomness of the algorithm, the output $\mathbf{v}$ of Algorithm 2 satisfies that*

$$\mathcal{L}_{\mathcal{D}}(\mathbf{v}) \leq \operatorname{err}_{\mathcal{D}}(f^\star) + \sqrt{C\operatorname{err}_{\mathcal{D}}(f^\star)\cdot\beta} + C\cdot\beta,$$

*where*

$$\beta = \frac{\hat{d}}{(\gamma - \varepsilon_0)^2 m} \cdot \mathrm{Ln}^{3/2}\left(\frac{(\gamma - \varepsilon_0)^2 m}{\hat{d}}\right) + \frac{1}{m}\ln\frac{\ln m}{\delta}$$

*with* $\hat{d} = \mathrm{fat}_{c(\gamma - \varepsilon_0)/32}(\mathcal{H})$.

*Proof.* We will now show that with probability atleast $1 - \delta$ over $\mathbf{S}$ and the randomness of Algorithm 1, we have that

$$\mathcal{L}_{\mathcal{D}}(v) \le \mathrm{err}_{\mathcal{D}}(f^\star) + \sqrt{\frac{11C\,\mathrm{err}_{\mathcal{D}}(f^\star)}{m} \cdot \left(\frac{\hat{d}}{\theta^2} \cdot \mathrm{Ln}^{3/2}\left(\frac{\theta^2 m}{\hat{d}}\right) + \ln\left(\frac{28\ln(m)}{\delta}\right)\right)} +$$

$$\frac{14C}{m}\left[\frac{\hat{d}}{\theta^2} \cdot \mathrm{Ln}^{3/2}\left(\frac{\theta^2 m}{\hat{d}}\right) + 16\ln\left(\frac{28\ln(m)}{\delta}\right)\right]$$

with $\hat{d} = \mathrm{fat}_{\hat{c}\theta/32}(\mathcal{H})$, $C \ge 1$ and $\hat{c} > 0$ being the universal constant of Lemma 3.3. Thus, it suffices to consider $m \ge 14$ else the right hand-side of the inequality is greater than 1 and we are done by the left hand-side being at most 1. Now by Lemma 3.3 we have that with probability at least $1 - \delta/2$ over $\mathbf{S}_1$ and $\mathbf{S}_2$, and the randomness of Algorithm 1 it holds that there exists $v_g \in \mathcal{B}_2$ such that

$$\mathcal{L}_{\mathcal{D}}(v_g) \le \mathrm{err}_{\mathcal{D}}(f^\star) + \sqrt{\frac{3C\,\mathrm{err}_{\mathcal{D}}(f^\star)}{m} \cdot \left(\frac{\hat{d}}{\theta^2} \cdot \mathrm{Ln}^{3/2}\left(\frac{\theta^2 m}{\hat{d}}\right) + \ln\frac{10e}{\delta}\right)}$$

$$+ \frac{3C}{m}\left[\frac{\hat{d}}{\theta^2} \cdot \mathrm{Ln}^{3/2}\left(\frac{\theta^2 m}{\hat{d}}\right) + \ln\frac{10e}{\delta}\right] \tag{35}$$

with $\hat{d} = \mathrm{fat}_{\hat{c}\theta/32}(\mathcal{H})$, $C \ge 1$ and $\hat{c} > 0$ being the universal constant of Lemma 3.3, call this event $E_1$. Now consider any realization $S_1, S_2$ of $\mathbf{S}_1, \mathbf{S}_2$ and the randomness used in Algorithm 1 such that the above holds (so, within event $E_1$) and let $v_g$ denote an arbitrary $v \in \mathcal{B}_2$ such that the above holds.

We now invoke both equations of Lemma D.1 with $\delta = \delta/(4|\mathcal{B}_2|)$ (abusing notation of $\delta$) $\gamma = 0$ for each $v \in \mathcal{B}_2$ which combined with a union bound give use that it holds with probability at least $1 - \delta/2$ over $\mathbf{S}_3 \sim \mathcal{D}^{m/3}$ that

$$\mathcal{L}_{\mathcal{D}}(v) \le \mathcal{L}_{\mathbf{S}_3}(v) + \sqrt{\frac{6\,\mathcal{L}_{\mathbf{S}_3}(v)\ln(4|\mathcal{B}_2|/\delta)}{m}} + \frac{12\ln(4|\mathcal{B}_2|/\delta)}{m}$$

and

$$\mathcal{L}_{\mathbf{S}_3}(v) \le \mathcal{L}_{\mathcal{D}}(v) + \sqrt{\frac{6\,\mathcal{L}_{\mathcal{D}}(v)\ln(4|\mathcal{B}_2|/\delta)}{3m}} + \frac{6\ln(4|\mathcal{B}_2|/\delta)}{m}. \tag{36}$$

Consider such a realization $S_3$ of $\mathbf{S}_3$, and denote an event where the above inequalities hold by $E_2$. Now let $v$ be the voting classifier in $\mathcal{B}_2$ with the smallest empirical 0-margin loss - $v = argmin_{v \in \mathcal{B}_2}\,\mathcal{L}_{\mathbf{S}_3}(v)$, with ties broken arbitrary (i.e., the output of Algorithm 2). Now by Eq. (36) and $v \in \mathcal{B}_2$, we have that

$$\mathcal{L}_{\mathcal{D}}(v) \le \mathcal{L}_{\mathbf{S}_3}(v) + \sqrt{\frac{6\,\mathcal{L}_{\mathbf{S}_3}(v)\ln(4|\mathcal{B}_2|/\delta)}{m}} + \frac{12\ln(4|\mathcal{B}_2|/\delta)}{m}$$

$$\le \mathcal{L}_{\mathbf{S}_3}(v_g) + \sqrt{\frac{6\,\mathcal{L}_{\mathbf{S}_3}(v_g)\ln(4|\mathcal{B}_2|/\delta)}{m}} + \frac{12\ln(4|\mathcal{B}_2|/\delta)}{m}$$

$$\le \mathcal{L}_{\mathcal{D}}(v_g) + \sqrt{\frac{6\,\mathcal{L}_{\mathcal{D}}(v_g)\ln(4|\mathcal{B}_2|/\delta)}{3m}} + \sqrt{\frac{6\,\mathcal{L}_{\mathbf{S}_3}(v_g)\ln(4|\mathcal{B}_2|/\delta)}{m}} + \frac{18\ln(4|\mathcal{B}_2|/\delta)}{m}, \tag{37}$$

where the first inequality follows from Eq. (36) and $v \in \mathcal{B}_2$, the second inequality from $v$ being a empirical minimizer of $\mathcal{L}_{\mathbf{S}_3}$, so $\mathcal{L}_{\mathbf{S}_3}(v) \le \mathcal{L}_{\mathbf{S}_3}(v_g)$ and the last Eq. (36). Now by Eq. (36) we have

$$\mathcal{L}_{\mathbf{S}_3}(v_g) \le 2\,\mathcal{L}_{\mathcal{D}}(v_g) + \frac{12\ln(4|\mathcal{B}_2|/\delta)}{m} \tag{38}$$

where the inequality follows by $\sqrt{ab} \leq a + b$ for $a, b > 0$. This implies that

$$\sqrt{\frac{6\,\mathcal{L}_{\mathbf{S}_3}(v_g)\ln(4|\mathcal{B}_2|/\delta)}{m}} \leq \sqrt{\frac{6\left(2\,\mathcal{L}_{\mathcal{D}}(v_g) + \frac{12\ln(4|\mathcal{B}_2|/\delta)}{m}\right)\ln(4|\mathcal{B}_2|/\delta)}{m}}$$

$$\leq \sqrt{\frac{12\,\mathcal{L}_{\mathcal{D}}(v_g)\ln(4|\mathcal{B}_2|/\delta)}{m}} + \frac{18\ln\left(4|\mathcal{B}_2|/\delta\right)}{m}, \qquad (39)$$

where the first inequality follows from Eq. (38) and the second by $\sqrt{a+b} \leq \sqrt{a} + \sqrt{b}$. Whereby plugging Eq. (39) into Eq. (37) gives that

$$\mathcal{L}_{\mathcal{D}}(v) \leq \mathcal{L}_{\mathcal{D}}(v_g) + \sqrt{\frac{6\,\mathcal{L}_{\mathcal{D}}(v_g)\ln(4|\mathcal{B}_2|/\delta)}{3m}} + \sqrt{\frac{12\,\mathcal{L}_{\mathcal{D}}(v_g)\ln(4|\mathcal{B}_2|/\delta)}{m}}$$

$$+ \frac{18\ln\left(4|\mathcal{B}_2|/\delta\right)}{m} + \frac{18\ln(4|\mathcal{B}_2|/\delta)}{m}$$

$$\leq \mathcal{L}_{\mathcal{D}}(v_g) + \sqrt{\frac{36\,\mathcal{L}_{\mathcal{D}}(v_g)\ln(4|\mathcal{B}_2|/\delta)}{3m}} + \frac{36\ln\left(4|\mathcal{B}_2|/\delta\right)}{m}$$

$$\leq \mathcal{L}_{\mathcal{D}}(v_g) + \sqrt{\frac{36\,\mathcal{L}_{\mathcal{D}}(v_g)\ln(16\ln\left(m\right)/\delta)}{3m}} + \frac{36\ln\left(16\ln\left(m\right)/\delta\right)}{m}, \qquad (40)$$

where the first inequality follows by Eq. (39), and the last by $|\mathcal{B}_2| \leq \lfloor \log_2(\sqrt{m}) \rfloor + 2 \leq 4\ln\left(m\right)$, since we consider the case $m \geq 14$. Now by using Eq. (35) and $\sqrt{ab} \leq a + b$ we get that

$$\mathcal{L}_{\mathcal{D}}(v_g) \leq 2\operatorname{err}_{\mathcal{D}}(f^\star) + \frac{6C}{m}\left[\frac{\hat{d}}{\theta^2} \cdot \operatorname{Ln}^{3/2}\left(\frac{\theta^2 m}{\hat{d}}\right) + \ln\frac{10e}{\delta}\right] \qquad (41)$$

Thus, we have that

$$\sqrt{\frac{36\,\mathcal{L}_{\mathcal{D}}(v_g)\ln(12\ln\left(m\right)/\delta)}{m}}$$

$$\leq \sqrt{\frac{36\left(2\operatorname{err}_{\mathcal{D}}(f^\star) + \frac{6C}{m}\left[\frac{\hat{d}}{\theta^2} \cdot \operatorname{Ln}^{3/2}\left(\frac{\theta^2 m}{\hat{d}}\right) + \ln\frac{10e}{\delta}\right]\right)\ln(12\ln\left(m\right)/\delta)}{m}}$$

$$\leq \sqrt{\frac{72\operatorname{err}_{\mathcal{D}}(f^\star)\ln(12\ln\left(m\right)/\delta)}{m}} + \frac{14C}{m}\left[\frac{\hat{d}}{\theta^2} \cdot \operatorname{Ln}^{3/2}\left(\frac{\theta^2 m}{\hat{d}}\right) + 12\ln\left(\frac{28\ln\left(m\right)}{\delta}\right)\right], \quad (42)$$

where the first inequality follows from Eq. (41), and the second by $\sqrt{a+b} \leq \sqrt{a} + \sqrt{b}$. Now plugging in Eq. (35) and Eq. (42) into Eq. (40) we get that

$$\mathcal{L}_{\mathcal{D}}(v) \leq \mathcal{L}_{\mathcal{D}}(v_g) + \sqrt{\frac{36\,\mathcal{L}_{\mathcal{D}}(v_g)\ln(16\ln\left(m\right)/\delta)}{3m}} + \frac{36\ln\left(16\ln\left(m\right)/\delta\right)}{m}$$

$$\leq \operatorname{err}_{\mathcal{D}}(f^\star) + \sqrt{\frac{3C\operatorname{err}_{\mathcal{D}}(f^\star)}{m} \cdot \left(\frac{\hat{d}}{\theta^2} \cdot \operatorname{Ln}^{3/2}\left(\frac{\theta^2 m}{\hat{d}}\right) + \ln\frac{10e}{\delta}\right)}$$

$$+ \frac{3C}{m}\left[\frac{\hat{d}}{\theta^2} \cdot \operatorname{Ln}^{3/2}\left(\frac{\theta^2 m}{\hat{d}}\right) + \ln\frac{10e}{\delta}\right]$$

$$+ \sqrt{\frac{72\operatorname{err}_{\mathcal{D}}(f^\star)\ln(12\ln\left(m\right)/\delta)}{m}} + \frac{14C}{m}\left[\frac{\hat{d}}{\theta^2} \cdot \operatorname{Ln}^{3/2}\left(\frac{\theta^2 m}{\hat{d}}\right) + 12\ln\left(\frac{28\ln\left(m\right)}{\delta}\right)\right]$$

$$+ \frac{36\ln\left(16\ln\left(m\right)/\delta\right)}{m}$$

$$\leq \operatorname{err}_{\mathcal{D}}(f^\star) + \sqrt{\frac{11C\operatorname{err}_{\mathcal{D}}(f^\star)}{m} \cdot \left(\frac{\hat{d}}{\theta^2} \cdot \operatorname{Ln}^{3/2}\left(\frac{\theta^2 m}{\hat{d}}\right) + \ln\left(\frac{28\ln\left(m\right)}{\delta}\right)\right)}$$

$$+ \frac{14C}{m}\left[\frac{\hat{d}}{\theta^2} \cdot \operatorname{Ln}^{3/2}\left(\frac{\theta^2 m}{\hat{d}}\right) + 16\ln\left(\frac{28\ln\left(m\right)}{\delta}\right)\right]. \qquad (43)$$

Let the above event be denoted $E_3$. Thus, we have shown the above for any realization $S_1$ and $S_2$ of $\mathbf{S}_1$ and $\mathbf{S}_2$ and the randomness of Algorithm 1 which are in $E_1$, and $\mathbf{S}_3$ on th event $E_2$ the output of Algorithm 2 achieves the error bound of Eq. (43). Thus, since the randomness of $\mathbf{S}_1, \mathbf{S}_2, \mathbf{S}_3$ and the randomness over Algorithm 1 are independent and $E_1$ and $E_2$ both happened with probability at least $1 - \delta/2$ over respectively $\mathbf{S}_1, \mathbf{S}_2$ and the randomness of Algorithm 1 and $\mathbf{S}_3$, the proof follows by (let $\mathbf{r}$ denote the randomness of Algorithm 1)

$$
\begin{aligned}
\mathbb{P}_{\mathbf{S}\sim\mathcal{D}^m,\mathbf{r}}\left[E_3\right] &\geq \mathbb{E}_{\mathbf{S}_1,\mathbf{S}_2\sim\mathcal{D}^{m/3},\mathbf{r}}\left[\mathbb{P}_{\mathbf{S}_3\sim\mathcal{D}^{m/3}}\left[E_3\right]\mathbb{1}\{E_1\}\right]\\
&\geq \mathbb{E}_{\mathbf{S}_1,\mathbf{S}_2\sim\mathcal{D}^{m/3},\mathbf{r}}\left[\mathbb{P}_{\mathbf{S}_3\sim\mathcal{D}^{m/3}}\left[E_2,E_3\right]\mathbb{1}\{E_1\}\right]\\
&\geq \mathbb{E}_{\mathbf{S}_1,\mathbf{S}_2\sim\mathcal{D}^{m/3},\mathbf{r}}\left[\mathbb{P}_{\mathbf{S}_3\sim\mathcal{D}^{m/3}}\left[E_2\right]\mathbb{1}\{E_1\}\right]\\
&\geq (1-\delta/2)^2\\
&\geq 1-\delta,
\end{aligned}
$$

where the third inequality follows from $E_1$ and $E_2$ implying $E_3$ and the fourth inequality by $E_2$ given $E_1$ happens holds with probability at least $1 - \delta/2$ over $\mathbf{S}_3$ and that $E_1$ holds with probability at least $1 - \delta/2$ over $\mathbf{S}_1, \mathbf{S}_2$ and the randomness $\mathbf{r}$ over Algorithm 1 which concludes the proof of the theorem. $\qquad\square$

# E   Lower Bound

In this section, we present the proof of our lower bound on the sample complexity of agnostic weak-to-strong learning. We first re-state the result in a more general form than that in Section 1.

**Theorem E.1.** *For all integer $d > 0$ and $\gamma \in (0, 1]$ such that $d \geq 8\log_2(2/\gamma^2)$, and all $\varepsilon_0, \delta_0 \in (0, 1]$, there exist a universe $\mathcal{X}$, a base class $\mathcal{B} \subseteq \{\pm 1\}^{\mathcal{X}}$ with VC dimension at most $d$, a reference class $\mathcal{F} \subseteq \{\pm 1\}^{\mathcal{X}}$, and a $(\gamma, \delta_0, \varepsilon_0, m_0, \mathcal{F}, \mathcal{B})$ agnostic weak learner with $m_0 = \left\lceil \frac{8d\ln(4/(\delta_0\gamma^2))}{\varepsilon_0^2} \right\rceil$ such that for any $L \in (0, 1/2)$ and any learner $\mathcal{A}$, it holds that there exists data distribution $\mathcal{D}$ such that $\inf_{f\in\mathcal{F}}\{\mathrm{err}_{\mathcal{D}}(f)\} = L$ and for $m \geq \frac{d}{\gamma^2 L(1-2L)^2}$ we have with probability at least $1/50$ over $\mathbf{S} \sim \mathcal{D}^m$ that*

$$
\mathbb{E}_{\mathbf{S}\sim\mathcal{D}^m}[\mathrm{err}_{\mathcal{D}}(\mathcal{A}(\mathbf{S}))] \geq \inf_{f\in\mathcal{F}}\mathrm{err}_{\mathcal{D}}(f) + \frac{2}{50}\sqrt{\frac{d\inf_{f\in\mathcal{F}}\mathrm{err}_{\mathcal{D}}(f)}{32\gamma^2 m\log_2(2/\gamma^2)}}.
$$

*Furthermore, for all $\varepsilon \in (0, \sqrt{2}/2]$, $\delta \in (0, 1)$, and any learning algorithm $\mathcal{A}$, there exists data distribution $\mathcal{D}$ such that if $m \leq \ln(1/(4\delta))/(2\varepsilon^2)$, then with probability at least $\delta$ over $\mathbf{S} \sim \mathcal{D}^m$ we have that*

$$
\mathrm{err}_{\mathcal{D}}(\mathcal{A}(\mathbf{S})) \geq \inf_{f\in\mathcal{F}}\mathrm{err}_{\mathcal{D}}(f) + \varepsilon,
$$

*and for any $\varepsilon \in (0, \sqrt{2}/16]$, and any learning algorithm $\mathcal{A}\colon (\mathcal{X}\times\{\pm 1\})^* \to \{\pm 1\}^{\mathcal{X}}$ there exists data distribution $\mathcal{D}$ such that if $m < \frac{d}{2048\gamma^2\log_2(2/\gamma^2)\varepsilon^2}$, then with probability at least $1/8$ over $\mathbf{S} \sim \mathcal{D}^m$ we have that*

$$
\mathrm{err}_{\mathcal{D}}(\mathcal{A}(\mathbf{S})) \geq \inf_{f\in\mathcal{F}}\mathrm{err}_{\mathcal{D}}(f) + \varepsilon.
$$

To prove Theorem E.1 we need the following lemma giving the construction of a hard instance.

**Lemma E.2.** *Let $n, s > 0$ be integers, and $\varepsilon_0, \delta_0 \in (0, 1]$. If $n$ is a power of 2, then there exists a universe $\mathcal{X} = [n \cdot s]$, a base class $\mathcal{B} \subseteq \{\pm 1\}^{\mathcal{X}}$ with $|\mathcal{B}| = (2n)^s$, a reference class $\mathcal{F} = \{\pm 1\}^{\mathcal{X}}$ (all possible mappings $\mathcal{X} \to \{\pm 1\}$), and a $(1/\sqrt{n}, \delta_0, \varepsilon_0, m_0, \mathcal{F}, \mathcal{B})$ agnostic weak learner for any $m_0 \geq \left\lceil \frac{8\ln(2|\mathcal{B}|/\delta_0)}{\varepsilon_0^2} \right\rceil$.*

We postpone the proof of Lemma E.2 to the end of this section, and now show how to combine it with the following classic results to obtain the claimed bounds.

**Lemma E.3** (Devroye et al. [1996, Theorem 14.5]). *Let $\mathcal{X}$ be a universe and $\mathcal{F} \subseteq \mathcal{X} \to \{\pm 1\}$ be a function class with $\mathrm{VC}(\mathcal{F}) = d \geq 2$. Then, for any $L \in (0, 1/2)$, and any learning algorithm*

$\mathcal{A}\colon (\mathcal{X} \times \{\pm 1\})^* \to \{\pm 1\}^{\mathcal{X}}$ *there exists data distribution* $\mathcal{D}$ *such that* $\inf_{f \in \mathcal{F}}\{\mathrm{err}_{\mathcal{D}}(f)\} = L$ *and for* $m \geq \frac{d-1}{2L}\max\{9, \frac{1}{(1-2L)^2}\}$ *we have that*

$$\mathbb{E}_{\mathbf{S} \sim \mathcal{D}^m}[\mathrm{err}_{\mathcal{D}}(\mathcal{A}(\mathbf{S}))] \geq \inf_{f \in \mathcal{F}} \mathrm{err}_{\mathcal{D}}(f) + \sqrt{\frac{(d-1) \cdot \inf_{f \in \mathcal{F}} \mathrm{err}_{\mathcal{D}}(f)}{24m}} e^{-8}.$$

By slightly modifying the proof of Devroye et al. [1996], we conclude that the above lower bound holds with constant probability, a result provided in the next lemma. For completeness, we provide its proof in the end of this appendix.

**Lemma E.4.** *Let* $\mathcal{X}$ *be a universe and* $\mathcal{F} \subseteq \mathcal{X} \to \{\pm 1\}$ *be a function class with* $\mathrm{VC}(\mathcal{F}) = d \geq 2$. *Then, for any* $L \in (0, 1/2)$, *and any learning algorithm* $\mathcal{A}\colon (\mathcal{X} \times \{\pm 1\})^* \to \{\pm 1\}^{\mathcal{X}}$ *there exists data distribution* $\mathcal{D}$ *such that* $\inf_{f \in \mathcal{F}}\{\mathrm{err}_{\mathcal{D}}(f)\} = L$ *and for* $m \geq \frac{d}{L(1/2 - L)^2}$ *it holds with probability at least* $1/50$ *over* $\mathbf{S} \sim \mathcal{D}^m$ *that*

$$\mathrm{err}_{\mathcal{D}}(\mathcal{A}(\mathbf{S})) \geq \inf_{f \in \mathcal{F}} \mathrm{err}_{\mathcal{D}}(f) + \frac{2}{50}\sqrt{\frac{d \cdot \inf_{f \in \mathcal{F}} \mathrm{err}_{\mathcal{D}}(f)}{16m}}.$$

We furthermore need the following lower bound on the sample complexity of agnostic learning.

**Lemma E.5** (Shalev-Shwartz and Ben-David [2014, Section 28.2, pgs. 393-398])**.** *Let* $\mathcal{X}$ *be a universe and* $\mathcal{F} \subseteq \{\pm 1\}^{\mathcal{X}}$ *be a function class with* $\mathrm{VC}(\mathcal{F}) = d \geq 2$. *Then, for any* $\varepsilon \in (0, 1/\sqrt{2}]$, $\delta \in (0, 1)$, *and any learning algorithm* $\mathcal{A}\colon (\mathcal{X} \times \{\pm 1\})^* \to \{\pm 1\}^{\mathcal{X}}$ *there exists a data distribution* $\mathcal{D}$ *such that if* $m \leq \ln(1/(4\delta))/(2\varepsilon^2)$, *then with probability at least* $\delta$ *over* $\mathbf{S} \sim \mathcal{D}^m$ *we have that*

$$\mathrm{err}_{\mathcal{D}}(\mathcal{A}(\mathbf{S})) \geq \inf_{f \in \mathcal{F}} \mathrm{err}_{\mathcal{D}}(f) + \varepsilon.$$

*Furthermore, for any* $\varepsilon \in (0, 1/(8\sqrt{2})]$ *and any learning algorithm* $\mathcal{A}\colon (\mathcal{X} \times \{\pm 1\})^* \to \{\pm 1\}^{\mathcal{X}}$ *there exists data distribution* $\mathcal{D}$ *such that if* $m < \frac{d}{512\varepsilon^2}$, *then with probability at least* $1/8$ *over* $\mathbf{S} \sim \mathcal{D}^m$ *we have that*

$$\mathrm{err}_{\mathcal{D}}(\mathcal{A}(\mathbf{S})) \geq \inf_{f \in \mathcal{F}} \mathrm{err}_{\mathcal{D}}(f) + \varepsilon.$$

With the above lemmas in place we now give the proof of Theorem E.1.

*Proof of Theorem E.1.* We start by applying Lemma E.2 with parameters $\varepsilon_0, \delta_0, n = 2^r$ for $r \in \mathbb{Z}_{\geq 0}$ such that

$$n = 2^r \leq 1/\gamma^2 < 2^{r+1}, \tag{44}$$

and $s = \lfloor d/\log_2(2n) \rfloor$. Notice that $s$ is a positive integer since $n \leq 1/\gamma^2$, and, by hypothesis, $d \geq \log_2(2/\gamma^2)$.

$$\frac{d}{\log_2(2n)} \geq \frac{d}{\log_2(2/\gamma^2)} \qquad \text{(by Eq. (44))}$$

$$\geq 1. \qquad \text{(as, by hypothesis, } d \geq \log_2(2/\gamma^2)\text{)}$$

The base class $\mathcal{B}$ ensured by Lemma E.2 satisfies $|\mathcal{B}| = (2n)^s$, so

$$\begin{aligned} \mathrm{VC}(\mathcal{B}) &\leq \log_2(|\mathcal{B}|) \\ &= s \log_2(2n) \\ &= \left\lfloor \frac{d}{\log_2(2n)} \right\rfloor \log_2(2n) \qquad \text{(by the choice of } s\text{)} \\ &\leq d, \end{aligned}$$

as desired.

Moreover, Lemma E.2 guarantees the existence of a $(\frac{1}{\sqrt{n}}, \delta_0, \varepsilon_0, m_0, \mathcal{F}, \mathcal{B})$ agnostic weak learner, denoted $\mathcal{W}$, for the reference class $\mathcal{F} = \{\pm 1\}^{\mathcal{X}}$ for any $m_0 \geq \lceil 8\ln(2|\mathcal{B}|/\delta_0)/\varepsilon_0^2 \rceil$. For later use, we

choose $m_0 = \left\lceil 8d \ln(4/(\delta_0 \gamma^2))/\varepsilon_0^2 \right\rceil$, which is a valid choice since

$$
\left\lceil \frac{8 \ln(2|\mathcal{B}|/\delta_0)}{\varepsilon_0^2} \right\rceil = \left\lceil \frac{8 \ln(2(2n)^s/\delta_0)}{\varepsilon_0^2} \right\rceil
$$

$$
\leq \left\lceil \frac{8s \ln(2(2n)/\delta_0)}{\varepsilon_0^2} \right\rceil \qquad \text{(as } s \geq 1\text{)}
$$

$$
\leq \left\lceil \frac{8d \ln(4/(\delta_0 \gamma^2))}{\log_2(2n)\varepsilon_0^2} \right\rceil \qquad \text{(as } s = \lfloor d/\log_2(2n) \rfloor \text{ and, by choice, } n \leq 1/\gamma^2\text{)}
$$

$$
\leq \left\lceil \frac{8d \ln(4/(\delta_0 \gamma^2))}{\varepsilon_0^2} \right\rceil. \qquad \text{(as } n \geq 1\text{)}
$$

We claim that $\mathcal{W}$ is also a $(\gamma, \delta_0, \varepsilon_0, m_0, \mathcal{F}, \mathcal{B})$ agnostic weak learner. Indeed, given any $\mathcal{D}' \in \Delta(\mathcal{X} \times \{\pm 1\})$, as $\mathcal{F}$ consists of all possible mappings from $\mathcal{X}$ to $\{\pm 1\}$, we have that $\sup_{f \in \mathcal{F}} \mathbb{E}_{(\mathbf{x}, \mathbf{y}) \sim \mathcal{D}'}[\mathbf{y} \cdot f(\mathbf{x})] \geq 0$. Thus, it holds that $\frac{1}{\sqrt{n}} \sup_{f \in \mathcal{F}} \mathbb{E}_{(\mathbf{x}, \mathbf{y}) \sim \mathcal{D}'}[\mathbf{y} \cdot f(\mathbf{x})] \geq \gamma \sup_{f \in \mathcal{F}} \mathbb{E}_{(\mathbf{x}, \mathbf{y}) \sim \mathcal{D}'}[\mathbf{y} \cdot f(\mathbf{x})]$, since $1/\sqrt{n} \geq \gamma$, by Eq. (44).

Finally, since $\mathcal{F}$ is the set of all possible mappings from $\mathcal{X} = [n \cdot s]$ to $\{\pm 1\}$, we have that

$$
\mathrm{VC}(\mathcal{F}) = n \cdot s
$$

$$
= 2^r \cdot \left\lfloor \frac{d}{\log_2(2^{r+1})} \right\rfloor \qquad \text{(by the choice of } n \text{ and } s\text{)}
$$

$$
= 2^r \cdot \left\lfloor \frac{d}{r+1} \right\rfloor
$$

$$
\leq \frac{d}{\gamma^2}. \qquad \text{(by Eq. (44))}
$$

On the other hand,

$$
\mathrm{VC}(\mathcal{F}) = 2^r \cdot \left\lfloor \frac{d}{\log_2(2^{r+1})} \right\rfloor
$$

$$
\geq \frac{1}{2\gamma^2} \cdot \left\lfloor \frac{d}{\log_2(2/\gamma^2)} \right\rfloor \qquad \text{(by Eq. (44))}
$$

$$
\geq \frac{8}{2\gamma^2} \qquad \text{(as, by hypothesis, } d/\log_2(2/\gamma^2) \geq 8\text{)}
$$

$$
\geq 2, \qquad \text{(as, by hypothesis, } \gamma \leq 1\text{)}
$$

allowing us to apply Lemma E.4 and Lemma E.5, respectively, to obtain the thesis. $\qquad \square$

With the proof of Theorem E.1 done, we now prove Lemma E.2.

*Proof of Lemma E.2.* We start by considering $[n]$ as our universe for $n$ a power of 2. Let $v^{(1)}, \ldots, v^{(n)}$ be a set of $n$ pairwise orthogonal vectors in $\{\pm 1\}^n$, which can be chosen as the rows of a Hadamard matrix of size $n \times n$.[7]

Let now $\mathcal{D}$ be a probability distribution over $[n]$, which can be seen as a vector in $[0, 1]^n$ with $\sum_{i=1}^n \mathcal{D}_i = 1$. Scaling $v^{(1)}, \ldots, v^{(n)}$ by $1/\sqrt{n}$ we obtain a orthonormal basis, whereby

$$
\mathcal{D} = \sum_{i=1}^n \left\langle \mathcal{D}, \frac{v^i}{\sqrt{n}} \right\rangle \cdot \frac{v^i}{\sqrt{n}}. \tag{45}
$$

---

[7]We assume $n$ to be a power of 2 as this suffices to ensure the existence of such an $n \times n$ Hadamard matrix.

Moreover, we have that

$$\frac{1}{n} = \frac{(\sum_{i=1}^{n} \mathcal{D}_i)^2}{n}$$

$$\leq \sum_{i=1}^{n} \mathcal{D}_i^2 \qquad \text{(by Cauchy-Schwarz)}$$

$$= \|\mathcal{D}\|_2^2$$

$$= \Big\| \sum_{i=1}^{n} \Big\langle \mathcal{D}, \frac{v^{(i)}}{\sqrt{n}} \Big\rangle \cdot \frac{v^{(i)}}{\sqrt{n}} \Big\|_2^2 \qquad \text{(by Eq. (45))}$$

$$= \frac{1}{n} \sum_{i=1}^{n} \langle \mathcal{D}, v^{(i)} \rangle^2,$$

where the last equality follows from $\langle v^{(i)}, v^{(j)} \rangle$ being 0 for $i \neq j$ and $n$ for $i = j$. By averaging, the above implies that for any $\mathcal{D} \in \Delta([n])$ there exists $i \in [n]$ such that $\langle \mathcal{D}, v^{(i)} \rangle^2 \geq 1/n$, so that either $\langle \mathcal{D}, v^{(i)} \rangle \geq 1/\sqrt{n}$ or $-\langle \mathcal{D}, v^{(i)} \rangle \geq 1/\sqrt{n}$. Similarly, denoting by $\odot$ the entry-wise product, we have that for any $y \in \{\pm 1\}^n$, the vectors $y \odot v^{(1)}/\sqrt{n}, \ldots, y \odot v^{(n)}/\sqrt{n}$ form an orthonormal basis of $\mathbb{R}^n$. Thus, an analogous implies that for any $\mathcal{D} \in \Delta([n])$ and $y \in \{\pm 1\}^n$ there exists $i \in [n]$ such that either $\langle \mathcal{D}, y \odot v^{(i)} \rangle \geq 1/\sqrt{n}$ or $\langle \mathcal{D}, y \odot (-v^{(i)}) \rangle \geq 1/\sqrt{n}$. Overall, we can conclude that for any $\mathcal{D} \in \Delta([n])$ and any labeling $y \in \{\pm 1\}^n$ there exists $v \in \{v^{(1)}, \ldots, v^{(n)}, -v^{(1)}, \ldots, -v^{(n)}\}$ such that

$$\mathbb{E}_{\mathbf{j} \sim \mathcal{D}}[y_{\mathbf{j}} \cdot v_{\mathbf{j}}] = \sum_{j=1}^{n} \mathcal{D}_j y_j v_j$$

$$= \langle \mathcal{D}, y \odot v \rangle$$

$$\geq \frac{1}{\sqrt{n}}. \tag{46}$$

Consider the base class $\mathcal{B} \colon [n \cdot s] \to \{\pm 1\}$ consisting of the possible concatenations of $s$ vectors in $V \coloneqq \{v^{(1)}, \ldots, v^{(n)}, -v^{(1)}, \ldots, -v^{(n)}\}$. That is, $\mathcal{B} = \{(w^{(1)}, \ldots, w^{(s)}) \in \{\pm 1\}^{n \cdot s} : w^{(1)}, \ldots, w^{(s)} \in V\}$. We claim that for any $\mathcal{D} \in \Delta([n \cdot s])$ and $y \in \{\pm 1\}^{n \cdot s}$ there exists $h \in \mathcal{B}$ such that $\mathbb{E}_{\mathbf{j} \sim \mathcal{D}}[y_{\mathbf{j}} h_{\mathbf{j}}] \geq 1/\sqrt{n}$. To see this, consider $h = (w^{(1)}, \ldots, w^{(s)})$ such that each $w^{(i)}$ satisfies $\sum_{j=(i-1) \cdot n + 1}^{i \cdot n} \mathcal{D}_j y_j w_{j-(i-1) \cdot n}^{(i)} \geq \frac{1}{\sqrt{n}} \sum_{j=(i-1) \cdot n + 1}^{i \cdot n} \mathcal{D}_j$. There must exist such $w^{(i)}$ as this is trivially the case when $\sum_{j=(i-1) \cdot n + 1}^{i \cdot n} \mathcal{D}_j = 0$ and, otherwise, we have that

$$\sum_{j=(i-1) \cdot n + 1}^{i \cdot n} \mathcal{D}_j y_j w_{j-(i-1) \cdot n}^{(i)} = \sum_{j=(i-1) \cdot n + 1}^{i \cdot n} \mathcal{D}_j \cdot \sum_{j=(i-1) \cdot n + 1}^{i \cdot n} y_j w_{j-(i-1) \cdot n}^{(i)} \frac{\mathcal{D}_j}{\sum_{k=(i-1) \cdot n + 1}^{i \cdot n} \mathcal{D}_k},$$

with $\big(\mathcal{D}_j / (\sum_{k=(i-1) \cdot n + 1}^{i \cdot n} \mathcal{D}_k)\big)_{j=(i-1) \cdot n + 1}^{i \cdot n}$ being a probability distribution, so the existence of the desired $w^{(i)}$ follows from Eq. (46).

So far, for any $n$ integer power of 2 and $s \in \mathbb{N}$ we have constructed a universe $\mathcal{X} = [n \cdot s]$ and a base class $\mathcal{B} \in \{\pm 1\}^{\mathcal{X}}$ such that for any $\mathcal{D} \in \Delta(\mathcal{X})$ and $f \colon \mathcal{X} \to \{\pm 1\}$ there exists $h \in \mathcal{B}$ such that

$$\mathbb{E}_{x \sim \mathcal{D}}[h(x) f(x)] \geq \frac{1}{\sqrt{n}}. \tag{47}$$

To conclude, we show the existence of a $(\frac{1}{\sqrt{n}}, \delta_0, \varepsilon_0, m_0, \mathcal{F}, \mathcal{B})$ agnostic weak learner, for the reference class $\mathcal{F} = \{\pm 1\}^{\mathcal{X}}$, the set of all functions from $\mathcal{X}$ to $\{\pm 1\}$, as long as $m_0 \geq \big\lceil \frac{8 \ln(2|\mathcal{B}|/\delta_0)}{\varepsilon_0^2} \big\rceil$. Concretely, we shall prove that there exists a mapping $\mathcal{W} \colon (\mathcal{X} \times \{\pm 1\})^* \to \{\pm 1\}^{\mathcal{X}}$ (from training sequences to classifiers), such that for all $\mathcal{D}' \in \Delta(\mathcal{X} \times \{\pm 1\})$, when $\mathcal{W}$ is provided a sample $\mathbf{S} \sim \mathcal{D}'^{m_0}$ we have that $\mathcal{W}(\mathbf{S}) = h \in \mathcal{B}$ and with probability at least $1 - \delta_0$ over $\mathbf{S}$ it holds that

$$\mathbb{E}_{(\mathbf{x}, \mathbf{y}) \sim \mathcal{D}'}[h(\mathbf{x}) \mathbf{y}] \geq \frac{1}{\sqrt{n}} \cdot \sup_{f \in \mathcal{F}} \mathbb{E}_{(\mathbf{x}, \mathbf{y}) \sim \mathcal{D}'}[f(\mathbf{x}) \mathbf{y}] - \varepsilon_0.$$

To this end, it suffices to show that for any $\mathcal{D}'$

$$\underset{h \in \mathcal{B}}{\arg\max} \left\{ \mathbb{E}_{(\mathbf{x},\mathbf{y}) \sim \mathcal{D}'}[h(\mathbf{x})\mathbf{y}] \right\} \geq \frac{1}{\sqrt{n}} \cdot \sup_{f \in \mathcal{F}} \mathbb{E}_{(\mathbf{x},\mathbf{y}) \sim \mathcal{D}'}[f(\mathbf{x})\mathbf{y}] \tag{48}$$

and to let $\mathcal{W}(S) = \arg\max_{h \in \mathcal{B}} \{\mathbb{E}_{(\mathbf{x},\mathbf{y}) \sim S}[h(\mathbf{x})\mathbf{y}]\}$. To see why this is a $(\frac{1}{\sqrt{n}}, \delta_0, \varepsilon_0, m_0, \mathcal{F}, \mathcal{B})$ agnostic weak learner for any $m_0 \geq \lceil \frac{8 \ln(2|\mathcal{B}|/\delta_0)}{\varepsilon_0^2} \rceil$, notice that, by Hoeffding's inequality, for any $h \in \mathcal{B}$ it holds by the choice of $m_0$ that

$$\mathbb{P}_{\mathbf{S}}\left[ |\mathbb{E}_{(\mathbf{x},\mathbf{y}) \sim \mathcal{D}'}[h(\mathbf{x})\mathbf{y}] - \mathbb{E}_{(\mathbf{x},\mathbf{y}) \sim \mathbf{S}}[h(\mathbf{x})\mathbf{y}]| \leq \varepsilon_0/2 \right] \geq 1 - 2\exp\left(-\frac{m_0\varepsilon_0^2}{8}\right)$$

$$\geq 1 - \delta_0/|\mathcal{B}|.$$

So, by the union bound, it holds with probability at least $1 - \delta_0$ over $\mathbf{S} \sim \mathcal{D}'^{m_0}$ that for all $h \in \mathcal{B}$

$$|\mathbb{E}_{(\mathbf{x},\mathbf{y}) \sim \mathcal{D}'}[h(\mathbf{x})\mathbf{y}] - \mathbb{E}_{(\mathbf{x},\mathbf{y}) \sim \mathbf{S}}[h(\mathbf{x})\mathbf{y}]| \leq \varepsilon_0/2, \tag{49}$$

and, thus,

$$\begin{aligned}
\mathbb{E}_{(\mathbf{x},\mathbf{y}) \sim \mathcal{D}'}[\mathcal{W}(\mathbf{S})(\mathbf{x})\mathbf{y}] &\geq \mathbb{E}_{(\mathbf{x},\mathbf{y}) \sim \mathbf{S}}[\mathcal{W}(\mathbf{S})(\mathbf{x})\mathbf{y}] - \varepsilon_0/2 && \text{(by Eq. (49))} \\
&= \sup_{h \in \mathcal{B}} \mathbb{E}_{(\mathbf{x},\mathbf{y}) \sim \mathbf{S}}[h(\mathbf{x})\mathbf{y}] - \varepsilon_0/2 \\
&\geq \sup_{h \in \mathcal{B}} \mathbb{E}_{(\mathbf{x},\mathbf{y}) \sim \mathcal{D}'}[h(\mathbf{x})\mathbf{y}] - \varepsilon_0 && \text{(by Eq. (49))} \\
&\geq \frac{1}{\sqrt{n}} \sup_{f \in \mathcal{F}} \mathbb{E}_{(\mathbf{x},\mathbf{y}) \sim \mathcal{D}'}[f(\mathbf{x})\mathbf{y}] - \varepsilon_0,
\end{aligned}$$

where last inequality follows as long as for any distribution $\mathcal{D}' \in \Delta(\mathcal{X} \times \{\pm 1\})$ there exists $h \in \mathcal{B}$ satisfying Eq. (48).

To see why Eq. (49) holds let now $\mathcal{D}'$ be a probability distribution over $\mathcal{X} \times \{\pm 1\}$. We then have that for any $h \colon \mathcal{X} \to \{\pm 1\}$ that

$$\begin{aligned}
\mathbb{E}_{(\mathbf{x},\mathbf{y}) \sim \mathcal{D}'}[h(\mathbf{x})\mathbf{y}] &= \sum_{(x,y) \in \mathcal{X} \times \{\pm 1\}} h(x)y \cdot \mathcal{D}'(x,y) \\
&= \sum_{x \in \mathcal{X}} h(x)(\mathcal{D}'(x,1) - \mathcal{D}'(x,-1)). \tag{50}
\end{aligned}$$

If $\mathcal{D}'(x,1) - \mathcal{D}'(x,-1) = 0$ for all $x \in \mathcal{X}$, then $\mathbb{E}_{(\mathbf{x},\mathbf{y}) \sim \mathcal{D}'}[f(\mathbf{x})\mathbf{y}] = 0$ for any $f \in \{\pm 1\}^{\mathcal{X}}$, so the claim holds for any $h \in \mathcal{B}$. If $\mathcal{D}'(x,1) - \mathcal{D}'(x,-1) \neq 0$ for some $x \in \mathcal{X}$, we further write that

$$\begin{aligned}
\mathbb{E}_{(\mathbf{x},\mathbf{y}) \sim \mathcal{D}'}[h(\mathbf{x})\mathbf{y}] &= \sum_{x \in \mathcal{X}} h(x)\operatorname{sign}(\mathcal{D}'(x,1) - \mathcal{D}'(x,-1)) \cdot |\mathcal{D}'(x,1) - \mathcal{D}'(x,-1)| \\
&= \left( \sum_{x \in \mathcal{X}} h(x)\operatorname{sign}(\mathcal{D}'(x,1) - \mathcal{D}'(x,-1)) \frac{|\mathcal{D}'(x,1) - \mathcal{D}'(x,-1)|}{\sum_{x \in \mathcal{X}}|\mathcal{D}'(x,1) - \mathcal{D}'(x,-1)|} \right) \\
&\quad \cdot \sum_{x \in \mathcal{X}} |\mathcal{D}'(x,1) - \mathcal{D}'(x,-1)|, \tag{51}
\end{aligned}$$

where we recall that we define $\operatorname{sign}(0) = 1$. We now notice that $\operatorname{sign}(\mathcal{D}'(x,1) - \mathcal{D}'(x,-1))$ is indeed a mapping from $\mathcal{X}$ to $\{\pm 1\}$, and that $|\mathcal{D}'(x,1) - \mathcal{D}'(x,-1)|/\sum_{x \in \mathcal{X}}|\mathcal{D}'(x,1) - \mathcal{D}'(x,-1)|$ defines a probability distribution over $\mathcal{X}$. Thus, by Eq. (47), there exists $h \in \mathcal{B}$ such that

$$\sum_{x \in \mathcal{X}} h(x)\operatorname{sign}(\mathcal{D}'(x,1) - \mathcal{D}'(x,-1)) \frac{|\mathcal{D}'(x,1) - \mathcal{D}'(x,-1)|}{\sum_{x \in \mathcal{X}}|\mathcal{D}'(x,1) - \mathcal{D}'(x,-1)|} \geq \frac{1}{\sqrt{n}}.$$

Also, by Eq. (50), we have that $\operatorname{sign}(\mathcal{D}'(x,1) - \mathcal{D}'(x,-1))$ is a maximizer of $\mathbb{E}_{(\mathbf{x},\mathbf{y}) \sim \mathcal{D}'}[f(\mathbf{x})\mathbf{y}]$ over $f \in \mathcal{F}$ and is such that

$$\begin{aligned}
\mathbb{E}_{(\mathbf{x},\mathbf{y}) \sim \mathcal{D}'}[\operatorname{sign}(\mathcal{D}'(x,1) - \mathcal{D}'(x,-1))(\mathbf{x})\mathbf{y}] &= \sum_{x \in \mathcal{X}} |\mathcal{D}'(x,1) - \mathcal{D}'(x,-1)| \\
&= \sup_{f \in \mathcal{F}} \mathbb{E}_{(\mathbf{x},\mathbf{y}) \sim \mathcal{D}'}[f(\mathbf{x})\mathbf{y}].
\end{aligned}$$

Combining these two observations we conclude from Eq. (51) that there exists $h \in \mathcal{B}$ such that

$$\mathbb{E}_{(\mathbf{x},\mathbf{y}) \sim \mathcal{D}'}[h(\mathbf{x})\mathbf{y}] \geq \frac{1}{\sqrt{n}} \sup_{f \in \mathcal{F}} \mathbb{E}_{(\mathbf{x},\mathbf{y}) \sim \mathcal{D}'}[f(\mathbf{x})\mathbf{y}].$$

as claimed, which combined with the case where $\mathcal{D}'(x,1) - \mathcal{D}'(x,-1) = 0$ for all $x \in \mathcal{X}$ concludes the proof. $\qquad\square$

With the proof of Lemma E.2 done we now give the proof of Lemma E.3, for completeness. To this end we need the following two technical lemmas.

**Lemma E.6** (Shalev-Shwartz and Ben-David [2014], page 422, Lemma B.1). *Let $\mathbf{Z}$ be a random variable that takes values in $[0,1]$. Assume that $\mathbb{E}[\mathbf{Z}] = \mu$. Then, for any $a \in (0,1)$,*

$$\mathbb{P}[\mathbf{Z} > 1 - a] \geq \frac{\mu - (1-a)}{a}.$$

*This also implies that for every $a \in (0,1)$,*

$$\mathbb{P}[\mathbf{Z} > a] \geq \frac{\mu - a}{1 - a} \geq \mu - a.$$

**Lemma E.7** (Shalev-Shwartz and Ben-David [2014], page 428, Lemma B.11). *Let $\mathbf{X}$ be a $(m,p)$ binomial variable, i.e., $\mathbf{X} = \sum_{i=1}^{m} \mathbf{Z}_i$, where $\mathbf{Z}_i \in \{0,1\}$ are i.i.d. with $\mathbb{E}[\mathbf{Z}_i] = p$, and assume that $p = (1 - \epsilon)/2$. Then,*

$$\mathbb{P}[\mathbf{X} \geq m/2] \geq \frac{1}{2}\left(1 - \sqrt{1 - \exp\left(-\frac{m\epsilon^2}{1 - \epsilon^2}\right)}\right).$$

We now give the proof of Lemma E.4.

*Proof of Lemma E.4.* Consider a sequence $x_1, \ldots, x_d$ which is shattered by the function class $\mathcal{F}$, we will in the following, for convenience consider, the enumeration of these points as the universe $[d]$, i.e., $i \in [d]$ is $x_i$. Furthermore, let $b \in \{\pm 1\}^d$, and for each such $b$ consider a distribution $\mathcal{D}_b$ on $[d] \times \{\pm 1\}$, where $p$ in the following is strictly less than $1/(d-1)$,

$$\mathcal{D}_b(x,y) = \begin{cases} (1/2 + c)p & \text{if } y = b_i, x \in [d-1] \\ (1/2 - c)p & \text{if } y = -b_i, x \in [d-1] \\ 1 - (d-1)p & \text{if } y = 1, x = d \end{cases}$$

that is any point $i$ in $[d-1]$ is chosen with probability $p$ and with probability $1/2 + c$ it gets the same label as $b_i$ or with probability $1/2 - c$ it gets the label $-b_i$. We can see drawing a sample from the above distribution as follows: We first draw a random point $\mathbf{X}$ from $[d]$, where $\mathbf{X}$ is equal to $x$ for $x \in [d-1]$ with probability $p$ and $\mathbf{X}$ is equal to $d$ with probability $1 - (d-1)p$, (for later convince let this distribution over $[d]$ be denoted by $\mathcal{D}$). Furthermore, we draw a uniform random variable $\mathbf{U}$, in $[0,1]$ and let

$$\mathbf{Y} = \in d\{\mathbf{X} = d\} + \sum_{j=1}^{d-1} 2 \in d\{\mathbf{X} = j\}(\in d\{\mathbf{U} \leq 1/2 + cb_j\} - 1/2)$$

$$= \ell_b(\mathbf{X}, \mathbf{U}).$$

We notice the loss of any $f$ is given by

$$\mathcal{L}_{\mathcal{D}_b}(f) = \sum_{i=1}^{d} \sum_{y \in \{\pm 1\}} \mathcal{D}_b(i,y) \cdot \in d\{f(i) \neq y\}$$

$$= p\left(\sum_{i=1}^{d-1} (1/2 + c) \cdot \in d\{f(i) \neq b_i\} + (1/2 - c) \cdot \in d\{f(i) \neq -b_i\}\right)$$

$$+ (1 - (d-1)p) \cdot \in d\{f(d) \neq 1\}. \tag{52}$$

Thus, we see that a hypothesis in $\mathcal{F}$ that evaluates to $b$ on $[d-1]$ and $1$ on $d$ will have the smallest possible loss (notice that such a hypothesis in $\mathcal{F}$ exists since the hypothesis class shatters $x_1, \ldots, x_d$), being equal to $p(d-1)(1/2-c)$, thus let $f_b$ be such a hypothesis in $\mathcal{F}$, for a given $b \in \{\pm 1\}^d$, minimizing the loss function $\mathcal{L}_{\mathcal{D}_b}$. We will set $p = \frac{L}{(d-1)(1/2-c)}$, such that the loss of $f_b$ is equal to $L$. We notice that this implies that we have to set $c$ small enough such that $p < 1/(d-1)$, that is we have to have that $0 < L/(1/2-c) < 1$. To this end, we will set $c = \sqrt{\frac{d}{64mL}}$, which satisfies the condition $c < 1/2$ since $m \geq d/(L(1/2-L)^2)$, implying that $L/(1/2-c) < 1$ is equivalent to $L < 1/2 - c = 1/2 - \sqrt{\frac{d}{64mL}}$, since $m \geq d/(L(1/2-L)^2)$ we have that $1/2 - \sqrt{\frac{d}{64mL}} \geq 1/2 - (1/2-L)\sqrt{\frac{1}{32}} > L$, thus for these values of $c$, $p$ and $m \geq d/(L(1/2-L)^2)$ we have that $p < 1/(d-1)$. Now using the expression of Eq. (52) we get that the excess risk of any $f$ to $f_b$ is lower bounded as follows,

$$\mathcal{L}_{\mathcal{D}_b}(f) - \mathcal{L}_{\mathcal{D}_b}(f_b) \geq 2pc \sum_{i=1}^{d-1} \in d\{f(i) \neq f_b(i)\}.$$

this also implies that given a sample $\mathbf{S} = (\mathbf{X}_1, \mathbf{Y}_1), \ldots, (\mathbf{X}_m, \mathbf{Y}_m)$ drawn from $\mathcal{D}_b$ we have for any learning algorithm $\mathcal{A}$ that

$$\mathcal{L}_{\mathcal{D}_b}(\mathcal{A}(\mathbf{S})) - \mathcal{L}_{\mathcal{D}_b}(f_b) \geq 2pc \sum_{i=1}^{d-1} \in d\{\mathcal{A}(\mathbf{S})(i) \neq f_b(i)\}. \tag{53}$$

We will now show that

$$\mathbb{E}_{\mathbf{b} \sim \{\pm 1\}^d} \left[ \mathbb{E}_{\mathbf{S} \sim \mathcal{D}_{\mathbf{b}}^m} \left[ 2pc \sum_{i=1}^{d-1} \in d\{\mathcal{A}(\mathbf{S})(i) \neq f_{\mathbf{b}}(i)\} \right] \right] \geq \frac{2pcd}{25}. \tag{54}$$

We notice that since $2pc \sum_{i=1}^{d-1} \in d\{\mathcal{A}(\mathbf{S})(i) \neq f_{\mathbf{b}}(i)\} \leq 2pcd$, and is non negative, we have that for $a < 2pcd$ an application of Lemma E.6 gives us that

$$\mathbb{E}_{\mathbf{b} \sim \{\pm 1\}^d} \left[ \mathbb{P}_{\mathbf{S} \sim \mathcal{D}_{\mathbf{b}}^m} \left[ 2pc \sum_{i=1}^{d-1} \in d\{\mathcal{A}(\mathbf{S})(i) \neq f_{\mathbf{b}}(i)\} \geq a \right] \right]$$

$$= \mathbb{E}_{\mathbf{b} \sim \{\pm 1\}^d} \left[ \mathbb{P}_{\mathbf{S} \sim \mathcal{D}_{\mathbf{b}}^m} \left[ \frac{2pc \sum_{i=1}^{d-1} \in d\{\mathcal{A}(\mathbf{S})(i) \neq f_{\mathbf{b}}(i)\}}{2pcd} \geq \frac{a}{2pcd} \right] \right]$$

$$\geq \mathbb{E}_{\mathbf{b} \sim \{\pm 1\}^d} \left[ \mathbb{E}_{\mathbf{S} \sim \mathcal{D}_{\mathbf{b}}^m} \left[ 2pc \sum_{i=1}^{d-1} \in d\{\mathcal{A}(\mathbf{S})(i) \neq f_{\mathbf{b}}(i)\} \right] \right] / (2pcd) - \frac{a}{2pcd}$$

$$\geq \frac{1}{25} - \frac{a}{2pcd},$$

where we in the last inequality have used the lower bound of Eq. (54), thus setting $a = 2pcd/50$ we get that

$$\mathbb{E}_{\mathbf{b} \sim \{\pm 1\}^d} \left[ \mathbb{P}_{\mathbf{S} \sim \mathcal{D}_{\mathbf{b}}^m} \left[ \mathcal{L}_{\mathcal{D}_{\mathbf{b}}}(\mathcal{A}(\mathbf{S})) - \mathcal{L}_{\mathcal{D}_{\mathbf{b}}}(f_{\mathbf{b}}) \geq 2pcd/50 \right] \right]$$

$$\geq \mathbb{E}_{\mathbf{b} \sim \{\pm 1\}^d} \left[ \mathbb{P}_{\mathbf{S} \sim \mathcal{D}_{\mathbf{b}}^m} \left[ 2pc \sum_{i=1}^{d-1} \in d\{\mathcal{A}(\mathbf{S})(i) \neq f_{\mathbf{b}}(i)\} \geq 2pcd/50 \right] \right]$$

$$\geq \frac{1}{25} - \frac{1}{50}$$

$$= \frac{1}{50},$$

implying that there exists $b \in \{\pm 1\}^d$ such that

$$\mathbb{P}_{\mathbf{S} \sim \mathcal{D}_b^m} \left[ \mathcal{L}_{\mathcal{D}_b}(\mathcal{A}(\mathbf{S})) - \mathcal{L}_{\mathcal{D}_b}(f_b) \geq 2pcd/50 \right] \geq \frac{1}{50}.$$

Now using that $p = \frac{L}{(d-1)(1/2-c)}$, $c = \sqrt{\frac{d}{64mL}}$ we get that

$$pcd = \frac{L}{(d-1)(1/2-c)} \cdot \sqrt{\frac{d}{64mL}} \cdot d$$

$$\geq \sqrt{\frac{dL}{16n}},$$

where the first inequality follows from $d \geq 2$ so $1/(d-1) \geq 1/d$ and $1/(1/2-c) \geq 2$. Thus, we conclude that there exists $b \in \{\pm 1\}^d$ such that

$$\mathbb{P}_{\mathbf{S} \sim \mathcal{D}_b^m} \left[ \mathcal{L}_{\mathcal{D}_b}(\mathcal{A}(\mathbf{S})) - \mathcal{L}_{\mathcal{D}_b}(f_b) \geq \frac{2}{50} \sqrt{\frac{dL}{16n}} \right] \geq \frac{1}{50},$$

as claimed.

Thus, we now show Eq. (54) that is

$$\mathbb{E}_{\mathbf{b} \sim \{\pm 1\}^d} \left[ \mathbb{E}_{\mathbf{S} \sim \mathcal{D}_{\mathbf{b}}^m} \left[ 2pc \sum_{i=1}^{d-1} \in d\{\mathcal{A}(\mathbf{S})(i) \neq f_{\mathbf{b}}(i)\} \right] \right] \geq \frac{2pcd}{25}.$$

We now use that $\mathbf{S} \sim \mathcal{D}_b^m$ has the same distribution as

$$\mathbf{S} = ((\mathbf{X}_1, \mathbf{Y}_1), \dots, (\mathbf{X}_m, \mathbf{Y}_m))$$

$$\stackrel{\text{distribution}}{=} ((\mathbf{X}_1, \ell_b(\mathbf{X}_1, \mathbf{U}_1)), \dots, (\mathbf{X}_m, \ell_b(\mathbf{X}_m, \mathbf{U}_m))) := (\mathbf{X}, \ell_b(\mathbf{X}, \mathbf{U})),$$

where $\mathbf{X} \sim \mathcal{D}^m$ and $\mathbf{U} \sim [0,1]^m$, and we use the above entrywise notation for $(\mathbf{X}, \ell_b(\mathbf{X}, \mathbf{U}))$. Using the above and Eq. (53) we have that

$$\mathbb{E}_{\mathbf{S} \sim \mathcal{D}_b^m} [\mathcal{L}_{\mathcal{D}_b}(\mathcal{A}(\mathbf{S})) - \mathcal{L}_{\mathcal{D}_b}(f_b)]$$

$$\geq 2pc \sum_{i=1}^{d-1} \mathbb{E}_{\mathbf{X} \sim \mathcal{D}^m} \left[ \mathbb{E}_{\mathbf{U} \sim [0,1]^m} \left[ \in d\{\mathcal{A}((\mathbf{X}, \ell_b(\mathbf{X}, \mathbf{U})))(i) \neq f_b(i)\} \right] \right],$$

and taking expectation over $\mathbf{b} \sim \{\pm 1\}^d$ we get that

$$\mathbb{E}_{\mathbf{b} \sim \{\pm 1\}^d} \left[ \mathbb{E}_{\mathbf{S} \sim \mathcal{D}_{\mathbf{b}}^m} [\mathcal{L}_{\mathcal{D}_{\mathbf{b}}}(\mathcal{A}(\mathbf{S})) - \mathcal{L}_{\mathcal{D}_{\mathbf{b}}}(f_{\mathbf{b}})] \right] \qquad (55)$$

$$\geq 2pc \sum_{i=1}^{d-1} \mathbb{E}_{\mathbf{X} \sim \mathcal{D}^m} \left[ \mathbb{E}_{\mathbf{b} \sim \{\pm 1\}^d} \left[ \mathbb{E}_{\mathbf{U} \sim [0,1]^m} \left[ \in d\{\mathcal{A}((\mathbf{X}, \ell_{\mathbf{b}}(\mathbf{X}, \mathbf{U})))(i) \neq f_{\mathbf{b}}(i)\} \right] \right] \right]$$

$$= 2pc \sum_{i=1}^{d-1} \mathbb{E}_{\mathbf{X} \sim \mathcal{D}^m} \left[ \mathbb{E}_{\mathbf{b}_{-i} \sim \{\pm 1\}^{d-1}} \left[ \mathbb{E}_{\mathbf{b}_i \sim \{\pm 1\}} \left[ \mathbb{E}_{\mathbf{U} \sim [0,1]^m} \left[ \in d\{\mathcal{A}((\mathbf{X}, \ell_{\mathbf{b}}(\mathbf{X}, \mathbf{U})))(i) \neq \mathbf{b}_i\} \right] \right] \right] \right],$$

where we use $\mathbf{b}_i$ to denote the $i^{\text{th}}$ entry of $\mathbf{b}$ and $\mathbf{b}_{-i}$ the remaining $d-1$ entries. We now bound each term in this sum.

To that end we now consider any realization of $\mathbf{X} = x$ and $\mathbf{b}_{-i} = b_{-i}$. Furthermore let $J_i = (j_1^i, \dots, j_k^i)$, such that $j_1^i < \dots < j_k^i$ and $x_{j_1^i}, \dots, x_{j_k^i} = i$, the indexes where $x$ is equal to $i$. We recall that $(x, \ell_{\mathbf{b}}(x, \mathbf{U})) = ((x_1, \ell_{(b_1, \dots, \mathbf{b}_i, \dots, b_d)}(x_1, \mathbf{U}_1)), \dots, (x_m, \ell_{(b_1, \dots, \mathbf{b}_i, \dots, b_d)}(x_m, \mathbf{U}_m)))$, and let $(x, \ell_{\mathbf{b}}(x, \mathbf{U}))_{J_i} = ((x_{j_1^i}, \ell_{(b_1, \dots, \mathbf{b}_i, \dots, b_d)}(x_{j_1^i}, \mathbf{U}_{j_1^i})), \dots, (x_{j_k^i}, \ell_{(b_1, \dots, \mathbf{b}_i, \dots, b_d)}(x_{j_k^i}, \mathbf{U}_{j_k^i})))$ and $(x, \ell_{\mathbf{b}}(x, \mathbf{U}))_{-J_i}$ be the remaining entries of $(x, \ell_{\mathbf{b}}(x, \mathbf{U}))$. In words, $(x, \ell_{\mathbf{b}}(x, \mathbf{U}))_{J_i}$ are the entries of $(x, \ell_{\mathbf{b}}(x, \mathbf{U}))$ that are has $x_j$ equal to $i$ and $(x, \ell_{\mathbf{b}}(x, \mathbf{U}))_{-J_i}$ are the remaining entries of $(x, \ell_{\mathbf{b}}(x, \mathbf{U}))$ that are not equal to $i$. By the definition of $\ell_{(b_1, \dots, \mathbf{b}_i, \dots, b_d)}$ we have that $\ell_{(b_1, \dots, \mathbf{b}_i, \dots, b_d)}(x_{j_t^i}, \mathbf{U}_{j_t^i}) = 2(\in d\{\mathbf{U}_{j_t^i} \leq 1/2 + c\mathbf{b}_i\} - 1/2) := \ell'(\mathbf{U}_{j_t^i}, \mathbf{b}_i)$ for $t = 1, \dots, k$ so only a function of $\mathbf{U}_{j_t^i}$ and $\mathbf{b}_i$. Furthermore by the definition of $\ell_{(b_1, \dots, \mathbf{b}_i, b_d)}(x, \mathbf{U})_{-J_i} = \ell''_{b_{-i}}(\mathbf{U}_{-J_i})$ is only a function of $\mathbf{U}_{-J_i}$ (the coordinates of $\mathbf{U}$ not with index in $J_i$) and $b_{-i}$. Using these observa-

tion we get that

$$\mathbb{E}_{\mathbf{U}\sim[0,1]^m}\left[\in d\{\mathcal{A}((x,\ell_{(b_1,\ldots,\mathbf{b}_i,\ldots,b_d)}(x,\mathbf{U})))(i)\neq\mathbf{b}_i\}\right]$$

$$=\sum_{y\in\{\pm1\}^m}\mathbb{P}_{\mathbf{U}\sim[0,1]^m}\left[(\ell_{(b_1,\ldots,\mathbf{b}_i,\ldots,b_d)}(x,\mathbf{U}))=y\right]\cdot\in d\{\mathcal{A}((x,y))(i)\neq\mathbf{b}_i\}$$

$$=\sum_{y\in\{\pm1\}^m}\mathbb{P}_{\mathbf{U}\sim[0,1]^{|J_i|}}\left[\ell'(\mathbf{U}_{J_i},\mathbf{b}_i)=y_{J_i}\right]\cdot\mathbb{P}_{\mathbf{U}\sim[0,1]^{m-|J_i|}}\left[\in d\{\ell''_{b_{-i}}(\mathbf{U}_{-J_i})\}=y_{-J_i}\right]$$

$$\cdot\in d\{\mathcal{A}((x,y))(i)\neq\mathbf{b}_i\}$$

$$=\sum_{y_{-J_i}\in\{\pm1\}^{d-|J_i|}}\mathbb{P}_{\mathbf{U}\sim[0,1]^{m-|J_i|}}\left[\ell''_{b_{-i}}(\mathbf{U}_{-J_i})=y_{-J_i}\right]$$

$$\cdot\sum_{y_{J_i}\in\{\pm1\}^{|J_i|}}\mathbb{P}_{\mathbf{U}\sim[0,1]^{|J_i|}}\left[\ell'(\mathbf{U}_{J_i},\mathbf{b}_i)=y_{J_i}\right]\cdot\in d\{\mathcal{A}((x,y))(i)\neq\mathbf{b}_i\}. \tag{56}$$

We notice that the first sum in the above expression is independent of $\mathbf{b}_i$ thus we focus on the sum over $y_{J_i}$. To this end consider any $y_{-J_i}$ we then have when taking expectation of $\mathbf{b}_i$ over the second sum in the above that

$$\mathbb{E}_{\mathbf{b}_i\sim\{\pm1\}}\left[\sum_{y_{J_i}\in\{\pm1\}^{|J_i|}}\mathbb{P}_{\mathbf{U}\sim[0,1]^{|J_i|}}\left[\ell'(\mathbf{U}_{J_i},\mathbf{b}_i)=y_{J_i}\right]\cdot\in d\{\mathcal{A}((x,y))(i)\neq\mathbf{b}_i\}\right]$$

$$=\frac{1}{2}\sum_{y_{J_i}\in\{\pm1\}^{|J_i|}}\mathbb{P}_{\mathbf{U}\sim[0,1]^{|J_i|}}\left[\ell'(\mathbf{U}_{J_i},1)=y_{J_i}\right]\cdot\in d\{\mathcal{A}((x,y))(i)\neq1\}$$

$$+\frac{1}{2}\sum_{y_{J_i}\in\{\pm1\}^{|J_i|}}\mathbb{P}_{\mathbf{U}\sim[0,1]^{|J_i|}}\left[\ell'(\mathbf{U}_{J_i},1)=y_{J_i}\right]\cdot\in d\{\mathcal{A}((x,y))(i)\neq1\}$$

We have by independence of $\mathbf{U}_{j_1^i},\ldots,\mathbf{U}_{j_k^i}$ that

$$\mathbb{P}_{\mathbf{U}\sim[0,1]^{|J_i|}}\left[\ell'(\mathbf{U}_{J_i},1)=y_{J_i}\right]=\prod_{t=1}^k\mathbb{P}_{\mathbf{U}_{j_t^i}\sim[0,1]}\left[\ell'(\mathbf{U}_{j_t^i},1)=y_{j_t^i}\right]$$

$$=\prod_{t=1}^k(1/2+c)^{\in d\{y_{j_t^i}=1\}}\cdot(1/2-c)^{\in d\{y_{j_t^i}=-1\}}$$

$$=(1/2+c)^{\sum_{t=1}^k\in d\{y_{j_t^i}=1\}}\cdot(1/2-c)^{\sum_{t=1}^k\in d\{y_{j_t^i}=-1\}},$$

and similarly we have that

$$\mathbb{P}_{\mathbf{U}\sim[0,1]^{|J_i|}}\left[\ell'(\mathbf{U}_{J_i},-1)=y_{J_i}\right]=\prod_{t=1}^k\mathbb{P}_{\mathbf{U}_{j_t^i}\sim[0,1]}\left[\ell'(\mathbf{U}_{j_t^i},-1)=y_{j_t^i}\right]$$

$$=\prod_{t=1}^k(1/2-c)^{\in d\{y_{j_t^i}=1\}}\cdot(1/2+c)^{\in d\{y_{j_t^i}=-1\}}$$

$$=(1/2-c)^{\sum_{t=1}^k\in d\{y_{j_t^i}=1\}}\cdot(1/2+c)^{\sum_{t=1}^k\in d\{y_{j_t^i}=-1\}},$$

implying that if $\sum_{t=1}^k\in d\{y_{j_t^i}=1\}\geq\sum_{t=1}^k\in d\{y_{j_t^i}=-1\}$ or equivalently $\mathrm{sign}(\sum_{t=1}^k y_{j_t^i})=1$ (we take $\mathrm{sign}(0)=1$) then

$$\mathbb{P}_{\mathbf{U}\sim[0,1]^{|J_i|}}\left[\ell'(\mathbf{U}_{J_i},1)=y_{J_i}\right]\geq\mathbb{P}_{\mathbf{U}\sim[0,1]^{|J_i|}}\left[\ell'(\mathbf{U}_{J_i},-1)=y_{J_i}\right]$$

and if $\sum_{t=1}^k\in d\{y_{j_t^i}=1\}<\sum_{t=1}^k\in d\{y_{j_t^i}=-1\}$ or equivalently $\mathrm{sign}(\sum_{t=1}^k y_{j_t^i})=-1$ then

$$\mathbb{P}_{\mathbf{U}\sim[0,1]^{|J_i|}}\left[\ell'(\mathbf{U}_{J_i},1)=y_{J_i}\right]<\mathbb{P}_{\mathbf{U}\sim[0,1]^{|J_i|}}\left[\ell'(\mathbf{U}_{J_i},-1)=y_{J_i}\right]$$

whereby we conclude that

$$\mathbb{E}_{\mathbf{b}_i \sim \{\pm 1\}} \left[ \sum_{y_{J_i} \in \{\pm 1\}^{|J_i|}} \mathbb{P}_{\mathbf{U} \sim [0,1]^{|J_i|}} \left[ \ell'(\mathbf{U}_{J_i}, \mathbf{b}_i) = y_{J_i} \right] \in d\{\mathcal{A}((x,y))(i) \neq \mathbf{b}_i\} \right]$$

$$= \frac{1}{2} \sum_{y_{J_i} \in \{\pm 1\}^{|J_i|}} \mathbb{P}_{\mathbf{U} \sim [0,1]^{|J_i|}} \left[ \ell'(\mathbf{U}_{J_i}, 1) = y_{J_i} \right] \cdot \in d\{\mathcal{A}((x,y))(i) \neq 1\}$$

$$+ \frac{1}{2} \sum_{y_{J_i} \in \{\pm 1\}^{|J_i|}} \mathbb{P}_{\mathbf{U} \sim [0,1]^{|J_i|}} \left[ \ell'(\mathbf{U}_{J_i}, -1) = y_{J_i} \right] \cdot \in d\{\mathcal{A}((x,y))(i) \neq -1\}$$

$$\geq \frac{1}{2} \sum_{y_{J_i} \in \{\pm 1\}^{|J_i|}} \mathbb{P}_{\mathbf{U} \sim [0,1]^{|J_i|}} \left[ \ell'(\mathbf{U}_{J_i}, 1) = y_{J_i} \right] \cdot \in d\left\{ \mathrm{sign}\left( \sum_{t=1}^{k} y_{j_t^i} \right) \neq 1 \right\}$$

$$+ \frac{1}{2} \sum_{y_{J_i} \in \{\pm 1\}^{|J_i|}} \mathbb{P}_{\mathbf{U} \sim [0,1]^{|J_i|}} \left[ \ell'(\mathbf{U}_{J_i}, -1) = y_{J_i} \right] \cdot \in d\left\{ \mathrm{sign}\left( \sum_{t=1}^{k} y_{j_t^i} \right) \neq -1 \right\}$$

$$= \mathbb{E}_{\mathbf{b}_i \sim \{\pm 1\}} \left[ \sum_{y_{J_i} \in \{\pm 1\}^{|J_i|}} \mathbb{P}_{\mathbf{U} \sim [0,1]^{|J_i|}} \left[ \ell'(\mathbf{U}_{J_i}, \mathbf{b}_i) = y_{J_i} \right] \cdot \in d\left\{ \mathrm{sign}\left( \sum_{t=1}^{k} y_{j_t^i} \right) \neq \mathbf{b}_i \right\} \right],$$

which furthermore by Eq. (56) implies that

$$\mathbb{E}_{\mathbf{b}_i \sim \{\pm 1\}} \mathbb{E}_{\mathbf{U} \sim [0,1]^m} \left[ \in d\left\{ \mathcal{A}((x, \ell_{(b_1,\ldots,\mathbf{b}_i,\ldots,b_d)}(x, \mathbf{U})))(i) \neq \mathbf{b}_i \right\} \right]$$

$$\geq \mathbb{E}_{\mathbf{b}_i \sim \{\pm 1\}} \left[ \sum_{y_{J_i} \in \{\pm 1\}^{|J_i|}} \mathbb{P}_{\mathbf{U} \sim [0,1]^{|J_i|}} \left[ \ell'(\mathbf{U}_{J_i}, \mathbf{b}_i) = y_{J_i} \right] \cdot \in d\left\{ \mathrm{sign}\left( \sum_{t=1}^{k} y_{j_t^i} \right) \neq \mathbf{b}_i \right\} \right]$$

$$= \mathbb{E}_{\mathbf{b}_i \sim \{\pm 1\}} \left[ \mathbb{E}_{\mathbf{U} \sim [0,1]^{|J_i|}} \left[ \in d\left\{ \mathrm{sign}\left( \sum_{t=1}^{k} \ell'(\mathbf{U}_{j_t^i}, \mathbf{b}_i) \right) \neq \mathbf{b}_i \right\} \right] \right],$$

where the sum over $y_{-J_i}$ becomes one since $\in d\left\{ \mathrm{sign}\left( \sum_{t=1}^{k} y_{j_t^i} \right) \neq \mathbf{b}_i \right\}$, does not depend on $y_{-J_i}$ and thus we can take it out of the sum over $y_{-J_i}$.

Now in the case that $|J_i| = 0$ we have that $\mathrm{sign}\left( \sum_{t=1}^{k} \ell'(\mathbf{U}_{j_t^i}, \mathbf{b}_i) \right) = \mathrm{sign}(0) = 1$ and thus we have the above is $1/2$. Now in the case that $|J_i| > 0$ we have that

$$\mathbb{E}_{\mathbf{b}_i \sim \{\pm 1\}} \left[ \mathbb{E}_{\mathbf{U} \sim [0,1]^{|J_i|}} \left[ \in d\left\{ \mathrm{sign}\left( \sum_{t=1}^{k} \ell'(\mathbf{U}_{j_t^i}, \mathbf{b}_i) \right) \neq \mathbf{b}_i \right\} \right] \right]$$

$$= \frac{1}{2} \left( \mathbb{P}_{\mathbf{U} \sim [0,1]^{|J_i|}} \left[ \mathrm{sign}\left( \sum_{t=1}^{k} \ell'(\mathbf{U}_{j_t^i}, 1) \right) \neq 1 \right] + \mathbb{P}_{\mathbf{U} \sim [0,1]^{|J_i|}} \left[ \mathrm{sign}\left( \sum_{t=1}^{k} \ell'(\mathbf{U}_{j_t^i}, -1) \right) \neq -1 \right] \right).$$

By the definition of $\ell'(\mathbf{U}_{j_t^i}, b_i) := 2(\in d\{\mathbf{U}_{j_t^i} \leq 1/2 + cb_i\} - 1/2)$ we have that the event $\mathrm{sign}\left( \sum_{t=1}^{k} \ell'(\mathbf{U}_{j_t^i}, -1) \right) \neq -1$ happens when $k/2 \leq |\{t : \ell'(\mathbf{U}_{j_t^i}, -1) = 1\}|$, where the less than or equal to is due to us taking $\mathrm{sign}(0) = 1$. We notice that $|\{t : \ell'(\mathbf{U}_{j_t^i}, -1) = 1\}|$ has a binomial distribution with $k$ trials and success probability $1/2 - c$. Thus, we have by Lemma E.6

$$\mathbb{P}_{\mathbf{U} \sim [0,1]^{|J_i|}} \left[ \mathrm{sign}\left( \sum_{t=1}^{k} \ell'(\mathbf{U}_{j_t^i}, -1) \right) \neq -1 \right] \geq \mathbb{P}_{\mathbf{U} \sim [0,1]^{|J_i|}} \left[ \{t : \ell'(\mathbf{U}_{j_t^i}, -1) = 1\} \geq k/2 \right]$$

$$\geq \frac{1}{2} \left( 1 - \sqrt{1 - \exp\left( -|J_i|(2c)^2/(1 - (2c)^2) \right)} \right)$$

$$\geq \frac{1}{2} \left( 1 - \sqrt{|J_i|(2c)^2/(1 - (2c)^2)} \right),$$

where the last inequality follows from $\exp(x) \geq 1 + x$ for all $x \in \mathbb{R}$. Which furthermore implies that

$$\mathbb{E}_{\mathbf{b}_i \sim \{\pm 1\}} \left[ \mathbb{E}_{\mathbf{U} \sim [0,1]^{|J_i|}} \left[ \in d \left\{ \text{sign}\left( \sum_{t=1}^{k} \ell'(\mathbf{U}_{j_t^i}, \mathbf{b}_i) \right) \neq \mathbf{b}_i \right\} \right] \right]$$

$$= \frac{1}{4}(1 - \sqrt{|J_i|(2c)^2/(1-(2c)^2)}),$$

which also holds for $|J_i| = 0$ since $\mathbb{P}_{\mathbf{U} \sim [0,1]^{|J_i|}} \left[ \text{sign}\left( \sum_{t=1}^{k} \ell'(\mathbf{U}_{j_t^i}, -1) \right) \neq -1 \right] = 1/2$ in this case.

Thus, by the above we showed that for any realization $b_{-i}$ of $\mathbf{b}_{-i}$ and $x$ of $\mathbf{X}$ we have that

$$\mathbb{E}_{\mathbf{b}_i \sim \{\pm 1\}} \left[ \mathbb{E}_{\mathbf{U} \sim [0,1]^m} \left[ \in d \left\{ \mathcal{A}((\mathbf{X}, \ell_{\mathbf{b}}(\mathbf{X}, \mathbf{U})))(i) \neq \mathbf{b}_i \right\} \right] \right]$$

$$\geq \mathbb{E}_{\mathbf{b}_i \sim \{\pm 1\}} \mathbb{E}_{\mathbf{U} \sim [0,1]^m} \left[ \in d \left\{ \mathcal{A}((x, \ell_{(b_1,\ldots,\mathbf{b}_i,\ldots,b_d)}(x, \mathbf{U})))(i) \neq \mathbf{b}_i \right\} \right]$$

$$\geq \frac{1}{4}(1 - \sqrt{|J_i|(2c)^2/(1-(2c)^2)}).$$

Now using this and plugging into Eq. (55) we get the following lower bounded

$$\mathbb{E}_{\mathbf{b} \sim \{\pm 1\}^d} \left[ \mathbb{E}_{\mathbf{S} \sim \mathcal{D}_{\mathbf{b}}^m} \left[ \mathcal{L}_{\mathcal{D}_{\mathbf{b}}}(\mathcal{A}(\mathbf{S})) - \mathcal{L}_{\mathcal{D}_{\mathbf{b}}}(f_{\mathbf{b}}) \right] \right] \tag{57}$$

$$\geq 2pc \sum_{i=1}^{d-1} \mathbb{E}_{\mathbf{X} \sim \mathcal{D}^m} \left[ \frac{1}{4}(1 - \sqrt{|J_i|(2c)^2/(1-(2c)^2)}) \right],$$

where we recall that $\mathbf{J}_i$ is the indexes of $\mathbf{X}$ that are equal to $i$. Now using that $\sqrt{\cdot}$ is a concave function and Jensen's inequality we have that

$$\mathbb{E}_{\mathbf{X} \sim \mathcal{D}^m} \left[ \frac{1}{4}(1 - \sqrt{|J_i|(2c)^2/(1-(2c)^2)}) \right] \geq \frac{1}{4} \left( 1 - \sqrt{\frac{(2c)^2}{(1-(2c)^2)} \cdot \mathbb{E}_{\mathbf{X} \sim \mathcal{D}^m}[|\mathbf{J}_i|]} \right)$$

$$= \frac{1}{4} \left( 1 - \sqrt{\frac{(2c)^2 \cdot m \cdot p}{(1-(2c)^2)}} \right),$$

where the last inequality follows from that $|\mathbf{J}_i| = \sum_{j=1}^{m} \in d\{\mathbf{X}_j = i\}$ and $p = \mathbb{P}_{\mathbf{X}_j \sim \mathcal{D}}[\mathbf{X}_j = i]$ for $i \in [d-1]$, and $j \in [m]$. Recalling that we had $p = \frac{L}{(d-1)(1/2-c)}$, and $c = \sqrt{\frac{d}{64mL}}$, which by $m \geq \frac{d}{L(1/2-L)^2}$ implies $c \leq \sqrt{1/64}$, we make the following calculations on the right hand side of the above to get that

$$(2c)^2 \cdot m \cdot p = \left( 2\sqrt{\frac{d}{64mL}} \right)^2 \cdot m \cdot \frac{L}{(d-1)(1/2-c)}$$

$$= \frac{1}{16} \frac{d}{(d-1)(1/2-c)}$$

$$\leq \frac{1}{8} \frac{1}{1/2 - \sqrt{1/64}},$$

where the last inequality follows from $d \geq 2$ so $d/(d-1) \leq 2$ and $c \leq \sqrt{1/64}$, furthermore since we have that $1/(1-(2c)^2) \leq \frac{1}{1-4/64}$ implying that

$$\frac{(2c)^2 \cdot m \cdot p}{1-(2c)^2} \leq \frac{1}{8} \cdot \frac{1}{1/2 - \sqrt{1/64}} \cdot \frac{1}{1-4/64} \leq \frac{2}{3},$$

so that

$$\mathbb{E}_{\mathbf{X} \sim \mathcal{D}^m} \left[ \frac{1}{4}(1 - \sqrt{|\mathbf{J}_i|(2c)^2/(1-(2c)^2)}) \right] \geq \frac{1}{4} \left( 1 - \sqrt{2/3} \right) \geq 1/25,$$

implying by Eq. (57) that we have shown that

$$\mathbb{E}_{\mathbf{b} \sim \{\pm 1\}^d} \left[ \mathbb{E}_{\mathbf{S} \sim \mathcal{D}_{\mathbf{b}}^m} \left[ \mathcal{L}_{\mathcal{D}_{\mathbf{b}}}(\mathcal{A}(\mathbf{S})) - \mathcal{L}_{\mathcal{D}_{\mathbf{b}}}(f_{\mathbf{b}}) \right] \right] \geq \frac{2pcd}{25},$$

which was the claim of Eq. (54), concluding the proof of Lemma E.4. $\qquad \square$

