# OpenReview forum: "Revisiting Agnostic Boosting"
_NeurIPS.cc/2025/Conference — NeurIPS 2025 poster_

### Official Review · Reviewer_tVkc · 2025-06-28

**Clarity:** 4
**Significance:** 4
**Originality:** 3
**Rating:** 5
**Confidence:** 4

**Summary:**

This paper revisits the classic problem of agnostic boosting. Leveraging recent techniques by Hopkins et al. [2024] , the authors propose a new agnostic boosting algorithm which, under some general assumptions, improves on the sample complexity of existing algorithms. Finally, the authors complement their upper bound with a near matching lower bound.

**Questions:**

In line 270, the authors justify the need for the filtering step by arguing that the size of $B_1$ is exponential in the sample size. However, instead of taking *all* possible labels of the training sample, it seems to me that one can just loop over all possible behaviors of the class $F$ on $x_1, ..., x_m$. If $F$ has finite VC, $B_1$ now has size polynomial in $m$ (compared to exponential). It would be great if the authors could comment on this and what sort of upper bounds this approach would yield.

**Ethical Concerns:**

["NO or VERY MINOR ethics concerns only"]

**Final Justification:**

I maintain my positive score as the authors have adequately answered my questions.

**Limitations:**

yes

**Quality:**

4

**Strengths And Weaknesses:**

Strengths:
- This paper is extremely well-written and organized. The authors do a great job at explaining their algorithm and providing the right balance between intuition and technical details.
- The results seem novel and provide a clear contribution to the literature on agnostic boosting. To the best of my knowledge, this is the first time that the "all possible labelings" approach of Hopkins et al. [2024] has been used for agnostic boosting.

Weaknesses: I don't have much to say beyond a few presentational choices.

- The Theorem statements in the introduction are a bit dense. I think it would be better to replace these with informal Theorem statements that capture only the point that is trying to be made, perhaps by dropping some of the quantifiers.
- The majority of the main text is dedicated to proving the upper bound in Theorem 1.3. However, in the introduction, Theorem 1.2 is presented before Theorem 1.3. I think it would be better to swap the order of these two Theorems and place more emphasis on Theorem 1.3. Or, I think it would be beneficial to include more intuition about Theorem 1.2 in the main text.
- As written, it is a bit difficult to contextualize the results of this paper with existing results in agnostic boosting. The authors attempt to do this in Section 1.1. I think it might be nice to include a table that compares your upper/lower bounds with existing results when the assumptions and weak learning definitions line up.

---

> ### Author Rebuttal · Authors · 2025-07-31
>
> We thank the reviewer for taking the time to read our work carefully.
>
> We appreciate the reviewers feedback on the presentation (Weaknesses section of the review), we address the comments in order.
>
> > The Theorem statements in the introduction are a bit dense. I think it would be better to replace these with informal Theorem statements that capture only the point that is trying to be made, perhaps by dropping some of the quantifiers.
>
> **Answer:** We will revisit our formulation to try to improve the clarity.
>
> > The majority of the main text is dedicated to proving the upper bound in Theorem 1.3. However, in the introduction, Theorem 1.2 is presented before Theorem 1.3. I think it would be better to swap the order of these two Theorems and place more emphasis on Theorem 1.3. Or, I think it would be beneficial to include more intuition about Theorem 1.2 in the main text.
>
> **Answer:**
> We actually switched the order of the two theorems.
> The idea was to have our Theorem 1.2 (sample complexity lower bound) immediately after the definition of agnostic weak learner since the main challenge of proving lower bounds in weak learning is to construct the weak learner itself.
> So, stating the lower bound right after the definition felt natural to us.
> However, with your feedback we will reconsider this decision.
>
> > As written, it is a bit difficult to contextualize the results of this paper with existing results in agnostic boosting. The authors attempt to do this in Section 1.1. I think it might be nice to include a table that compares your upper/lower bounds with existing results when the assumptions and weak learning definitions line up.
>
> **Answer:**
> The addition of a table summarizing the existing results was also suggested by **Reviewer aHg7**.
> As mentioned there, we deliberately avoided it to highlight the nuances that arise if one is fully rigorous when comparing the works.
> The goal was to sidestep even slight misinterpretations by keeping only explicit and detailed discussions in Section 1.1 (Related Works) and, more thoroughly, in Appendix A.
> Still, since both you and **Reviewer aHg7** have suggested the addition, if you maintain the preference after taking our reply into account, we would not object to adding such a table (accompanied by a clear remark of its limitations).
>
> Turning to the reviewer's (particularly thoughtful) question,
> > In line 270, the authors justify the need for the filtering step by arguing that the size of $B_1$ is exponential in the sample size. However, instead of taking all possible labels of the training sample, it seems to me that one can just loop over all possible behaviors of the class $F$ on $x_1, ..., x_m$. If $F$ has finite VC, $B_1$ now has size polynomial in $m$ (compared to exponential). It would be great if the authors could comment on this and what sort of upper bounds this approach would yield.
>
> we do not do this because the weak learning setup only assumes access to the hypothesis provided by the weak learner from the base class $\mathcal{B}$.
> So, neither $\mathcal{B}$ nor the reference class $\mathcal{F}$ is known to the learner, making it a powerful setup as, in particular, $\mathcal{F}$ needs only to exist.
>
> Now, if we granted the learner knowledge of $\mathcal{F}$, we could---as the reviewer pointed out---label the training set with all possible behaviors of the class $\mathcal{F}$ on the training set.
> By Sauer-Shelah, there are at most $O((m/d')^{d'})$ such possible patterns, where $d' = \mathrm{VC}(\mathcal{F})$ (notice that in our bounds use instead $\mathrm{VC}(\mathcal{B})$).
> Thus, the size of the bag $\mathcal{B}_1$ considered in the first step would also be $O((m/d')^{d'})$, and the argument in section 3.1 guaranteeing the existence of a classifier $v\_g$ in $\mathcal{B}\_1$ with loss close to the optimal classifier in $\mathcal{F}$ would still hold.
> With $\lvert\mathcal{B}\_1\rvert$ polynomial in the sample size, we would not need the next filtering step as foreseen of the reviewer: we could use a Hoeffding bound (or a Bernstein bound if wanting $\sqrt{\mathrm{err}(f^\star) \ldots}$) plus a union bound to guarantee that the classifiers in the bag $\mathcal{B}\_1$ has generalization error close to their empirical error up to $\pm \Theta\left(\sqrt{\frac{\ln(|\mathcal{B}\_{1}|/\delta)}{m}}\right) = \pm \Theta\left(\sqrt{\frac{d'\ln(m/d') + \ln(\delta)}{m}}\right)$,
> so the classifier $v\_{g}$ with the smallest empirical loss in $\mathcal{B}\_1$ would satisfy $\mathcal{L}\_{\mathcal{D}}(v\_{g}) = \mathrm{err}\_{\mathcal{D}}(f^{\star}) + O\left(\sqrt{\frac{d'\ln(m/d') + \ln(\delta)}{m}}\right) + \tilde{O}\left(\frac{d}{\gamma^2 m}\right)$.
>
> Also, even if $\mathcal{F}$ is known, there is no general way to efficiently obtain all possible labellings associated with it.
> Indeed, if there was, the work of Hopkins et al. (2024) would yield a general efficient agnostic learner, which is not possible.
>
>
> Realizable Learning is All You Need. Max Hopkins, Daniel M. Kane, Shachar Lovett, Gaurav Mahajan. 2024

---

> > ### Comment · Reviewer_tVkc · 2025-08-01
> >
> > I thank the authors for their response. I maintain my positive score.

---

### Official Review · Reviewer_jF1w · 2025-07-03

**Clarity:** 3
**Significance:** 2
**Originality:** 3
**Rating:** 4
**Confidence:** 1

**Summary:**

This paper presents a new agnostic boosting method with comprehensive theoretical analysis. The new algorithm can convert any non-trivial agnostic weak learner into a strong learner. The core contributions are (1) the proposed algorithm achieves a significantly improved sample complexity upper bound compared to prior work under very general assumptions. (2) the paper establishes a nearly matching lower bound on the sample complexity for agnostic boosting, effectively settling the statistical complexity of the problem up to logarithmic factors.

**Questions:**

1. Do you have any experiments, including synthetic ones, to show the proposed modified Adaboost in Algorithm 1 works in practice?

2. Can you further discuss how this work relates to / extends Hopkins et al. (2024) (in line 195)?

3. As one limitation is the algorithm's computational inefficiency steming from iterating over all possible re-labelings. Are there any foreseeable approaches or heuristics to make this algorithm (or a similar one) computationally feasible while substantially preserving its strong statistical guarantees?

**Ethical Concerns:**

["NO or VERY MINOR ethics concerns only"]

**Final Justification:**

I have read the author rebuttal and decided to keep the original score.

**Limitations:**

yes

**Quality:**

3

**Strengths And Weaknesses:**

Strengths:
1. Establishes nearly tight upper and lower bounds on the sample complexity of agnostic weak-to-strong learning, which advances the fundamental understanding of boosting in the challenging agnostic setting
2. The proposed algorithm and its analysis operate under remarkably mild and general assumptions

Weakness:
1. The proposed algorithm is computationally inefficient and could be hard to implement in practice.
2. There is a lack of experiments to demonstrate the practice usefulness of this method.

---

> ### Author Rebuttal · Authors · 2025-07-31
>
> We thank the reviewer for taking the time to read our paper and provide feedback.
>
> We address the reviewers questions in order:
> > There is a lack of experiments to demonstrate the practice usefulness of this method.
>
> and
>
> > Do you have any experiments, including synthetic ones, to show the proposed modified AdaBoost in Algorithm 1 works in practice?
>
> **Answer:**
> We thank the reviewer for the suggestion.
> As Algorithm 1 was just a subroutine of our main algorithm, we did not consider evaluating it empirically.
> Moreover, Algorithm 1 closely resembles AdaBoost as the minor changes were meant only to ensure desirable theoretical properties and ease our analysis.
> We believe this similarity would translate into good empirical performance, though a thorough experimental investigation is an interesting direction for future work.
>
>
> > Can you further discuss how this work relates to / extends Hopkins et al. (2024) (in line 195)?
>
> **Answer:**
> The seminal work of Hopkins et al. (2024) develops methods for several learning settings, including malicious noise, semi private-learning, and partial learning.
> Our work focuses on the agnostic boosting setting, which compared to their baseline algorithm for agnostic learning, does not assume access to the reference class $\mathcal{F}$ (or the base class $\mathcal{B}$), but  merely access to a weak learner that can learn from the base class $\mathcal{B}$.
> The agnostic algorithm of Hopkins et al. generates all possible labellings that the reference class $\mathcal{F}$ could have produced, which as **Reviewer tVkc** also pointed out (see the reply to **Reviewer tVkc** for further details) would yield a bag in the first step ($\mathcal{B}\_{1}$) of size would be $O((d'/m)^{d'})$.
> From this, one could use the guarantee that $\mathcal{B}\_{1}$ contains a good hypothesis together with a Hoeffding and union bound to conclude that the empirical risk minimizer in $\mathcal{B}\_1$, $v\_g$, has error $\mathcal{L}\_{\mathcal{D}}(v\_{g}) = \mathrm{err}\_{\mathcal{D}}(f^{\star}) + O\left(\sqrt{\frac{d'\ln(m/d') + \ln(\delta)}{m}}\right) + \tilde{O}\left(\frac{d}{\gamma^2 m} + \frac{\ln(1/\delta)}{n}\right)$.
> Yet, as we do not assume access to a way to generate all possible labellings of the training set by $\mathcal{F}$---in fact, we don't assume any access to $\mathcal{F}$---we instead generate all possible labellings of the training data.
> An alternative we also considered was to take all possible subsets of the training set, but that would lead to the same number of runs, but with a smaller effective sample size.
> Hopkins et al. also consider the approach of calling the realizable learning algorithm on all the training subsets.
> For instance, in their Theorem 54 for partial concept classes, where the upper bound on the sample complexity is $O\left(\frac{n(\Theta(\varepsilon), \Theta(\delta)) + \ln(1/\delta)}{\varepsilon^{2}}\right)$ (omitting $\ln(1/\delta)$), where $n(\varepsilon, \delta)$ is the sample complexity of learning the partial concept class $\mathcal{F}$.
> The best known bound for learning partial classes is $\tilde{O}(d/\varepsilon)$ ([3]) which yields a bound of order $\tilde{O}\left(\frac{d}{\varepsilon^{3}}\right)$.
> The strategy of obtaining a learner by running the realizable learner on all subsets and picking the best one is also used in recent work by [4] in the multiclass agnostic setting, which yields a sample complexity of $O(1/\varepsilon^{3})$ in terms of the dependency of $\varepsilon$ (cf. Corollary 4.3).
>
>
> > The proposed algorithm is computationally inefficient and could be hard to implement in practice.
>
> and
>
> > As one limitation is the algorithm's computational inefficiency steming from iterating over all possible re-labelings. Are there any foreseeable approaches or heuristics to make this algorithm (or a similar one) computationally feasible while substantially preserving its strong statistical guarantees?
>
> **Answer:**
> We thank the reviewer for raising this important point.
> Currently, it is unclear to us how to extend the algorithm to be computationally efficient while preserving the same theoretical guarantees on the sample complexity.
> However, our algorithm and analysis do suggest several heuristic directions worth exploring.
>
> One possible approach is to divide the training data into smaller subsets and run AdaBoost separately on each subset.
> If the resulting classifier achieves good margins on its respective subset, one could then iteratively merge it with it's neighboring subset (which also has good margins), re-run AdaBoost on this merged subset, and continue this procedure until all the sets are merged or removed.
> However, this method does not come with any guarantees and may not work reliably in general.
>
> Another potential heuristic is to run AdaBoost on the full dataset for a fixed number of iterations, then identify the points with the most negative margins and flip their label.
> AdaBoost could then be trained again on this altered data set.
> This process can then be repeated for a fixed number of rounds.
> While intuitive, this approach also lacks a formal guarantees.
>
> Finally, one might consider a modification of Algorithm 2 in which lines 3–7 are replaced by a parametrized re-labeling strategy, where the user specifies the number of re-labelings to consider.
> This could make the algorithm more scalable in practice.
> However, to make such an approach effective, one would need a reliable mechanism for selecting promising re-labelings, a component we currently do not have.
> As such, this alternative too would lack theoretical guarantees.
>
> In summary, while several heuristics can be inspired by our algorithm, their sample complexity remain open questions.
> We hope our work provides a foundation for further research in this direction.
>
>
> [3] Optimal PAC Bounds Without Uniform Convergence Ishaq Aden-Ali, Yeshwanth Cherapanamjeri, Abhishek Shetty, Nikita Zhivotovskiy 2023
>
> [4] Representation Preserving Multiclass Agnostic to Realizable Reduction Steve Hanneke, Qinglin Meng, Amirreza Shaeiri 2025

---

### Official Review · Reviewer_tFLU · 2025-07-05

**Clarity:** 3
**Significance:** 3
**Originality:** 3
**Rating:** 4
**Confidence:** 3

**Summary:**

The paper introduces a novel agnostic boosting algorithm that significantly improves sample complexity under general assumptions. The proposed method reduces the problem to the realizable case and employs a margin-based filtering step to select high-quality hypotheses. Additionally, the paper establishes a lower bound on the sample complexity of agnostic boosting, matching the upper bound up to logarithmic factors.

**Questions:**

see my comments above.

**Ethical Concerns:**

["NO or VERY MINOR ethics concerns only"]

**Final Justification:**

Thanks for the efforts in rebuttal. I remain my support to this paper :)

**Quality:**

3

**Strengths And Weaknesses:**

**Strengths**:
(1) The paper presents a new agnostic boosting algorithm that offers substantial improvements in sample complexity compared to previous works, which is a novel contribution to the community.

(2) The authors provide thorough and solid theoretical analysis, including a lower bound on the sample complexity that aligns closely with the upper bound.

(3) The approach is based on very general assumptions, making it broadly applicable across different scenarios, which is indeed an advantage of the paper.

(4) The paper includes detailed proofs and theoretical justifications for the proposed algorithm and its properties, which facilitates people's understanding of the proposed method.

**Weaknesses**:
(1) While the algorithm is statistically near-optimal, I am not sure about its computationally efficiency, which may limit its practical applicability.

(2) I understand this paper is mainly focused on the theory. Although not always required, it might be a bit beneficial if the paper also provides empirical results or experiments to validate the theoretical findings and demonstrate the algorithm's performance in practice. For example, some other boosting-related theoretical papers also have some (though not too many) experiments included in their papers (see references 1-3 from the list below).

**Reference**:
[1] Ghai, et al. "Sample-Efficient Agnostic Boosting." arXiv preprint arXiv:2410.23632 (2024).
[2] Liu, et al. "An analysis of boosted linear classifiers on noisy data with applications to multiple-instance learning." 2017 IEEE International Conference on Data Mining (ICDM). IEEE, 2017.
[3] Bootkrajang, et al. "Boosting in the presence of label noise." arXiv preprint arXiv:1309.6818 (2013).

---

> ### Author Rebuttal · Authors · 2025-07-31
>
> We thank the reviewer for taking the time to read our paper and provide feedback.
>
> We address the reviewers questions/comments below:
>
> > While the algorithm is statistically near-optimal, I am not sure about its computationally efficiency, which may limit its practical applicability.
>
> **Answer:**
> We thank the reviewer for raising this important point.
> Indeed, our primary focus is not on computational efficiency, but rather on the statistical aspects of agnostic boosting.
> Our main contribution is to demonstrate that it is possible to achieve near-optimal sample complexity even under the very general assumption of black-box access to a weak learner, without requiring knowledge of the reference class or any additional structural assumptions on the problem.
> This level of generality is not only theoretically appealing: in typical boosting applications, one runs algorithms such as AdaBoost with a fixed base class (e.g., decision stumps), without knowing which reference class is effectively being weak learned by the combination of base learner and data.
> Our results show that strong statistical guarantees are achievable even in this practically realistic setting, where the reference class remains implicit.
>
> While we acknowledge that the resulting algorithm may not yet be computationally efficient or practical in all settings, we view our work as an important theoretical step that clarifies what is statistically achievable in principle.
> We hope it will inspire further research aimed at developing computationally efficient instantiations or approximations guided by these insights.
>
>
> > I understand this paper is mainly focused on the theory. Although not always required, it might be a bit beneficial if the paper also provides empirical results or experiments to validate the theoretical findings and demonstrate the algorithm's performance in practice. For example, some other boosting-related theoretical papers also have some (though not too many) experiments included in their papers (see references 1-3 from the list below).
>
> **Answer:**
> We appreciate the reviewer’s thoughtful suggestion regarding the inclusion of empirical results.
> However, our paper is intentionally focused on the theoretical aspects of boosting, specifically on the sample complexity of the problem.
> While we acknowledge that certain heuristics could be derived from our theoretical insights (see answer to **Reviewer aHg7** (search for `[TAG1]`)), we currently lack a rigorous understanding of their behavior.
> As such, incorporating empirical results at this stage risks being ad hoc and could distract from the rigorous and theoretical nature of our contribution.
>
> We agree that a meaningful empirical evaluation is, in principle, valuable and feasible.
> However, designing such experiments carefully, especially in a way that faithfully reflects the theoretical framework and explores the appropriate range of algorithmic instantiations, requires a non-trivial amount of time and effort.
> Given the scope and time constraints of the rebuttal phase, we believe it would not be appropriate to rush this process or include preliminary experiments that may not do justice to the theoretical foundation we aim to establish.
>
> We hope the reviewer will understand our decision to prioritize a focused theoretical exposition, and see potential for future work to build on our results with a careful experimental study.

---

### Official Review · Reviewer_aHg7 · 2025-07-05

**Clarity:** 3
**Significance:** 3
**Originality:** 3
**Rating:** 5
**Confidence:** 3

**Summary:**

This paper introduces a new boosting algorithm in the agnostic setting where labels are arbitrary, with no realizability assumption. Their method reduces agnostic boosting to the realizable case through a margin‑based filtering step that selects high‑quality hypotheses, achieving near-optimal sample complexity (up to logarithmic factors). This paper focuses only on statistical and information-theoretic
aspects of the problem, without constructing a computationally efficient algorithm.

**Questions:**

1. The proposed algorithm is acknowledged to be computationally inefficient. To what extent can the current theoretical results be extended to efficient algorithms without significant degradation in sample complexity?
2. The lower bound (Theorem 1.2) includes an extra logarithmic factor. Is this believed to be an artifact of the proof, or is there any evidence that it reflects an actual complexity barrier?
3. I suggest that the authors put a table summarizing the existing works and the proposed method. The table can include sample complexity, whether it is computationally efficient, and any other notable things, so that the readers can more easily understand the current status of this problem.
4. What is $D_t^{m_0}$ in Algorithm 1? It is not clearly defined.

**Ethical Concerns:**

["NO or VERY MINOR ethics concerns only"]

**Final Justification:**

The authors addressed most of my questions. I will keep my original positive rating.

**Limitations:**

Yes.

**Paper Formatting Concerns:**

No paper formatting concerns.

**Quality:**

3

**Strengths And Weaknesses:**

**Strengths**

- Boosting in the agnostic setting is a problem known to be challenging. The proposed boosting algorithm, using the technique from a recent work to reduce the agnostic problem to the realizable one, is a fresh approach, offering tighter sample complexity guarantees in the agnostic setting.
- The authors adopt a highly general notion of agnostic weak learners that encompasses many variants from prior literature, providing a unifying framework and fair comparison.
- This paper also proves a lower bound on the problem, demonstrating that the proposed method achieves the optimal sample complexity bound up to logarithmic factors.
- The paper is clearly written with enough information about the background and existing results. The mathematical notations are a bit dense, but the authors did a good job explaining the intuitions.

**Weakness**
- The proposed algorithm is not computationally efficient due to the exhaustive relabeling step and enumeration over exponentially many hypotheses. While the authors acknowledge this, some discussion on possible heuristics or approximations would be helpful.
- While the bounds are tight asymptotically, the constants hidden in the big-O notation may be large due to the fat-shattering terms. There is no discussion on this.

---

> ### Author Rebuttal · Authors · 2025-07-31
>
> We thank the reviewer for taking the time to read our paper and provide feedback.
>
> We address the reviewer's comments in order:
>
> > The proposed algorithm is acknowledged to be computationally inefficient. To what extent can the current theoretical results be extended to efficient algorithms without significant degradation in sample complexity?
>
> **[TAG1]** (just a reference for other reviewers)
>
> **Answer:** (This is also addressing the first point under the weakness section of the review)
> We thank the reviewer for raising this important point.
> Currently, it is unclear to us how to extend the algorithm to be computationally efficient while preserving the same theoretical guarantees on the sample complexity.
> However, our algorithm and analysis do suggest several heuristic directions worth exploring.
>
> One possible approach is to divide the training data into smaller subsets and run AdaBoost separately on each subset.
> If the resulting classifier achieves good margins on its respective subset, one could then iteratively merge it with it's neighboring subset (which also has good margins), re-run AdaBoost on this merged subset, and continue this procedure until all the sets are merged or removed.
> However, this method does not come with any guarantees and may not work reliably in general.
>
> Another potential heuristic is to run AdaBoost on the full dataset for a fixed number of iterations, then identify the points with the most negative margins and flip their label.
> AdaBoost could then be trained again on this altered data set.
> This process can then be repeated for a fixed number of rounds.
> While intuitive, this approach also lacks a formal guarantees.
>
> Finally, one might consider a modification of Algorithm 2 in which lines 3–7 are replaced by a parametrized re-labeling strategy, where the user specifies the number of re-labelings to consider.
> This could make the algorithm more scalable in practice.
> However, to make such an approach effective, one would need a reliable mechanism for selecting promising re-labelings, a component we currently do not have.
> As such, this alternative too would lack theoretical guarantees.
>
> In summary, while several heuristics can be inspired by our algorithm, their sample complexity remain open questions.
> We hope our work provides a foundation for further research in this direction.
>
>
> > While the bounds are tight asymptotically, the constants hidden in the big-O notation may be large due to the fat-shattering terms. There is no discussion on this.
>
> **Answer:**
> We thank the reviewer.
> This is a good point.
> We will add a discussion of this nuance to the text making it clear that the constants in the O-notation are large and that we did not try to optimize for them.
>
>
> > The lower bound (Theorem 1.2) includes an extra logarithmic factor. Is this believed to be an artifact of the proof, or is there any evidence that it reflects an actual complexity barrier?
>
> **Answer:**
> We strongly believe this is an artifact of the proof.
> The reasoning is that one would expect a tight bound to hold for all values of $\mathrm{err}_{\mathcal{D}}(f^{\star})$, including 0 (the realizable case).
> However, adapting our Lemma E.2 to the realizable setting would still yield that $\mathcal{F}$ has VC-dimension $\Theta(\frac{d}{\ln(1/\delta) \gamma^{2}})$, so one could obtain (e.g., from Theorem 6.8 in [1]) the lower bound of $\Omega\left(\frac{d}{\ln(1/\delta) \gamma^{2}\varepsilon} + \frac{\ln(1/\delta)}{\varepsilon}\right)$ on the sample complexity of realizable boosting.
> If this was tight, it would contradict the known optimal bound of $\Theta\left(\frac{d}{\gamma^{2}\varepsilon} + \frac{\ln(1/\delta)}{\varepsilon}\right)$ (no extra logarithmic factor) by [2].
>
>
>
> > I suggest that the authors put a table summarizing the existing works and the proposed method. The table can include sample complexity, whether it is computationally efficient, and any other notable things, so that the readers can more easily understand the current status of this problem.
>
> **Answer:**
> We see the point in the reviewer's suggestion.
> In fact, we considered adding such a table but decided against it as we thought it could be misleading with regard to the different definitions of weak learner.
> Given that the situation is, rigorously, quite nuanced, we preferred to keep only explicit and detailed discussions in Section 1.1 (Related Works) and, more thoroughly, in Appendix A.
>
> Nonetheless, as you and Reviewer tVkc have suggested this, if you maintain your preference after taking our reply into account, we would not object to adding such a table (accompanied by a clear remark of its limitations).
>
>
> > What is $\mathbf{D}_{t}^{m_{0}}$ in Algorithm 1? It is not clearly defined
>
> **Answer:**
> We thank the reviewer for pointing this out.
> We indeed overlooked this notation in our definitions.
> We'll add to Section 2 that given a distribution $\mathcal{D}$ and an integer $m$, sampling according to $\mathcal{D}^m$ means taking $m$ i.i.d. samples from $\mathcal{D}$.
> In this specific case $m_0$ is the number of samples required by the weak learner to produce a good hypothesis (cf. Definition 1.1) and $\mathcal{D}_t$ is the $t^\text{th}$ distribution generated by Algorithm 1.
>
>
> [1] Understanding Machine Learning: From Theory to Algorithms. Shai Shalev-Shwartz and Shai Ben-David. 2014.
>
> [2] Optimal Weak to Strong Learning. Kasper Green Larsen and Martin Ritzert. 2022.

---

> > ### Comment · Reviewer_aHg7 · 2025-08-01
> >
> > Thank you for the response, which addressed most of my questions. I will keep my original positive rating.

---

### Decision · Program_Chairs · 2025-09-17

**Decision:**

Accept (poster)

**Comment:**

(a) Summary: The paper studies the sample complexity of the agnostic boosting problem under general assumptions. There are two main contributions: a sample complexity lower bound for agnostic boosting, and a nearly matching upper bound achieved by an inefficient algorithm. The upper bound significantly improves upon prior works, and the proposed algorithm is novel, incorporating recent results that reduce agnostic learning to realizable learning. The paper's focus is entirely on the statistical and information-theoretic aspects, not on computational efficiency.

(b) Strengths:
- Significant theoretical contribution: The paper establishes nearly matching upper and lower bounds on the sample complexity of agnostic boosting. As noted by Reviewers aHg7, tFLU, and jF1w, this is a major advance in the fundamental understanding of a classic and challenging machine learning problem.
- Novelty and soundness: The approach of using an "all possible labelings" technique based on recent work to reduce the agnostic case to the realizable one is novel. All reviewers consider the theoretical analysis sound and of high quality.
- The paper is very well-written and organized.

(c) Weaknesses:
- Computational inefficiency: The proposed algorithm is exponential time and hence not practical. The algorithm requires iterating over all possible re-labelings of the training data, which is exponential time in the sample size. All reviewers considered this as a major limitation.

(d) Reasons for recommendation: This paper makes valuable theoretical contributions to studying the sample complexity of agnostic boosting. Even though the algorithm is inefficient, the sample complexity improvements are notable and the technique is novel. I recommend acceptance.

(e) Summary of rebuttal discussion: The authors provided detailed responses to the reviewers in the rebuttal period, and the reviewers have positive or borderline final scores.
- On computational inefficiency: the authors agreed their algorithm is inefficient and clarified that their goal was to establish statistical results, not to propose a practical algorithm. They suggested several heuristics for future work but acknowledged these do not have theoretical guarantees.
- On lack of experiments: the authors emphasized the paper's theoretical focus, stating that adding ad-hoc experiments could detract from their contribution. They stated that a meaningful empirical study would be non-trivial and out of scope for the rebuttal.